PREREGISTERED RESEARCH ARTICLE

# Longitudinal single-subject neuroimaging study reveals effects of daily environmental, physiological, and lifestyle factors on functional brain connectivity

**Ana María Triana** [1,2]*, **Juha Salmi** [2,3,4,5], **Nicholas Mark Edward Alexander Hayward** [2], **Jari Saramäki** [1], **Enrico Glerean** [2,6]

1 Department of Computer Science, School of Science, Aalto University, Espoo, Finland, 2 Department of Neuroscience and Biomedical Engineering, School of Science, Aalto University, Espoo, Finland, 3 Aalto Behavioral Laboratory, Aalto Neuroimaging, Aalto University, Espoo, Finland, 4 MAGICS, Aalto Studios, Aalto University, Espoo, Finland, 5 Unit of Psychology, Faculty of Education and Psychology, Oulu University, Oulu, Finland, 6 Advanced Magnetic Imaging Centre, Aalto University, Espoo, Finland

* ana.trianahoyos@aalto.fi

**Academic Editor:** Laura D. Lewis, Massachusetts Institute of Technology, UNITED STATES OF AMERICA; Christopher D. Chambers, Cardiff University, UNITED KINGDOM OF GREAT BRITAIN AND NORTHERN IRELAND

**Note:** As this is a Preregistered Research Article, the study design and methods were peer-reviewed before data collection. The time to acceptance includes the experimental time taken to perform the study. Learn more about Preregistered Research Articles.

## Abstract

Our behavior and mental states are constantly shaped by our environment and experiences. However, little is known about the response of brain functional connectivity to environmental, physiological, and behavioral changes on different timescales, from days to months. This gives rise to an urgent need for longitudinal studies that collect high-frequency data. To this end, for a single subject, we collected 133 days of behavioral data with smartphones and wearables and performed 30 functional magnetic resonance imaging (fMRI) scans measuring attention, memory, resting state, and the effects of naturalistic stimuli. We find traces of past behavior and physiology in brain connectivity that extend up as far as 15 days. While sleep and physical activity relate to brain connectivity during cognitively demanding tasks, heart rate variability and respiration rate are more relevant for resting-state connectivity and movie-watching. This unique data set is openly accessible, offering an exceptional opportunity for further discoveries. Our results demonstrate that we should not study brain connectivity in isolation, but rather acknowledge its interdependence with the dynamics of the environment, changes in lifestyle, and short-term fluctuations such as transient illnesses or restless sleep. These results reflect a prolonged and sustained relationship between external factors and neural processes. Overall, precision mapping designs such as the one employed here can help to better understand intraindividual variability, which may explain some of the observed heterogeneity in fMRI findings. The integration of brain connectivity, physiology data and environmental cues will propel future environmental neuroscience research and support precision healthcare.

## Introduction

Every day, we wake up as a slightly different person, as our mental states are influenced by many external factors. The quality of sleep, the level of physical activity, and the nature of our

**Data Availability Statement:** Data collected from the experiment are available for the scientific community at https://zenodo.org/records/10571956. Since the data for this study are personal data, they are subject to the General Data Protection Regulation (GDPR). For this reason, access to the data can be requested from the institution data stewards (researchdata@aalto.fi) and a data transfer agreement contract must be stipulated between the organization who is the controller of the personal data (Aalto University) and the recipient institution. If the recipient country is not within the EU/EEA, the recipient must also provide a level of data security that is compatible with what is required by the GDPR. The data will be provided with a data processing agreement that the recipient shall also accept before initiating any data transfer. Data collected from the experiment are shared in a similar fashion across the scientific community for research purposes. All individual quantitative observations underlying the data summarized in the figures and results are available in two public repositories. Pilot data is available in the Zenodo release at https://doi.org/10.5281/zenodo.13208496. Unprocessed study data can be found in the Zenodo dataset release at https://zenodo.org/records/10571956. Processed results derived from the study data are accessible in the GIT repository at https://zenodo.org/doi/10.5281/zenodo.13208811.

**Funding:** AMT acknowledges the support from the Ella and Georg Ehrnrooth foundation. https://www.ellageorg.fi/en. The Aalto Brain Centre, Aalto University provided funding for MRI data collection. https://www.aalto.fi/en/school-of-science/aalto-brain-centre. JPS acknowledges support from the Research Council of Finland (project NetResilience, grant numbers 345188 and 345183). https://www.aka.fi/en/strategic-research/. JS acknowledges support from the Research Council of Finland (project Bringing real-life to attention, grant numbers 325981, 328954, and 353518). https://www.aka.fi/en/strategic-research/. The funders had no role in study design, data collection and analysis, decision to publish or preparation of the manuscript.

**Competing interests:** The authors have declared that no competing interests exist.

social interactions all affect the state of our brains at different timescales. These timescales range from milliseconds (rapid detection of sounds [1]) to seconds (preparation for motor action [2,3]), minutes (mood changes [4]), and days (fluctuations in attentional state) [5]. Thus, different timescales reveal different aspects of brain dynamics. For example, brain areas and networks are engaged differently over time when performing a specific task [6], major psychiatric disorders show large fluctuations in brain function over different timescales [7], and functional brain connectivity patterns accurately track daily fluctuations in mood [8]. Hence, the timescales of both brain activity and external factors are important. However, few studies have considered brain activity to be not only a function of the cognitive and psychological characteristics of the sampled individual, but also a function of the specific moment in time when sampling the individual.

Traditionally, the relationship between behavior and the state of the brain is studied with cross-sectional designs which sample many individuals at one specific point in time. Although cross-sectional designs have contributed greatly to the understanding of the brain–behavior relationship, there are concerns that studies sampling large populations over a short period of time may not translate into similar findings within single individuals [9,10]. More specifically, it has been shown that the variance within individual measurements can be up to 4 times greater than the variance in groups [11]. This lack of group-to-individual generalizability means that it is challenging to translate cognitive neuroscience findings into practice, as almost any treatment would benefit from personalized planning [12].

In light of the above, there is a clear need for longitudinal studies with frequent measurement points to study brain–behavior relationships. Moreover, analysis of longitudinal data should consider the dependence between time points, as external factors from previous days are known to be correlated with brain measurements on the following days [13]. While large sample sizes are needed in cross-sectional designs [14], small N repeated-measures designs are also desirable due to their high power and inferential validity [15]. Designs that carefully sample a small number of individuals have already been used since the 1960s [16,17]. For example, in psychophysics, researchers traditionally conduct numerous trials on a small number of participants [18]. However, this sampling method is still overlooked in cognitive neuroscience, where researchers strive to optimize the numbers of trials and participants in order to gain sufficient statistical power for significant group averages. Since it is commonly assumed that an individual's mental states and cognitive abilities are somewhat invariant, just a few trials are considered sufficient for correctly sampling an individual's brain activity and behavior. However, this assumption falls short, prompting researchers to use larger numbers of trials per participant instead [19].

In recent years, this need for more trials within individuals has translated into an increasing number of studies that intensively collect brain activity data via functional magnetic resonance imaging (fMRI) from a few subjects (see Gratton and colleague's work [12] for a full review). This approach is known as precision functional mapping [20,21]. Precision functional mapping studies have demonstrated that individual-specific functional connectivity differs from group averaged functional connectivity [22–25], and that groups of individuals who differ in behavior share common network variants (i.e., brain regions that differ from group network organization) [26]. Moreover, precision functional mapping studies have also illustrated that the picture emerging from intensive sampling of one individual's functional brain activity can be very different from that obtained from a single session recording, after taking into account temporal variation in factors such as behavior [27], hormonal changes [28], or even migraine symptoms [29].

While these studies have provided strong proof-of-concept for the benefits of repeatedly sampling the brain activity of an individual, 2 challenges remain. Firstly, although studies

**Abbreviations:** ACC, anterior cingulate cortex; AK, Anna Karenina; ANS, autonomic nervous system; BOLD, blood oxygen-level dependent; CON, cingulo-opercular network; CSF, cerebrospinal fluid; dlPFC, dorsolateral prefrontal cortex; DMN, default mode network; EMA, ecological momentary assessment; FD, framewise displacement; FDR, false discovery rate; fMRI, functional magnetic resonance imaging; FPN, fronto-parietal network; FWHM, full width at half maximum; GLM, general linear model; GM, gray matter; GPS, global positioning system; ID-RSA, inter-daily representational similarity analysis; INU, intensity non uniformity; ISC, inter-subject correlation; IS-RSA, intersubject representational similarity analysis; mPFC, medial prefrontal cortex; NAF, negative affect; NN, nearest neighbors; PAF, positive affect; PFC, prefrontal cortex; PPG, photoplethysmography; PSQI, Pittsburgh Sleep Quality Index; PSS, Perceived Stress Scale; PVT, Psychomotor Vigilance Test; ROI, regions of interest; RT, reaction time; SD, sleep deprivation; SDC, susceptibility distortion correction; SMA, supplementary motor area; tSNR, temporal signal-to-noise ratio; VIF, variance inflation factor; WM, white matter.

report dynamic changes in individuals' brain activity at rest [8,27,28,30], less is known about these fluctuations during tasks. This is noteworthy, as difficult cognitive tasks can provide a means for minimizing common confounds such as head motion, for obtaining more robust brain networks, and for exploring how the current findings generalize to ecologically valid contexts [31–33]. Secondly, even though precision functional mapping studies have shown that external factors may also modulate brain activity [8,27,28], automatic sensors have rarely been employed to collect this type of data. Nevertheless, collecting such data is entirely feasible, owing to the development of smartphones and wearables that can be more suitable than traditional log methods for obtaining objective, quantitative data in real-life settings with minimal subject burden (see Sheikh and colleagues [34] for a review). The use of automatic sensors could not only avoid key limitations of the gold-standard behavioral methods due to subjective biases [35], but could also improve the reliability of atypical conducts [36,37]. In fact, dynamic analyses of these data have yielded promising results as markers for mental health disorders [38–40].

To address the 2 challenges discussed above, we collected a precision functional mapping data set from a single individual. This data set contains both brain activity data under a set of different fMRI tasks and objective data from external factors collected through automatic sensors. As an index of brain function, we selected functional connectivity [41] (a pattern of statistical dependencies between brain areas), as this method can be applied across all selected experimental conditions. A carefully selected set of experimental conditions, including the Psychomotor Vigilance Test (PVT) [42], an adaptive n-back [43], resting-state, and movie-watching [44] tasks, allowed us to understand the daily within-subject variability in attention, working memory, resting, and naturalistic stimuli tasks. At the same time, state-of-the-art smartphones and wearable devices made it feasible to measure (with low effort) external factors like sleep, physical activity, autonomic nervous system (ANS) activity, and mood [34]. These data offers a mean in real-life settings to answer the following research questions posed in this study:

**Q1: How do behavioral, physiological, and lifestyle factors experienced by the individual on the previous day affect today's functional brain connectivity patterns?**

**Q2: Can behavioral, physiological, and lifestyle factors influence functional connectivity beyond the previous day, and up to the preceding 15 days?**

The relationship between the aforementioned factors (i.e., sleep, physical activity, ANS activity, and mood) and functional connectivity has been investigated cross-sectionally using a wide variety of paradigms. Although the setting differs from precision functional mapping, these studies still provide valuable hints about which brain areas and external factors are generally associated at the population level, and are thus worth investigating at the individual level as well. For example, several authors have reviewed in depth the relationship between brain activity and factors, such as sleep [45–47], physical activity [48,49], ANS activity [50,51] (comprising respiration rate, heart rate, and heart rate variability), emotions, and mood [52,53]. Based on these reviews and the studies cited therein, we approached Q1 and Q2 through 8 specific hypotheses (see Table 1).

**H1: Fluctuations in sleep patterns are correlated with functional connectivity within the fronto-parietal, default mode, somatomotor, and cingulo-opercular networks during sustained attention tasks**

We chose to investigate the relationship between sleep and attentional tasks in these regions because of 4 reasons. Firstly, results from sleep studies suggest that the amount of accumulated

**Table 1. Summary table.**

| Question | Hypothesis | Sampling plan | Analysis plan | Interpretation given to different outcomes | Outcomes |
|---|---|---|---|---|---|
| Q1: How do behavioral, physiological, and lifestyle factors experienced by the individual on the previous day affect today's functional brain connectivity patterns? | H1: Fluctuations in sleep patterns are correlated with functional connectivity within the fronto-parietal, default mode, somatomotor, and cingulo-opercular networks during sustained attention tasks. | Fourteen samples minimum according to power analysis based on pilot data. | We conducted a regression analysis of the effect of total sleep time, awake time, and restless sleep on the functional brain connectivity estimates of the ROIs in the selected networks. Functional brain connectivity estimates comprise functional connectivity links and graph-theoretical measures of between-network integration and within-network integration. Statistical validity was assessed via 10,000 nonparametric permutations and a corrected $p$-value via Benjamini–Hochberg according to the number of links or the number of networks for graph-theoretical measures. | A statistically significant interaction is interpreted as evidence of a relationship between sleep factors (sleep time, awake time, and restless sleep) and link weight in the fronto-parietal, default mode, somatomotor, and cingulo-opercular networks. Similarly, a statistically significant interaction will be interpreted as evidence of a relationship between sleep factors and functional connectivity estimates (participation coefficient and global efficiency). The coefficient of determination is interpreted as the strength of the relationship between the external factor and the functional connectivity. Absence of statistical significance is interpreted as failure to reject the null hypothesis. | Hypothesis partially confirmed. Fluctuations in sleep patterns are correlated with functional connectivity within the default mode, somatomotor, and cingulo-opercular network during sustained attention tasks. No associations between sleep patterns and functional connectivity in the fronto-parietal network are found. |
| | H2: Fluctuations in sleep and physical activity patterns are correlated with functional connectivity in the default mode, fronto-parietal, and somatomotor network during working memory tasks. | Thirteen samples minimum according to power analysis based on pilot data. | We conducted a regression analysis of the effect of total sleep time, awake time, restless sleep, steps, and inactive time on the functional brain connectivity estimates of the ROIs in the selected networks. Functional brain connectivity estimates comprise functional connectivity links and graph-theoretical measures of between-network integration and within-network integration. Statistical validity was assessed via 10,000 nonparametric permutations and a corrected $p$-value via Benjamini–Hochberg according to the number of links or the number of networks for graph-theoretical measures. | A statistically significant interaction is interpreted as evidence of a relationship between sleep and physical activity factors (sleep time, awake time, restless sleep, steps, and inactive time) and link weight in the default mode, fronto-parietal and somatomotor networks. Similarly, a statistically significant interaction will be interpreted as evidence of a relationship between sleep and physical activity factors and functional connectivity estimates (participation coefficient and global efficiency). The coefficient of determination is interpreted as the strength of the relationship between the external factor and the functional connectivity. Absence of statistical significance is interpreted as failure to reject the null hypothesis. | Hypothesis confirmed. Fluctuations in sleep and physical activity patterns are correlated with functional connectivity in the default mode, fronto-parietal, and somatomotor network during working memory tasks. |
| | H3: Fluctuations in sleep, autonomic nervous system activity, and mood patterns are correlated with functional connectivity in the default mode, fronto-parietal, and cingulo-opercular networks during resting-state fMRI. | Thirty samples as similarly used in (Rosenberg and colleagues, Prischet and colleagues) [5,28] | We conducted a regression analysis of the effect of total sleep time, awake time, restless sleep, heart rate variability, respiration rate, and mood on the functional brain connectivity estimates of the ROIs in the selected networks. Functional brain connectivity estimates comprise functional connectivity links of the mentioned networks and measures of between-network integration and within-network integration. Statistical validity was assessed via 10,000 nonparametric permutations and a corrected $p$-value via Benjamini–Hochberg according to the number of links or the number of networks for graph-theoretical measures. | A statistically significant interaction is interpreted as evidence of a relationship between sleep, ANS, and mood patterns (sleep time, awake time, restless sleep, heart rate variability, respiration rate, and mood) and link weight in the default mode, fronto-parietal, and cingulo-opercular networks. Similarly, a statistically significant interaction is interpreted as evidence of a relationship between external factors and functional connectivity estimates (participation coefficient and global efficiency). The coefficient of determination is interpreted as the strength of the relationship between the external factor and the functional connectivity. Absence of statistical significance is interpreted as failure to reject the null hypothesis. | Hypothesis partially confirmed. Fluctuations in sleep and autonomic nervous system activity are correlated with functional connectivity in the default mode, fronto-parietal, and cingulo-opercular networks during resting-state fMRI. No associations between mood patterns and resting-state functional connectivity are found. |
| | H4: Increased similarity in sleep, autonomic nervous system activity, or mood between days is seen as an increase of inter-day similarity in the fronto-parietal, default mode, and salience networks during movie-watching tasks. | Thirty samples (Pajula and Tohka) [133] | We applied Inter-daily Representational Similarity Analysis (ID-RSA) between the similarity matrix from daily behavioral scores and the similarity matrix from functional brain data between different days. Because linear combinations hinder our ability to trace the sources of variation effectively, we chose to conduct the analysis separately for each external factor. This decision was taken before any linear combination model was run. ID-RSA was tested with mantel test via 10,000 permutations of the rows and columns of the similarity matrices. | A statistically significant effect is interpreted as a positive relationship between the similarity of an ROI and the similarity of the behavioral score. For example, the more similar the signal in the precuneus between 2 days, the more similar the behavioral scores for those 2 same days, if the statistical significance is above the FDR corrected $p$-value. Absence of statistical significance is interpreted as failure of proof for the alternative hypothesis. | Hypothesis rejected. No associations between brain activity and sleep, autonomic nervous system activity or mood are found. |

*(Continued)*

**Table 1.** (Continued)

| Question | Hypothesis | Sampling plan | Analysis plan | Interpretation given to different outcomes | Outcomes |
|---|---|---|---|---|---|
| Q2: Can behavioral, physiological, and lifestyle factors influence functional connectivity beyond the previous day, up to the preceding 15 days? | H5: Sleep patterns experienced over the past 15 days are correlated with functional connectivity in the fronto-parietal, default mode, somatomotor, and cingulo-opercular networks during sustained attention tasks. | Fourteen samples minimum according to power analysis based on pilot data. | We conducted a lagged cross correlation analysis between total sleep time, awake time, and restless sleep in the past 15 days and the functional brain connectivity estimates the selected networks. Functional brain connectivity estimates comprise graph-theoretical measures of between-network integration and within-network integration. Statistical validity was assessed via 10,000 nonparametric permutations by synthesizing surrogate behavioral data. | A statistically significant correlation for a specific lag is interpreted as evidence that fluctuations in sleep are related to fluctuations in the global efficiency or participation coefficient for the selected networks. The set of lags determined the timescales that link functional connectivity to behavioral factors. The cross-correlation coefficient is interpreted as the strength of the relationship between the external factor and the functional connectivity. The sign of the cross-correlation coefficient is interpreted as the relationship between the external factor and the functional connectivity. Absence of statistical significance is interpreted as failure to reject the null hypothesis. | Hypothesis confirmed. Sleep patterns experienced over the past 15 days are correlated with functional connectivity in the fronto-parietal, default mode, somatomotor, and cingulo-opercular networks during sustained attention tasks. |
| | H6: Sleep and physical activity patterns experienced over the past 15 days are correlated with functional connectivity in the default mode, fronto-parietal, and somatomotor networks during working memory tasks. | Thirteen samples minimum according to power analysis based on pilot data. | We conducted a lagged cross correlation analysis between total sleep time, awake time, restless sleep, steps, and inactive time in the past 15 days and the functional brain connectivity estimates of the ROIs in the selected networks. Functional brain connectivity estimates comprise graph-theoretical measures of between-network integration and within-network integration. Statistical validity was assessed via 10,000 nonparametric permutations by synthesizing surrogate behavioral data. | A statistically significant correlation for a specific lag is interpreted as evidence that fluctuations in sleep and activity are related to fluctuations in the global efficiency or participation coefficient for the selected networks. The set of lags determined the timescales that link functional connectivity to behavioral factors. The cross-correlation coefficient is interpreted as the strength of the relationship between the external factor and the functional connectivity. The sign of the cross-correlation coefficient is interpreted as the relationship between the external factor and the functional connectivity. Absence of statistical significance is interpreted as failure to reject the null hypothesis. | Hypothesis confirmed. Sleep and physical activity patterns experienced over the past 15 days are correlated with functional connectivity in the default mode, fronto-parietal, and somatomotor networks during working memory tasks. |
| | H7: Sleep, autonomic nervous system activity, and mood patterns experienced over the past 15 days are correlated with functional connectivity in the default mode, fronto-parietal, and cingulo-opercular networks during resting-state fMRI. | Thirty samples as similarly used in (Rosenberg and colleagues, Prischet and colleagues) [5,28] | We conducted a lagged cross correlation analysis between total sleep time, awake time, restless sleep, respiration rate, heart rate variability, and mood in the past 15 days and the functional brain connectivity estimates of the ROIs in the selected networks. Functional brain connectivity estimates comprise measures of between-network integration and within-network integration. Statistical validity was assessed via 10,000 nonparametric permutations by synthesizing surrogate behavioral data. | A statistically significant correlation is interpreted as evidence that fluctuations in sleep, autonomic nervous activity, or mood are related to fluctuations in the within-network or between-network connectivity. The set of lags determined the timescales that link functional connectivity to behavioral factors. The cross-correlation coefficient is interpreted as the strength of the relationship between the external factor and the functional connectivity. The sign of the cross-correlation coefficient is interpreted as the relationship between the external factor and the functional connectivity. Absence of statistical significance is interpreted as failure to reject the null hypothesis. | Hypothesis confirmed. Sleep, autonomic nervous system activity, and mood patterns experienced over the past 15 days are correlated with functional connectivity in the default mode, fronto-parietal, and cingulo-opercular networks during resting-state fMRI. |
| | H8: Between-days time-segment classification accuracy is explained by daily behavioral, physiological, and lifestyle factors. | Thirty samples (Pajula and Tohka) [133] | We conducted a regression analysis of the total sleep time, awake time, restless sleep, mood, respiration rate, and heart rate variability on the classification accuracy brain maps. The classification brain maps were obtained following Visconti di Oleggio Castello and colleagues. For each 10-mm sphere, we divided the data in 25TR segments with a slide window of 1TR. Then, we trained a classifier calculating the pairwise similarity between the left-out segment of the left-out run and the average signal from the segments of the other runs. Statistical validity was assessed via FSL *randomise* over 10,000 permutations. This hypothesis is exploratory, other classification approaches were also tried (e.g., k-fold cross validation instead of leave-one out). | A statistically significant interaction is interpreted as evidence that a decrease in sleep, autonomic system variables, or mood are related to decreased classification accuracy. The coefficient of determination is interpreted as the strength of the relationship between the external factor and the classification accuracy. Absence of statistical significance is interpreted as failure to reject the null hypothesis. | Hypothesis confirmed. Between-days time-segment classification accuracy during movie-watching tasks is explained by sleep, mood, and autonomic nervous system activity experienced over the past 15 days |

awake time is associated with the performance in attentional tasks [54,55], thus making attentional maintenance more variable and inconsistent [56]. Secondly, such variability is manifested in errors of omission (i.e., failure to respond in a timely manner or attention lapses) and errors of commission (i.e., response to stimuli that are not present). These errors can be detected by specialized, reliable, and valid tests such as the PVT [46,57] which is simple enough to avoid learning effects and is sensitive to sleep loss [58]. Thirdly, brain studies have shown that sleep deprivation (SD) is positively correlated with decreased activity in the prefrontal cortex (PFC), visual, parietal, and premotor areas during attention tasks [59–62]. Sleep loss also affects the thalamus although not uniformly [47]. In the sleep-rested state, sustained ascending arousal input from the thalamus supports a reciprocal inhibition between the fronto-parietal network (FPN) and default mode network (DMN), which becomes erratic at sleep-deprived states [47]. Finally, sleep duration, onset and offset can be reliably recorded using wearables [63].

### H2: Fluctuations in sleep and physical activity patterns are correlated with functional connectivity within the default mode, fronto-parietal, and somatomotor networks during working memory tasks

Similarly to H1, our choices are motivated by 4 reasons. Firstly, sleep and physical activity are known to affect working memory [47]. While sleep deprivation causes deficits in working memory performance [64,65], physical activity improves working memory function [66,67]. Secondly, such working memory function can be measured using the n-back task [68] by measuring the accuracy of responses and reaction times [65]. Thirdly, previous studies have shown that during working memory tasks, connectivity in the FPN, DMN, and supplementary motor area (SMA) is affected by sleep [69,70] and physical activity [71]. Moreover, the type of physical activity influences activations in the anterior cingulate cortex (ACC) and SMA [72]. Finally, both external factors can be reliably measured using existing wearables [63,73,74].

### H3: Fluctuations in sleep, autonomic nervous system activity, and mood patterns are correlated within functional connectivity in the default mode, fronto-parietal, and cingulo-opercular networks during resting-state fMRI

We chose to study the link between sleep and ANS activity during the resting-state task in these regions because of the following reasons. Firstly, rs-fMRI paradigms are widely used to study the brain due to the low effort required from the subjects and simple signal acquisition [75]. Secondly, evidence suggests that brain activity in the absence of tasks (i.e., resting state) is affected by sleep [76–78], mood [79], and ANS functions (e.g., heart rate variability [50], breathing, and heart rate [80–82]. In fact, changes in the ANS can influence the blood oxygen-level dependent (BOLD) fMRI signal and elicit large fluctuations in the fMRI time series [79,82]. Thirdly, previous studies have shown several brain regions to be associated with sleep and ANS activity during the resting state. For example, functional connectivity within the DMN, insula, and intraparietal sulcus are known to be affected by sleep deprivation [76,77]. Moreover, functional connectivity between ACC, basal ganglia, thalamus, amygdala, midbrain, dorsolateral prefrontal cortex (dlPFC), and medial prefrontal cortex (mPFC) is correlated with heart rate variability [83–85]. Further, activity in the pons, thalamus, striatum, periaqueductal gray matter, hypothalamus, hippocampus, SMA, motor areas, and parietal cortices is related to breathing rate [86]. Note that most of these studies collect ANS data simultaneously with fMRI or for a very short period of time outside the scanner. This clearly differs from our study, in which we measured physiological markers of ANS activity inside and outside the scanner for a prolonged period of time. Finally, similarly to H1 and H2, sleep and ANS activity can be recorded using wearables [63,87].

**H4: Increased similarity in the sleep, autonomic nervous system activity, or mood patterns between days is reflected as an increase in inter-day similarity within the fronto-parietal, default mode, and salience networks during movie-watching tasks**

There are 3 reasons for formulating this hypothesis. Firstly, naturalistic tasks such as movie-watching offer a good trade-off for improving ecological validity and reducing vulnerability to confounds [88,89]. Secondly, although the effect of external factors has been less studied in these naturalistic tasks [88], there is evidence that sleep deprivation is correlated with decreased activation in the fronto-parietal and visual regions during visual selective attention tasks [60,90], and that physiological changes are correlated with different stimuli within the movies [91,92]. Regarding mood, Lyndon-Staley and colleagues [93] found that increased sadness is correlated with connectivity in the FPN and DMN. In a precision functional mapping study, Mirchi and colleagues [8] found that positive mood is related to the integration of the brain, while negative mood relates to segregation measurements. Furthermore, Nummenmaa and colleagues [94] have found that negative valence is associated with increased inter-subject correlation (ISC) in the thalamus, ventral striatum, insula, and in the DMN. Thirdly, similar to the other hypotheses, the effects of sleep and ANS were tested using wearables [63,87,95]. For mood assessment, we used ecological momentary assessments (EMAs) on smartphones [96].

While the prespecification of the hypotheses H1–H4 and their analyses makes this registered report ideally suited for confirming and extending previous research results, the richness of the data collected in this study should allow for further exploratory analyses [97], in particular for studying timescales (Q2). We approach the more exploratory question Q2 through 4 specific hypotheses.

**H5: Sleep patterns experienced over the past 15 days are correlated with functional connectivity in the fronto-parietal, default mode, somatomotor, and cingulo-opercular networks during sustained attention tasks**

We investigated how sleep affects functional connectivity on different timescales in terms of days and weeks based on 4 reasons. Firstly, previous precision functional mapping research has shown that sleep duration during the previous week is related to brain cortical thickness, and that these effects are stronger from the second to third night before measurement [13]. Secondly, it has been demonstrated that sustained attention networks are sensitive to within-subject variability at different timescales, including days and weeks [5]. Thirdly, the effects of sleep on the PVT task have been studied longitudinally, revealing a significant influence of the duration of sleep on alertness, a long-timescale modulating influence of sleep on vigilance performance, and significant co-deterioration in the sleep patterns and cognitive performance across long periods of sleep instability [98–100]. Finally, research on the longitudinal effects of sleep on functional connectivity has highlighted the association between poor sleep quality and a decrease in functional connectivity [101]. These findings lead us to hypothesize that variation in sleep patterns will correlate with vigilance performance, which modulates functional connectivity during attention tasks over days or even weeks. Therefore, building on H1, we chose to investigate further time-lagged cross-correlations of sleep and functional connectivity.

**H6: Sleep and physical activity patterns experienced over the past 15 days are correlated with functional connectivity in the default mode, fronto-parietal, and somatomotor networks during working memory tasks**

Similar to H5, we leveraged the previous hypothesis (H2) for investigating the influence of sleep and physical activity on functional connectivity across different timescales, taking into

account 3 factors. Firstly, the evidence of longitudinal effects of sleep and physical activity on improving working memory performance [102,103], and the evidence of a close relationship between motor and working memory development [104]. Secondly, better n-back performance has been shown to be positively correlated with increased sleep and physical activity [105,106]. Finally, research studying the longitudinal effects of sleep and physical activity on functional connectivity suggest that the connectivity of a set of brain regions increases when performing a working memory task following low sleep episodes [107].

### H7: Sleep, autonomic nervous system activity, and mood patterns experienced over the past 15 days are correlated with functional connectivity in the default mode, fronto-parietal, and cingulo-opercular networks during resting-state fMRI

Following the rationale of H5 and H6, we further investigate the effects of sleep, ANS activity, and mood on resting-state functional connectivity at different timescales. We leveraged previous longitudinal findings that have demonstrated that for rs-fMRI, increased sleep regularity is associated with a more efficient network structure, better sleep quality is associated with an increase in FC, and heart rate variability is associated with the functional connectivity trajectory over time when recovering from a head injury [101,108]. Similarly, using data from MyConnectome, Mirchi and colleagues [8] demonstrated that resting-state functional connectivity patterns can be used to track daily fluctuations in mood, with positive mood being marked by an integrated architecture of functional connectivity.

### H8: Between-days time-segment classification accuracy is explained by daily behavioral, physiological, and lifestyle factors

In machine learning approaches in neuroimaging, a classifier is often trained on data from N-1 individuals and its accuracy is then tested with the subject that is left out and the average classification accuracy is reported. However, any observed interindividual differences in classification accuracy in cross-sectional studies might depend on multiple confounding factors, which are difficult to disentangle. For example, one cannot tell if the low classification accuracy for a subject is due to the brain anatomy and function per se or whether it is confounded with a subject's state at the time of the measurement related to behavior. These include whether the subject slept well the night before the scanning session and the subject's phenotype (height, weight, etc.). Moreover, phenotypic factors have also been linked to in-scanner confounding variables such as spatiotemporal patterns of head motion [109], making the challenge even more difficult. Previous studies have tested the accuracy of a classifier which identified consistent spatiotemporal patterns for movie segments of 15 s using hyperalignment [110] to minimize interindividual differences and maximize classification accuracy [111]. Findings show highest classification accuracy in primary sensory areas as well as in areas related to the theory of mind, which are important during movie watching with social cues.

With multiple scans of a single subject, we could control for phenotypical, anatomical, and functional brain differences, so that the classification accuracy for each scanning session could then be directly related to daily fluctuations in behavioral, physiological, and lifestyle factors. As this type of analysis has not been conducted in prior studies, we kept this as an exploratory analysis. We conducted similar time-segment classification with the leave-one-out approach similar than Visconti di Oleggio Castello and colleagues [111], using fMRI movie data from N-1 days to train the classifier and testing it on the day that is left out for each of the days (runs) in the data set. The classification accuracy for each daily scanning session was then compared to other measured external factors using regression models, similarly to the other presented hypotheses H1–H4.

Here, we intensively collected fMRI, daily behavioral, and daily physiological data from a 33-year-old female subject over 133 days (19 weeks). The subject underwent 30 fMRI sessions

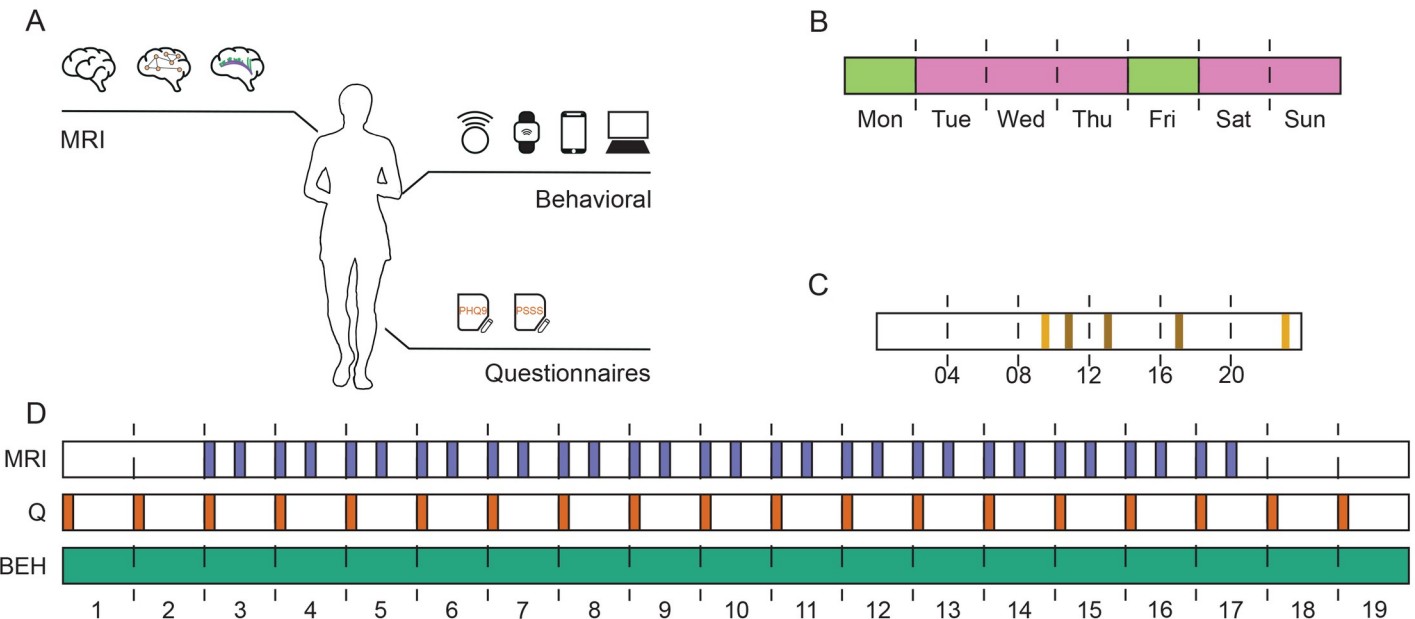

**Fig 1. Experiment overview.** (A) One subject provided frequently sampled data over 19 weeks from 3 data sources. (i) The subject underwent MRI sessions where structural, functional, and diffusion MRI data were collected. (ii) During the 19 weeks, we also employed smartphones and wearables to collect objective data outside the scanner about the subject's behavior. The subject also performed a short PVT and n-back test to assess cognition daily. (iii) The subject also answered questionnaires about her mental state on a regular basis. These multiple data sources allowed us to map data collected in the scanner, under laboratory conditions, with data collected outside the scanner, under quotidian situations. (B) MRI data was collected on Mondays and Fridays, depending on the availability of the scanner. During those days (in green), the participant performed a 10-min PVT in the scanner, while on the other days (pink), the participant performed a 5-min PVT at her office or home. (C) The participant was requested to answer 5 mood questionnaires every day. Two surveys were scheduled on a fixed time (yellow), while the other 3 surveys were scheduled at random between the fixed schedule (brown). (D) Timeline of data collection. All data was collected in 19 weeks. From day 1, behavioral data (green) was continuously collected using smartphones and wearables. Answers to PHQ9 [113] and PSS [114] questionnaires were also saved at the beginning of the experiment and were continuously collected each week (orange). Brain activity data collection began 2 weeks after the start of the experiment and was gathered twice per week (purple). PSS, Perceived Stress Scale; PVT, Psychomotor Vigilance Test.

in which we measured functional brain connectivity while performing 4 different tasks. Such tasks include PVT [42] to assess sustained attention, resting-state, movie-watching as a naturalistic paradigm (Part 2, The Grand Budapest Hotel [44]), and dual 1-back and 2-back tests to assess working memory [43]. To take into account possible confounds in the scanning sessions, concurrent eye-tracking, heart rate, and respiration rate signals were measured in the scanner. During these 19 weeks, we also collected daily objective physiological, behavioral, and lifestyle data employing a smartphone app and 2 wearable sensors that track the subject's signals continuously and passively, that is without the need of the subject's input (S1 Table provides a detailed description of these signals). Wearables and smartphone data were also collected 2 weeks before the MRI sessions began and 2 weeks after the MRI sessions ended. Fig 1 shows a description of the experimental design.

All data collected were pseudonymized and released with this registered report. This experiment presents a reductionist approach as reliability and robustness are embedded in the design. By using data from one subject, we removed many sources of confounds that are common in cross-sectional designs; for example, controlling for anatomical and functional factors, weight, movement, and many more. We hope this data set will be useful to other researchers for investigating within-subject variability in brain function during tasks, alongside natural physiological variations, and for developing new methodologies for the analysis of multivariate brain and sensor data. This data set allowed us to understand how task-evoked brain activity is influenced by daily physiological, behavioral, and lifestyle factors, advancing the

understanding of brain–behavior dynamics. It also allowed us to use fewer artificial stimuli while taking into account more natural situations over different timescales, which are a current challenge to current human behavior research [112]. This setting is not only important to understand how population-level findings are related to results at the individual level, but also to understand the conditions for the phenomena to be reproducible. We hope that the results from this experiment support the design of future precision-mapping studies with clinical populations for personalized medicine.

## Methods

### Ethics statement

This study was approved by the Aalto University Research Ethics Committee (28th of June, 2019, Helsinki, Finland) and was carried out in accordance with the principles expressed in the Declaration of Helsinki. Two updates were required to include new collaborators, adjust the timeline and to include the privacy notice for the study. Both updates were accepted by the Aalto University Research Ethics Committee. Written informed consent was obtained from the participant. As approved by the Committee, the participant did not receive any compensation for this study.

### Design

**Overview.** In this single-subject correlational study, the participant underwent 133 days (19 weeks) of data collection in total. On the first day, the participant signed the consent forms, answered a battery of questionnaires (see Design, Behavioral data) and the behavioral data collection began. To establish a baseline, only behavioral data was collected during the first 2 weeks (for details, please see Design, Behavioral data). The MRI data collection began in the third week. There were 2 MRI sessions per week over 15 weeks, where we collected structural and functional MRI (please see Design, Brain imaging data acquisition for details). The order of the acquisitions and tasks did not change in the MRI sessions. However, during the first and last session of the experiment, diffusion MRI and T2-weighted images were collected. After the 15 weeks of MRI data collection, the subject collected 2 more weeks of behavioral data. An overview of the experiment is shown in Fig 1. The Stage 1 protocol is available in the Open Science Framework https://osf.io/5hu9c/.

**Participant.** The participant (author AMT) is a right-handed female, aged 33 years for the majority of the study. The participant has no history of severe psychiatric disorders or any neurological disorders. To date, she is generally healthy with no history of chronic medical conditions. She is not prescribed special medication, except oral contraceptives (Levonorgestrel and Ethinylestradiol), which she has taken for more than a year. She is active, cycles regularly, and exercises 5 times a week (mean total energy expenditure 1.467, mean daily calories burned 2416.14, based on the pilot data, see S1 Text, pilot data, pilot III). She does not have any dietary restrictions, does not smoke, has never used any recreational drugs, and rarely consumes alcohol. Her native language is Spanish, but she also uses English at a professional level.

**Timeline.** An approximate timeline for the data collection is shown in Fig 1. Behavioral data collection lasted 19 weeks, whereas MRI data collection was completed in 15 weeks. Data analysis was performed after the data collection was completed.

### Behavioral data

**Initial assessment.** On the first day of the experiment, the participant answered a battery of questionnaires, which includes the Big Five Inventory [115–117], Perceived Stress Scale

(PSS) [114], Patient Health Questionnaire (PHQ9) [113], Generalized Anxiety Disorder 7 Item Scale (GAD-7) [118,119], and Pittsburgh Sleep Quality Index (PSQI) [120]. These allowed us to establish baseline metrics.

**Cognitive tasks.** Two cognitive tasks were employed in this study, a PVT and a dual n-back. Cognitive performance from these tasks was measured daily during the 19 weeks. Both tasks were shown using the Presentation software (Neurobehavioral Systems, Albany, California, United States of America) and their code is available (see Methods, Code availability).

PVT: The participant performed a PVT task [42]. In short, the participant was instructed to observe a red rectangular box on the computer screen and to press a button as soon as a yellow counter appears on it. When the button was pressed, the counter stopped and it was subsequently reset. The period between the last response and the new stimulus varied between 2 and 10 s. The participant was instructed to press the button as soon as the yellow counter appeared, so that the reaction time (RT) was as low as possible, but avoiding lapses (i.e., press the button when no stimulus is yet displayed). A schematic of the PVT stimuli is shown in the S1A Fig.

Based on previous results [42], this task has shown good performance in the partial sleep deprivation condition, with large effect sizes (d > 0.8) for the median 1/RT, number of lapses, and performance score for PVTs longer than 5 min. In fact, Basner and colleagues [121] have demonstrated that 3 to 5 min is usually sufficient to measure the effects of sleep on vigilance. Two versions of the PVT were employed, a 10-min version which was used in the scanner, and a 5-min version which was employed every day except for those days where the 10-min PVT was used (see Fig 1B). We used a longer PVT task in the scanner because usually in fMRI, longer versions are preferred to collect more volumes. Nevertheless, using a 5-min and a 10-min PVT task should not pose any problems given the 5-min PVT is a reliable substitute for the 10-min PVT [122].

Dual n-back: The participant performed an adaptive dual n-back task as to the one used by Salo and colleagues [43] for 1 and 2 back tasks in 8 alternating blocks of 20 trials each. At the beginning of each block, instructions showed which n-back task the participant should perform (1-back or 2-back). Then, the participant was presented with synchronous sinewave tones and sinewave gratings with occasional auditory or visual distractors. At the end of each block, general feedback was displayed. All instructions and feedback were one line of text in English. Stimuli and procedures have been previously described in other papers [43,123].

On each trial, auditory tones and visual gratings were presented, and either tone pitch or grating orientation changed compared to the previous trial. On 1/3 of the trial, either visual or auditory distractors concurred with the task-relevant stimuli. Visual distractors were spectrally complex textures and auditory distractors were spectrally complex sounds (e.g., car honks). The participant's task was to indicate with a button press, in which modality and to which direction the stimulus changed with respect to the previous trial (1-back) or to the 2 trials before (2-back). When there was a change in the auditory stimulus, the participant should have pressed "up" if the current pitch was higher, or "down" if the current pitch was lower, compared to the n trial. When the change was in the visual stimuli, the participant should have pressed "right" if the current grating orientation rotated clockwise, or "left" if the current grating orientation rotated counterclockwise, compared to the n-th trial. Only one modality changed per trial. The difficulty of the task was adapted to the participant's responses, i.e., with more accurate answers, the changes became more subtle. This adaptation aimed to keep the rate of correct responses at 70.7% with an adaptive staircase method based on trials with no distractors. A schematic of this process is shown in the S1B Fig.

The stimulus pairs were presented in onset-to-onset intervals of 1,800 ms and their duration was 300 ms. The frequency of the auditory stimuli ranged between 600 and 1,800 Hz in steps according to the participant's task performance in a staircase procedure. The maximum

change in pitch was limited to 0.5 octaves. The auditory distractors were band-pass filtered at 200 and 7,000 Hz and notch-filtered at 1,000 Hz with a two-octave wide filter. The sound maximum intensity was 80 dB SPL.

The grayscale gratings orientation ranged between 0˚ and 360˚ in steps depending on the task performance. Each grating had a spatial frequency of 2 c/deg and was displayed in a Gaussian envelope (diameter 3˚). The grating orientation varied according to the participant's performance in a staircase procedure. The lower part of the grating was kept at the center and the grating rotated. The maximum change is constrained to 90˚. Complex colored textures were used as visual distractors (size 16 × 24˚). For consistency across trials, we cut off a circular area of 6˚ from the center of the textures. The root mean square contrast of the visual distractors is 0.3.

**External factors.** Two wearables and 1 smartphone app were used to obtain objective quantitative data from external factors with minimal subject burden. External factors such as sleep, physical activity, autonomic nervous system activity, and mood were measured. The participant wore the devices at all times, except when charging the devices was required or when it was not allowed (e.g., going to the pool for some devices).

Sleep and physical activity were measured using the Oura ring (Oura Health Ltd, Oulu, Finland). The smartring is a commercially-off-the-shelf device that has been validated against actigraphy for sleep measurements under free-living conditions [63]. The ring has also been used in physical activity studies under normal life conditions [73,124]. It tracks 55 metrics that are daily aggregated. However, many of the metrics are highly correlated with each other and therefore, we only used a subset of them in our study (see S1 Text, experiment data, selection of features from wearables). We used the ring to track daily total sleep time, awake time, restless sleep, steps, and inactive time. Originally, we chose to use daily total sleep time, sleep efficiency, sleep latency, steps, and inactive time measurements based on pilot data (see S1 Text, pilot III, selection of features from smartring). However, during data quality check-ups (i.e., after finishing the data collection and before the initial analysis), we noticed a high correlation between other sleep measurements and sleep latency and sleep efficiency. This led us to reconsider the variables to employ. For a full description, please see S1 Text, experiment data, selection of features from wearables.

ANS activity was measured using a wrist monitor (EmbracePlus, Empatica, Empatica, Boston, Massachusetts, USA)**. The device is a medical-grade wearable that measures physiological data via photoplethysmography (PPG), accelerometry, and electrodermal activity. EmbracePlus is the successor of the E4 wristband, which has been validated for measurement of heart rate and heart rate variability [87,125,126] under different conditions. The E4 wristband was originally registered in the protocol. However, Empatica no longer sold the device when the data collection was to start. Therefore, to ensure the reproducibility of this protocol, we decided to employ the EmbracePlus device. We received the Editorial approval for this change on 18 January 2023. We used the monitor to track respiration rate and heart rate variability.

Mood and everyday experiences were recorded using the Aware Framework application (https://awareframework.com/) and the koota service [127]. The application is freely available for Android and iPhone. It is a tool to collect mobile context information by sensor instrumentation capable of gathering data passively (i.e., no user-input required) and actively (i.e., requires input from the user). We used the app to collect the participant's answers to 3 questionnaires, morning, evening, and mood. Morning questionnaires were asked at 10:00 AM and inquiry about the sleep on the previous night. Evening questionnaires were asked at 11:00 PM and included questions about social interactions, exercise, and alcohol, caffeine, and theine consumption. In addition, mood was monitored using the short version of the international

PANAS [128] that was asked with the morning and evening questionnaires, plus at 3 random times between 10:00 AM and 11:00 PM. In addition, a free-text box is available with the selected PANAS questionnaire in case the participant would like to provide further contextual information. Finally, the participant answered weekly PHQ9 and PSS questionnaires. The complete set of questions is available in the S2 Table.

**Other measurements.**   Aside from the previous measurements, the participant logged monthly weight and day 1 of the menstrual cycle using the free-text box from AWARE.

Additional passive data were collected using the AWARE app. In short, data from battery levels, screen use, calls, messages (SMS), and global positioning system (GPS) were collected. However, these data are not used in the current experiment.

After each fMRI session, the subject completed a questionnaire about the experience in the scanner. Using a 5-point Likert scale, the subject rated the engagement on each task. Additionally, the subject also wrote short freeform accounts on the thoughts while performing rs-fMRI and movie-watching tasks.

**Brain imaging data acquisition.**   MRI was performed in a fixed schedule, Mondays and Fridays at 1,130 hours, subject to the scanner availability. Imaging was performed on a Siemens MAGNETOM Skyra 3T MRI scanner (Siemens Healthcare, Erlangen, Germany) at the Advanced Magnetic Imaging Center, Aalto University, using a 30-channel head coil. At each session, the following data was collected in the same order: localizer, PVT, resting-state, movie watching, N-back, and structural T1-weighted imaging. At the first and last session, structural T2-weighted and diffusion MRI data were also gathered after the structural T1-weighted imaging (see Fig 1), although we did not analyze these in this study.

**Structural MRI.**   Anatomical images were acquired using a T1-weighted MPRAGE pulse sequence (TR 2530 ms, TE 3.3 ms, TI 1100 ms, flip angle 7˚, $256 \times 256$ matrix, 176 sagittal slices, 1 mm [3] resolution, distance factor 50%, 6 min 2 s total acquisition time).

T2-weighted data were collected using a T2-space sequence (TR 3200 ms, TE 412 ms, $256 \times 256$ matrix, 176 sagittal slices, 1 mm [3] resolution, 4 min 46 s total acquisition time).

**Diffusion MRI.**   Diffusion images were acquired using a dMRI sequence with fat suppression (TR 4400 ms, TE 128 ms, $256 \times 256$ matrix, 72 axial slices, 2 mm [3] resolution, 8 min 30 s total acquisition time).

**Functional MRI.**   Full-brain BOLD images were acquired in an interleaved fashion using gradient-echo-planar imaging with fat suppression, multiband acceleration factor 4, TR 594 ms, TE 16 ms, flip angle 50, $64 \times 64$ matrix, 44 axial slices, slice thickness 3 mm, $3 \times 3$ mm in plane resolution, anterior-posterior phase encoding. This sequence was used to record brain data from the subject while performing the PVT, resting-state, movie-watching, and n-back tasks, in that order. We opted to fix the order of scans to make sure that the same condition (PVT, resting state, movie, and n-back) would be comparable between different measurement sessions. Another option would have been to counterbalance the order of scans to account for potential effects changing over the whole session. However, a possible drawback of using counterbalancing could be that the amount of variability brought by the same task sometimes being conducted first and sometimes last (reflecting, for example, the level of arousal during the scanning session) could be greater than the variability of the external factors of interest. In addition, the fixed-order design has been previously used in massive data collection projects such as the Human Connectome Project [129,130]. While counterbalancing could be used with a larger number of scanning sessions by treating it as an external confound, in single-participant studies, fixing the order of scans could increase the likelihood in detecting brain behavior interactions while keeping the number of scanning sessions reasonably low.

All stimuli were back-projected on a semitransparent screen using a data projector (PRO-Pixx MRI/MEG, VPixx Technologies, Saint-Bruno, Canada) and the Presentation software.

Answers to the stimuli from the PVT and n-back were recorded using a 4-button diamond keypad (RESPONSEPixx/MRI, Pixx Technologies, Saint-Bruno, Canada).

**fMRI stimuli.** PVT: The participant performed a 10-min version of the behavioral PVT (see Behavioral data, Cognitive tasks, PVT) with 1 modification to adapt to the scanner conditions. To avoid noise from the scanning start, we introduced a 30-s washout before and after the PVT task. During this washout, the subject saw a centered white cross on a black background. This washout signal was not taken into account in the analysis.

Rest: a 10-min resting-state fMRI was included at each session. The first 30 s of the scanner were discarded before preprocessing to exclude drifting effects. The participant was instructed to let her mind wander while keeping the gaze fixated at a cross in the center of the screen (black background, white cross).

Movie: The participant watched part 2 from "The Grand Budapest Hotel" by Wes Anderson [44]. Similarly to the PVT stimuli, we introduced a 30-s washout before and after the 9.5-min movie stimuli. Likewise, the washout is not included in the analysis.

N-back: The participant performed the same task described in Methods, Behavioral data, Cognitive tasks with no modifications. This task lasts approximately 6.5 min.

**Other data collected during the scanning sessions.** Eye-tracking, breathing rate, and heart pulse signals were collected during fMRI. Eye movements were recorded during all fMRI acquisitions with an EyeLink 1000 eye-tracker (SR Research, Mississauga, Ontario, Canada). The eye-tracker was calibrated once with a five-point calibration prior to each session. Heart pulse and respiration were monitored with the Biopac system (Biopac Systems, Isla Vista, California, USA). Instantaneous values of heart rate and breathing rate were estimated with the Drifter software package [131].

Finally, after each session, a small questionnaire was administered to check the participant's feelings and sleepiness during the session.

**Blinding.** To prevent any conscious or unconscious bias in the experiment, no data analysis was performed during data collection. Instead, only quality control checks were run to ensure that the data is being correctly saved (see Design, Quality control). In addition, to adhere to best practices, initial hypothesis and analysis plans were recorded in the present registered report.

**Quality control.** Due to the blinding, only basic quality control was performed during the data collection. This includes daily checks to ensure that behavioral data were transferred and stored. We only checked the data timestamps for possible missingness, i.e., no data were analyzed. In case some data were missing, we investigated the possible causes (e.g., malfunction of the devices, low battery) and took corrective actions and avoided further data loss.

The quality control also included checking that MRI data were transferred and stored correctly. After the correct storage, the MRI data were preprocessed using fmriprep (see Analysis plan, MRI preprocessing) and framewise displacement (FD) and temporal signal-to-noise ratio (tSNR) were investigated. Similarly to behavioral data, these checkups were only performed to take corrective actions (e.g., remind the participant to stay still in the scanner) and avoid data loss.

We also checked the integrity of the eye-tracking, heart rate, and respiration rate data collected in the scanner to look for possible artifacts. In case more than 20% of the data per session were corrupted (e.g., outliers, NaNs), we investigated the causes and took corrective actions for the next session to avoid data loss.

## Sampling plan

**Sample size.** We collected daily behavioral data for 19 weeks from 1 individual ($N = 1$). Simultaneously, we also collected 30 MRI scans over 15 weeks from the same individual. We

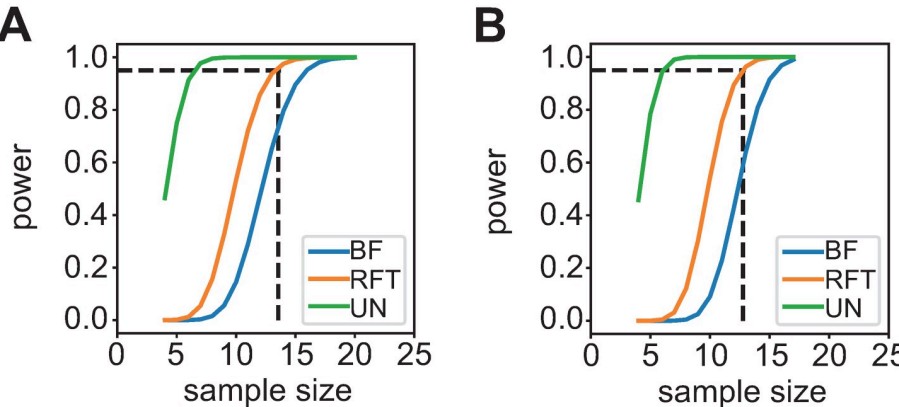

**Fig 2. Power analysis** for (A) PVT and (B) n-back tasks computed by neuropower [132]. Power curves are shown for uncorrected values (green), random field theory (orange), and Bonferroni (blue) corrected values. Fourteen scans are necessary to achieve 95% power using Random Field Theory for both tasks. The curves also show that samples bigger than 20 scans assure enough power for all types of corrections in both tasks. MRI Pilot data available on request (see Data availability). PVT, Psychomotor Vigilance Test.

opted for 30 scans as a compromise between the experiment load to the subject and sufficient number of scans for all tasks according to the power analysis.

**Power analysis.**   Power analysis of pilot data for PVT and n-back tasks revealed that a minimum of 14 samples are required to detect voxel activations given the presented stimuli. These 14 samples yielded 95% power, once values are corrected with random field theory. Power analysis for PVT and n-back tasks were done using second-level general linear models (GLMs) from the pilot data (see S1 Text, pilot III) and computed by neuropower [132]. Results from these analyses are shown in Fig 2 and are openly available (see Code availability).

As assessing power calculations for model-free stimuli (i.e., resting-state and ISC) is more difficult, we based our calculations on proposed sample sizes that have been previously employed. For resting-state fMRI, 30 scans have been used in similar precision mapping studies where fluctuations of functional connectivity across time were investigated [5,28]. For naturalistic stimuli, previous work [133] has established that 30 scans are within the optimal number of samples for the ISC statistics to converge.

**Exclusions.**   Data were only excluded because of low quality. Behavioral data were excluded if there are more than 40% missing values over the whole time series (i.e., 19 weeks). For data measured with wearables and smartphones, we completed missing values with the mean of the data (see Analysis plan, missing data). Structural MRI data were excluded in case part of the brain in the image was missing or if extreme artifacts are found; e.g., movement or ghosting artifacts that are notorious and were not corrected by the preprocessing. rs-fMRI volumes were excluded because of high movement (i.e., volumes whose FD > 0.2). Scrubbed volumes could be different for each session of the rs-fMRI data. For movie-fMRI, we detected the volumes with high movements (FD > 0.2) in each session and censored them across the 30 sessions. Discarding volumes did not affect the sampling size (i.e., the number of scans to be taken).

## Analysis plan

**Missing data.**   Due to the quality controls imposed on the data (see Methods, Quality control), we expected little to no missing data points. However, in case of missingness in the

measurement of external factors (see Methods, Behavioral data), we replaced missing values with the mean of each variable where missingness is found. The imputation method was selected after trials with the pilot data (see S1 Text, Pilot data, pilot III). In case of missingness in the MRI data, we rescheduled the session, schedule permitting; the length of the overall data collection was extended accordingly. If more than 20% of the sessions needed to be rescheduled to a different hour or day, time and day of the scanning session was taken into account in a regressor.

**Behavioral preprocessing.** Most behavioral data required little preprocessing. Data from the initial assessment were preprocessed by calculating the questionnaire scores according to each questionnaire's guidelines. Scores from the cognitive tasks were computed using in-house scripts (see Code availability). The scripts calculate PVT scores according to Basner and Dinges [42] based on the reaction times provided by the Presentation software PVT script. The scripts also calculate the median reaction time, and the number of correct, wrong, and missing answers for the n-back test based on the response and reaction time provided by the Presentation software n-back script.

Data from the smart ring and wristband were preprocessed automatically by the web services. For the smart ring, the data were automatically aggregated by day and did not require any further preprocessing. For the wristband, heart rate variability, and respiration rate are automatically computed by the platform and only needs to be aggregated daily. However, due to the change of device, we no longer need to compute the heart rate variability from the inter-beat interval, as stated in the Stage 1 manuscript. Change accepted by the Editor on 18 January 2023.

Data from mood and everyday experiences were preprocessed in 3 different ways. Firstly, negative affect and positive affect scores were computed from the selected PANAS questions. Negative affect (NAF) is the sum of the answers from the afraid, nervous, upset, hostile, and ashamed questions. Positive affect (PAF) is the sum of the answers from the active, determined, attentive, inspired, and alert questions. Then, daily median NAF and PAF scores were computed. Secondly, menstruation days were recorded via free-text in the smartphone app. Finally, free-text data were not preprocessed, but used as a possible guideline into the subject's life events.

Data from smartphones measuring battery life and screen were aggregated by calculating the mean per hour. Calls and SMS data were aggregated by day. GPS location data were pseudonymized by computing daily traveled distance and daily travel time.

**MRI preprocessing.** MRI data were preprocessed using fMRIPrep v.22.1.0 [134,135], which is based on Nipype 1.6.1 [136,137]; (RRID:SCR_002502). fMRIPrep is an analysis-agnostic tool that automatically adapts the workflow to the data set without manual intervention.

**Anatomical data preprocessing.** The T1-weighted (T1w) images were corrected for intensity non-uniformity (INU) with N4BiasFieldCorrection [138], distributed with ANTs 2.3.3 [139] (RRID:SCR_004757), and used as T1w-reference throughout the workflow. The T1w-reference was then skull-stripped with a Nipype implementation of the antsBrainExtraction.sh workflow (from ANTs), using OASIS30ANTs as target template. Brain tissue segmentation of cerebrospinal fluid (CSF), white matter (WM), and gray matter (GM) was performed on the brain-extracted T1w using fast FSL [140] (FSL 5.0.9, RRID:SCR_002823). Volume-based spatial normalization to 2 standard spaces (MNI152NLin6Asym, MNI152NLin2009cAsym) was performed through nonlinear registration with antsRegistration (ANTs 2.3.3), using brain-extracted versions of both T1w reference and the T1w template. The following templates were selected for spatial normalization: FSL's MNI ICBM 152 nonlinear sixth Generation Asymmetric Average Brain Stereotaxic Registration Model [141] [RRID:SCR_002823;

TemplateFlow ID: MNI152NLin6Asym], ICBM 152 Nonlinear Asymmetrical template version 2009c [142] [RRID:SCR_008796; TemplateFlow ID: MNI152NLin2009cAsym].

**Functional data preprocessing.** For each of the BOLD runs found per subject (across all tasks and sessions), the following preprocessing was performed. First, a reference volume and its skull-stripped version were generated using a custom methodology of fMRIPrep. Susceptibility distortion correction (SDC) was omitted. The BOLD reference was then co-registered to the T1w reference using flirt (FSL 5.0.9 [143]) with the boundary-based registration [144] cost-function. Co-registration was configured with 9 degrees of freedom to account for distortions remaining in the BOLD reference. Head-motion parameters with respect to the BOLD reference (transformation matrices and 6 corresponding rotation and translation parameters) were estimated before any spatiotemporal filtering using mcflirt [145] (FSL 5.0.9). BOLD runs were slice-time corrected using 3dTshift from AFNI 20160207 [146] (RRID:SCR_005927). The BOLD time series (including slice-timing correction when applied) was resampled onto their original, native space by applying the transforms to correct for head-motion. These resampled BOLD time series were referred to as preprocessed BOLD in original space or just preprocessed BOLD. The BOLD time series was resampled into standard space, generating a preprocessed BOLD run in MNI152NLin6Asym space. First, a reference volume and its skull-stripped version were generated using a custom methodology of fMRIPrep. Several confounding time series were calculated based on the preprocessed BOLD: FD, DVARS, and 3 region-wise global signals. FD was computed using 2 formulations following Power (absolute sum of relative motions [147]) and Jenkinson (relative root mean square displacement between affines [145]). FD and DVARS were calculated for each functional run, both using their implementations in Nipype (following the definitions by Power and colleagues [147]). The 3 global signals were extracted within the CSF, the WM, and the whole-brain masks. Additionally, a set of physiological regressors were extracted to allow for component-based noise correction (CompCor [148]). Principal components were estimated after high-pass filtering the preprocessed BOLD time series (using a discrete cosine filter with 128 s cut-off) for the 2 CompCor variants: temporal (tCompCor) and anatomical (aCompCor). tCompCor components were then calculated from the top 2% variable voxels within the brain mask. For aCompCor, 3 probabilistic masks (CSF, WM, and combined CSF+WM) were generated in anatomical space. The implementation differs from that of Behzadi and colleagues [148] in that instead of eroding the masks by 2 pixels on BOLD space, the aCompCor masks were subtracted from a mask of pixels that likely contain a volume fraction of GM. This mask was obtained by thresholding the corresponding partial volume map at 0.05, and it ensures components are not extracted from voxels containing a minimal fraction of GM. Finally, these masks were resampled into BOLD space and binarized by thresholding at 0.99 (as in the original implementation). The head-motion estimates calculated in the correction step were also placed within the corresponding confounds file. The confound time series derived from head motion estimates and global signals were expanded with the inclusion of temporal derivatives and quadratic terms for each [149]. Frames that exceeded a threshold of 0.5 mm FD or 1.5 standardized DVARS were annotated as motion outliers. All resamplings can be performed with a single interpolation step by composing all the pertinent transformations (i.e., head-motion transform matrices, susceptibility distortion correction when available, and co-registrations to anatomical and output spaces). Gridded (volumetric) resamplings were performed using antsApplyTransforms (ANTs), configured with Lanczos interpolation to minimize the smoothing effects of other kernels [150]. Non-gridded (surface) resamplings were performed using mri_vol2surf (FreeSurfer).

After the fMRIPrep preprocessing, we applied a 240-s-long Savitzky–Golay filter to remove scanner drift (similar to Çukur and colleagues [151]). To control for motion and physiological

artifacts, we regressed from the BOLD time series 24 motion-related regressors, 16 signals from the WM and CSF (the signals, their derivatives, and their powers), heart and respiration rate as preprocessed by the Drifter software package [131], and motion outliers as detected by fmriprep. All confounds were also filtered using a 240-s-long Savitzky–Golay filter before regressing out their effect to avoid re-introducing artifacts [152]. Finally, the cleaned BOLD signal was filtered with a high-pass filter at 0.01 Hz cut-off frequency. No spatial smoothing was applied for the connectivity analysis, as previous research has shown that spatial smoothing affects connectivity measurements in non-uniform and systematic ways [153,154]. However, for task-related activation analysis with voxelwise univariate general linear model (see Additional analysis), we applied a 6 mm full width at half maximum (FWHM) kernel to smooth the data spatially using the FSL software [155–157].

**Functional connectivity estimation.** Functional network nodes were defined based on the sets of brain regions defined by Seitzman and colleagues [158]. In their work, Seitzman and colleagues [158] generated new subcortex and cerebellum regions, which they later combined with previous parcellations from Power and colleagues [159] (set 1) and Gordon and colleagues [160] (set2). Set 1 comprises 300 regions of interest (ROI) and set 2 comprises 394 ROIs. We will employ set 1 to derive the main results in the manuscript. Results using set 2 are reported in S1 Text.

For rs-fMRI, functional links between nodes were defined as the Pearson correlation coefficients between the averaged time series of the voxels belonging to an ROI. This should yield a weighted adjacency matrix of size $300 \times 300$.

For task-fMRI (i.e., PVT and n-back), individual adjacency matrices were computed using the beta series analysis [161]. This multivariate method employs trial-to-trial variability to characterize dynamic inter-regional interactions. In short, we constructed a GLM in which every stage of every trial is modeled with a separate covariate and obtained trial-to-trial parameter estimates (beta series) for each voxel. Then, we computed the averaged beta series of the voxels belonging to an ROI and correlated these averaged beta series between ROIs. This should yield a weighted adjacency matrix of size $300 \times 300$.

Individual adjacency matrices were Fisher-transformed to stabilize the variance for all correlation values. Average head motion across sessions (as measured by mean FD of a session) were regressed out for each link in the network [28,162]. Finally, we applied the Fisher inverse transformation to the regressed adjacency matrices.

**Subnetwork measures.** We computed 2 network measures, participation coefficient and global efficiency using the Brain Connectivity Toolbox [41]. We opted for these 2 graph metrics as they have been previously used to analyze a precision functional mapping data set [8], and they include both between and within-network measurements. The participation coefficient quantifies the relation between the number of links connecting a node outside its community and the total number of links for that particular node (i.e., between-network measurement). To estimate the participation coefficient, we used the $300 \times 300$ connectivity matrices and a vector of network IDs defined by Seitzman and colleagues [158]. Summary participation coefficients were obtained by computing the mean of the participation coefficients of the network nodes. Global efficiency quantifies the ease of information transfer within a network and it is defined as the average inverse shortest path length in a network [163]. To calculate the global efficiency, the $300 \times 300$ adjacency matrices were subdivided into smaller network-specific matrices; this means that the global efficiency was only calculated among within-network nodes. Summary global efficiency coefficients were obtained by computing the mean of the global efficiency coefficients of the network nodes. Only the measurements for the networks of interest were taken into account, according to each hypothesis (see Table 1).

Both the participation coefficient and global efficiency estimates are computed on thresholded, binarized networks. Given the lack of consensus over a desirable value, we run all our models using 3 widely adopted values of proportional thresholding (10%, 20%, and 30%). These values are within the range known to produce the most consistent results [164]. To ensure a fully connected network, we first compute the maximum spanning tree of the correlation matrix, rank the links from strongest to weakest, and then add the strongest positive links to the network until reaching the chosen proportional threshold, similar to Kujala and colleagues [165]. Here, we reported results using the threshold 10%; results for other thresholds are reported in the S1 Text.

**Inter-daily representational similarity analysis.** Here, we adapted the intersubject representational similarity analysis (IS-RSA) framework [88] to a single subject. In this case, the intuition is that days in which the neural responses are more similar should also be more similar in behavior. To compute the inter-daily representational similarity analysis (ID-RSA), we constructed 2 pairwise (i.e., day-by-day) similarity matrices for the brain and the behavioral data. Then, we assessed the significance of the comparison between these matrices via the Mantel test. The brain similarity matrix was computed using the ISC framework between ROIs defined by Seitzman and colleagues [158]. Similarly to ID-RSA, we adapted the ISC framework to a single subject by treating daily data as one subject in the ISC framework. The behavior similarity matrix was computed using 2 models, the nearest neighbors (NN), and the Anna Karenina (AK) structure [88] based on the behavioral value extracted from the daily answers to the selected PANAS questionnaire, daily total sleep time, awake time, restless sleep, breathing rate, and heart rate variability.

**Regression analysis.** To assess time-synchronous variation in functional connectivity associated with external factors, we performed a standardized regression analysis between: (1) links from the FPN, DMN, somatomotor, and cingulo-opercular networks (CONs) and total sleep time, awake time, and restless sleep for H1; (2) links from the DMN, FPN, and somatomotor areas and total sleep time, awake time, and restless sleep, steps, and inactive time for H2; and (3) links from the DMN, FPN, and CONs and total sleep time, awake time, restless sleep, heart rate variability, respiration rate, and mood computed from the selected PANAS questionnaire for H3 (see Table 1). For each model, we computed empirical null distributions of test statistics via 10,000 iterations of nonparametric permutation testing under the null hypothesis of no temporal association between connectivity and external factors. We used the lmPerm R package.

Similarly, we also performed a standardized regression analysis between the subnetwork measurements and the external factors selected for each hypothesis (see Table 1). For each model, we performed a nonparametric permutation test using 10,000 iterations under the null hypothesis of no temporal association between subnetwork measurements and external factors. We used the lmPerm R package.

We did not include performance as a confound regressor in the PVT and n-back tasks models because our main focus is the variation of task performance as reflected in functional brain connectivity. Nevertheless, to verify the relationship between task performance and behavioral factors, we run 2 standardized regression analysis between: (1) mean 1/RT [42] and total sleep time, awake time, and restless sleep for H1; and (2) accuracy and total sleep time, awake time, and restless sleep, steps, and inactive time for H2. Results from these analyses are reported in the Additional analysis section.

Regarding H3, we included the percentage of prolonged eyes closure as a covariate, i.e., the percentage of time when the eyes have been closed for longer than 10 s (median brief microsleep episodes) [166].

**Cross-correlation analysis.** To understand what timescales behavioral, physiological, and lifestyle factors have effects on functional brain connectivity, we run a series of time-lagged cross-correlation analyses. This analysis helped us identify the relationship between functional brain connectivity estimates (participation coefficient and global efficiency) and past lags of the external factors. For each ROI, we computed the lagged cross-correlation between pairs of functional connectivity estimates and external factors, according to the hypothesis being tested. This means that, e.g., for H5, we computed the cross-correlation between total sleep duration and global efficiency for the DMN ROIs. Lagged-cross correlation coefficients were organized in carpet plots where the x-axis represented lags (in days) and y-axis represented behavioral factors. This carpet plot was organized by network, so following the example, this visualization yielded a global efficiency carpet plot for all behavioral factors and the FPN for H5 across 15 days.

Statistical significance was evaluated by estimating a null model with 10,000 nonparametric permutations between functional connectivity estimates and surrogate behavioral data. For each permutation, surrogate behavioral data were synthesized using spectral synthesis, by shuffling the phase of the Fourier transform of each behavioral predictor and applying the inverse Fourier transform. The surrogate behavioral data were then correlated with the subnetwork's measure (average participation coefficient or average global efficiency) for the estimation of the null model. P-values were then obtained from the null-model probability density function and corrected for multiple comparisons with the same approach used in other analysis (false discovery rate (FDR)).

**Multiple comparisons correction.** For each analysis of each hypothesis, we controlled the FDR by using the approach introduced by Benjamini and Hochberg [167].

After conducting the permutation tests, we applied corrections for multiple comparisons. For the network estimates (global efficiency and participation coefficient) in H1, H2, and H3, we corrected for the number of networks for each variable in the model, namely 4 for H1 and 3 for H2 and H3. For the links in H1, H2, and H3, we corrected for the number of links of the chosen subnetworks, for each variable in the model.

After conducting permutation models with synthetic data for each subnetwork in H5, H6, and H7, we adjusted for the number of subnetworks, assuming independence for each variable-lag pair. This means we treated variables on consecutive days as independent; for example, variable v at lag l1 is independent from a variable v at lag l2 and variables are independent. However, it is important to note that this assumption may not hold true for all variables, as autocorrelation levels and covariance can vary. For example, sleep patterns may exhibit higher autocorrelation compared to mood patterns or sleep patterns may be correlated to activity patterns. Therefore, while corrections were made for the number of networks in H5 (4), H6 (3), and H7 (3), there are still 15 lags to consider. We have provided all statistical values in the GIT (see Code availability) for transparency. Finally, multiple comparisons corrections for H4 and H8 were performed using FSL's Randomise tool.

## Between-days movie time-segment classification

Following the work of Visconti di Oleggio Catello and colleagues [111], we used the same leave-one-out classifier on our naturalistic movie-watching data. The approach is described in detail in the zenodo release [168]. Briefly, the movie fMRI time series were divided into overlapping segments of 14.85 s (25 TRs, in the original paper segments were 15 s long). The segments were obtained with a sliding window of 25 TRs, with sliding time equal to 1 TR, i.e., each segment has 24 time points overlapping with the previous segment for a total of 946 segments. Then, the classifier is implemented by calculating the pairwise similarity (correlation-

based distance) between the n-th segment from the left-out movie fMRI session and the average signal from the n-th segment from the other sessions. The pairwise similarity was computed over a searchlight sphere of 10 mm radius and the segment with the highest similarity was chosen as the predicted one (chance level equals 1/946). The accuracy of the classifier was then estimated as the percentage of correctly classified segments across the whole run that is left out. The procedure was repeated for all searchlight spheres covering the whole gray matter brain areas. For each day, a brain map with values of classification accuracy was obtained. These daily maps were then correlated with external factors as described in the previous sections using linear regression and correction for multiple comparisons with FSL randomise over 10,000 permutations.

The proposed classification analysis in this manuscript differs from the original one in the following: Firstly, in the original study the TR was 1 s, i.e., the temporal distance between consecutive time segments is 1 s which is almost twice as big than the distance in this study (TR = 0.594). Because of this, we explored a few temporal segmentation approaches by shifting the sliding window with 1TR, 2TRs, 4TRs. Secondly, contrary to the original study, we did not perform hyperalignment as the subject is the same across all sessions. Thirdly, in the original study, only the cortical surface was considered after transforming the volumetric data with freesurfer. Here, we kept the data in volumetric format as it has 2 advantages: (i) we could also explore subcortical areas; and (ii) we could better control for false positives using the threshold-free cluster enhancement correction as implemented in FSL randomize since the classification accuracy for each day was correlated with other behavioral and physiological time series obtained outside the scanner. Finally, although still widely used in the neuroimaging literature, leave-one-out approaches have limitations due to potential overfitting and outliers effect [169]. For this reason, we also explored other forms of cross-validation with repeated random splits by leaving out 20% of the sessions.

## Additional analysis

The richness of the collected data allowed several questions to be explored in different ways. Despite our intention to mainly focus on connectivity, we acknowledge some interesting analyses worth exploring. Many of these additional analyses are checkpoints to further what could be driving our results or serve as data checks. Because of their nature, these analyses are therefore reported in the Additional analysis section.

Although connectivity analysis can be applied to all tasks, more traditional methods such as standard mass univariate statistics can be used to analyze the PVT and n-back tasks. Consequently, we run a supplementary analysis for H1 and H2. In this analysis, we computed the activation maps for the PVT and n-back tasks using nilearn [170]. Task-related activation was determined by identifying voxels showing a significant difference in BOLD signal, taking into account the average hemodynamic response. Event-related design matrices were generated for the PVT tasks, where we took into account the RT. Block-related design matrices were built for the n-back task. Results from the first level analysis were employed to regress the effects of behavioral and lifestyle factors according to each hypothesis. Mean FD was included as a confound. Statistical validity was assessed via FSL randomise over 10,000 permutations.

We chose an adaptive n-back test design where the perceptual load varies at each session. By balancing the perceptual load across sessions, we wanted to make sure that the task remains challenging over time and that the participant's effort remains maximal and at a stable level across both, between and within sessions. To assure our results reflected the working memory load, we examined the influence of perceptual load and attention in the n-back task. To this end, we conducted a separate analysis, where the threshold value obtained with the adaptive n-

back task was included as a confound (see Salmela and colleagues [43]). Like the PVT, we expected these results to reflect the level of attention, instead of the working memory.

### Network visualization

Plots were generated using netplotbrain [171], nilearn [170], matplotlib [172], and seaborn [173].

### Pilot data

To demonstrate the feasibility of our design, we ran 3 separate pilot studies. The first 2 pilot studies aimed to check the data quality of the sensors we were to use, and to check the possible learning effects for the PVT and n-back tasks. The third pilot study aimed to test the tasks in the scanner, the tolerance of the subject to the protocol, the MRI preprocessing strategies, and to check the sensor data quality again.

Results from the first 2 pilot studies demonstrated reliable datastreams from wearables and smartphone sensors. For the PVT task, data suggest there are no learning effects. For the n-back task, data suggest there are some learning effects that span no longer than 10 days.

Results from the third pilot study suggest successful adaptation of the tasks in the MRI scanner, good tolerance of the subject to the protocol (as evidenced by FD < 0.2 for 99.2% of the time for all tasks), and reliable datastreams from wearables and smartphones. Thanks to the pilot data, we could select the preprocessing strategies stated in the analysis plan. The pilot data were also used to compute the power analysis (see Sampling Plan, Sampling size).

Full description of the pilot studies and their results are available in S1 Text.

## Results

### Data validation

Only 1 MRI session was rescheduled due to technical problems (session no. 9). Because of holidays, sessions 24 and 25 were taken on a Thursday and a Tuesday, respectively. Together, these sessions account for 10% of our sample and therefore, in line with our criteria (see Analysis plan, Missing data), we did not include the time and day of the scanning session as regressors in our models. Overall, the MRI data quality was good, with mean FD values below 0.17 for all sessions. BIOPAC and eye-tracker data were successfully collected in most of the sessions, with a few exceptions: (i) in the last session, the BIOPAC data for the movie-watching task was corrupted; (ii) in sessions no. 9, 11, 21, and 26, the eye-tracker data failed for the movie-watching and n-back tasks; and (iii) in sessions no. 3 and 9, the eye-tracker data was corrupted for the resting state acquisition. Due to these data issues, we did not include the heart and respiration rate as regressors in the movie-watching MRI preprocessing for the last session, and we could not estimate the percentage of microsleeps for 2 rs-fMRI sessions.

No data was excluded from the external factors. However, some data sources exhibited lower quality than others. Notably, the daily percentage of data collection for heart rate variability and breathing rate was under 50%. The low performance can be attributed to the challenges of deriving heart rate variability and respiration rate from PPG, which most of the time requires low light exposure and low movement [87]. This means that heart rate variability and respiration rate signals are reliable mostly during sleep and they should be interpreted as such for the rest of the paper. Conversely, the daily percentage of successfully collected data for other data sources exceeded 60% for most of the days, allowing us to compute representative daily averages. The GPS showed the largest data loss with 18% of the days having missing points.

For detailed information about the data quality and its technical validation, please refer to the Data quality: Main data set section in the S1 Text.

Finally, to ensure that there is no multicollinearity between the different external factors in the linear regression models, we computed their variance inflation factor (VIF) and their correlation coefficient. S19 Fig shows the covariance matrix for all external factors. We discarded variables whose VIF values were above 5. Moreover, if 2 variables were highly correlated ($\rho >$ 0.7), only one of these variables was used. For the full list of the chosen variables and additional information, see the Regressors section in the S1 Text.

The results presented in the subsequent sections have been corrected for multiple comparisons. Detailed statistics and $p$-values for these analyses are provided in the S1 Text. Additionally, all statistical results are available in the GitHub repository [176] at https://zenodo.org/doi/10.5281/zenodo.13208811.

## Q1: Behavioral, physiological, and lifestyle factors experienced by the individual on the previous day affect today's functional brain connectivity patterns

### H1: Fluctuations in sleep patterns are correlated with functional connectivity within the default mode, somatomotor, and cingulo-opercular networks during sustained attention tasks

We run a regression analysis of the effect of the previous day's sleep duration, awake time in bed, and restlessness on unthresholded link-weights for the FPN, DMN, CON, and somatomotor network. The networks were computed from functional connectivity data obtained during a PVT task. These analyses revealed that restlessness is the only significant predictor in the model, showing a notable impact on 5 links (Fig 3), with β coefficients exhibiting a range of absolute values between 0.66 and 0.86 ($p < 0.01$) (for a list of all link statistics, see S4 Table). Most of the links had a negative slope, indicating that overall, an increase in restlessness is associated with a decrease in link-weight. In other words, more interruptions during sleep are associated with a decrease in connectivity for the majority of links.

The majority of links were associated with DMN, while only few nodes belonged to the CON and somatomotor areas. This pattern persisted when the global signal was regressed from the fMRI data (S18 Fig and S6 Table), but not when the data is analyzed with an alternative parcellation (S17 Fig and S5 Table). In this latter case, more CON and somatomotor nodes were identified. Notably, few links were related to sleep restlessness independent of the global signal removal. These links are between the right inferior frontal gyrus pars orbitalis and the right dorsolateral superior frontal gyrus (β = −0.86, $p < 0.01$), the right inferior frontal gyrus pars orbitalis and the right angular gyrus (β = −0.66, $p < 0.01$), and the left middle occipital gyrus and the cerebellum (β = 0.8, $p < 0.01$).

Next, we computed the global efficiency and participation coefficient over thresholded networks. We found a significant negative association between the somatomotor global efficiency and restlessness when removing the fMRI global signal, with β values ranging from −0.56 to −0.61 ($p < 0.01$) (S18 Fig). This result was consistent across all proportional thresholds (S7 Table). Analyses for the between and within network estimates revealed no significant results for any networks.

### H2: Fluctuations in sleep and physical activity patterns are correlated with functional connectivity within the fronto-parietal, default mode, and somatomotor networks during working-memory tasks

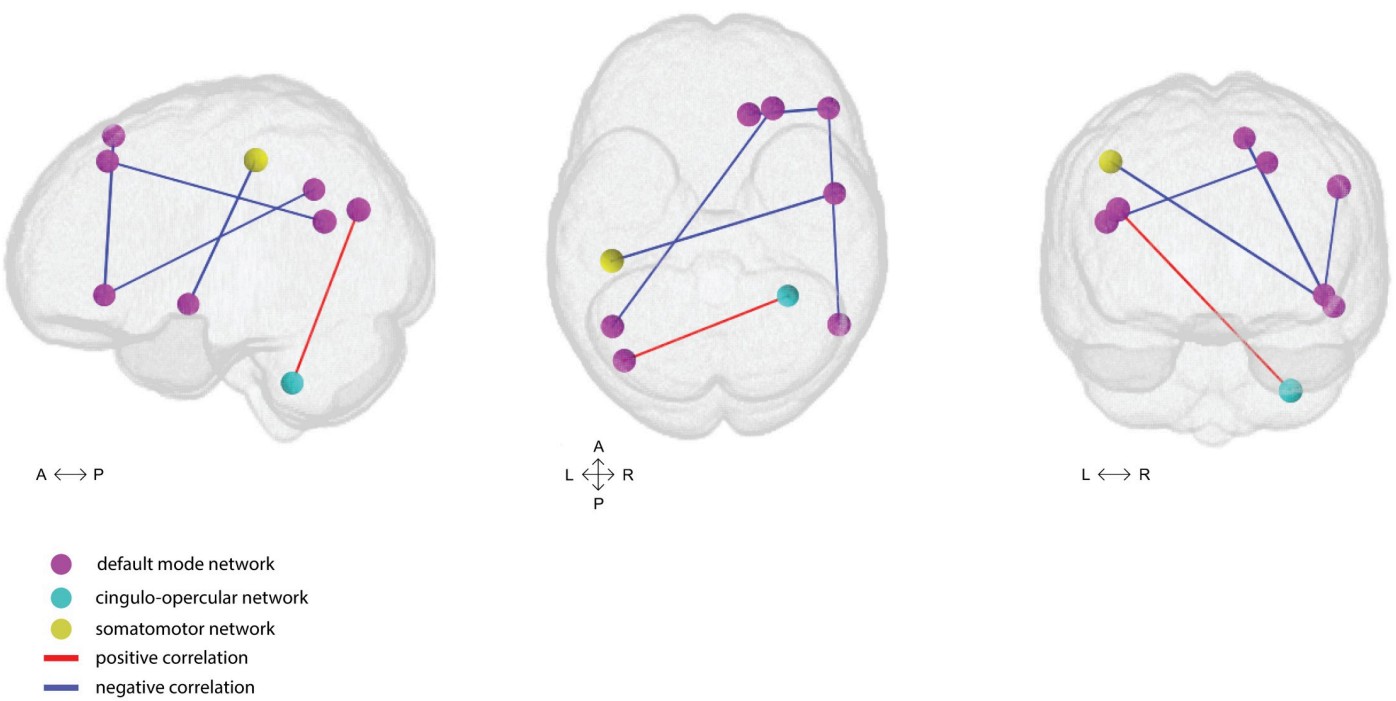

**Fig 3. Functional connectivity during sustained attention tasks is linearly dependent on the quality of sleep from the previous day.** Specifically, restless sleep is associated with connectivity among nodes within the DMN (purple), cingulo-opercular (cyan), and somatomotor (yellow) networks. Red colors indicate positive correlations and blue colors indicate negative correlations. Results are empirically thresholded via 10,000 iterations of nonparametric permutation testing and further corrected for multiple comparisons (corrected $p < 0.01$). Plots were generated with netplotbrain [171]. Unprocessed study data can be found in the Zenodo data set release [175]. Processed results derived from the study data are accessible in the GIT repository [176], under the results folder. DMN, default mode network.

We conducted regression analyses to investigate the effects of the previous day's sleep duration, awake time, restless sleep, number of steps, and inactive time on the unthresholded link-weights in the FPN, DMN, and somatomotor networks. These link-weights were derived from functional connectivity recorded during working-memory tasks. Regression results reveal that restless sleep and inactive time have a statistically significant relationship with link-weights in the FPN, DMN, and somatomotor network (Fig 4). All links exhibited robust relationships, with β coefficients falling within a range of absolute values from 0.67 to 0.89 ($p < 0.01$) (for a list of all link statistics, see S8 Table). We observed mixed-effects of inactive time and the working memory connectivity over time, with an equal number of links displaying negative and positive slopes (Fig 4A). Conversely, a significant number of links associated with restlessness showed negative slopes (Fig 4B), indicating that overall, more interruptions during sleep are related to decreased connectivity in nodes within the FPN, DMN, and somatomotor networks.

Re-analysis of the data with a different parcellation (S19 Fig and S9 Table) and global signal regression (S20 Fig and S10 Table) yielded similar negative connectivity patterns for restlessness. It also highlighted more negative links related to inactivity. Thus, supplementary analyses appear to support that less activity during the previous day is associated with lower connectivity. Notably, 6 links consistently showed a significant relationship with external factors, independent of the global signal removal, with β absolute values ranging from 0.7 to 0.88 ($p < 0.01$). Five of these links were associated with sleep restlessness and only 1 link was related to inactivity (S8 Table). Among these, we found links connecting the left anterior cingulate with the left superior (β = 0.88, $p < 0.01$) and middle frontal gyrus (β = 0.78, $p < 0.01$), the left

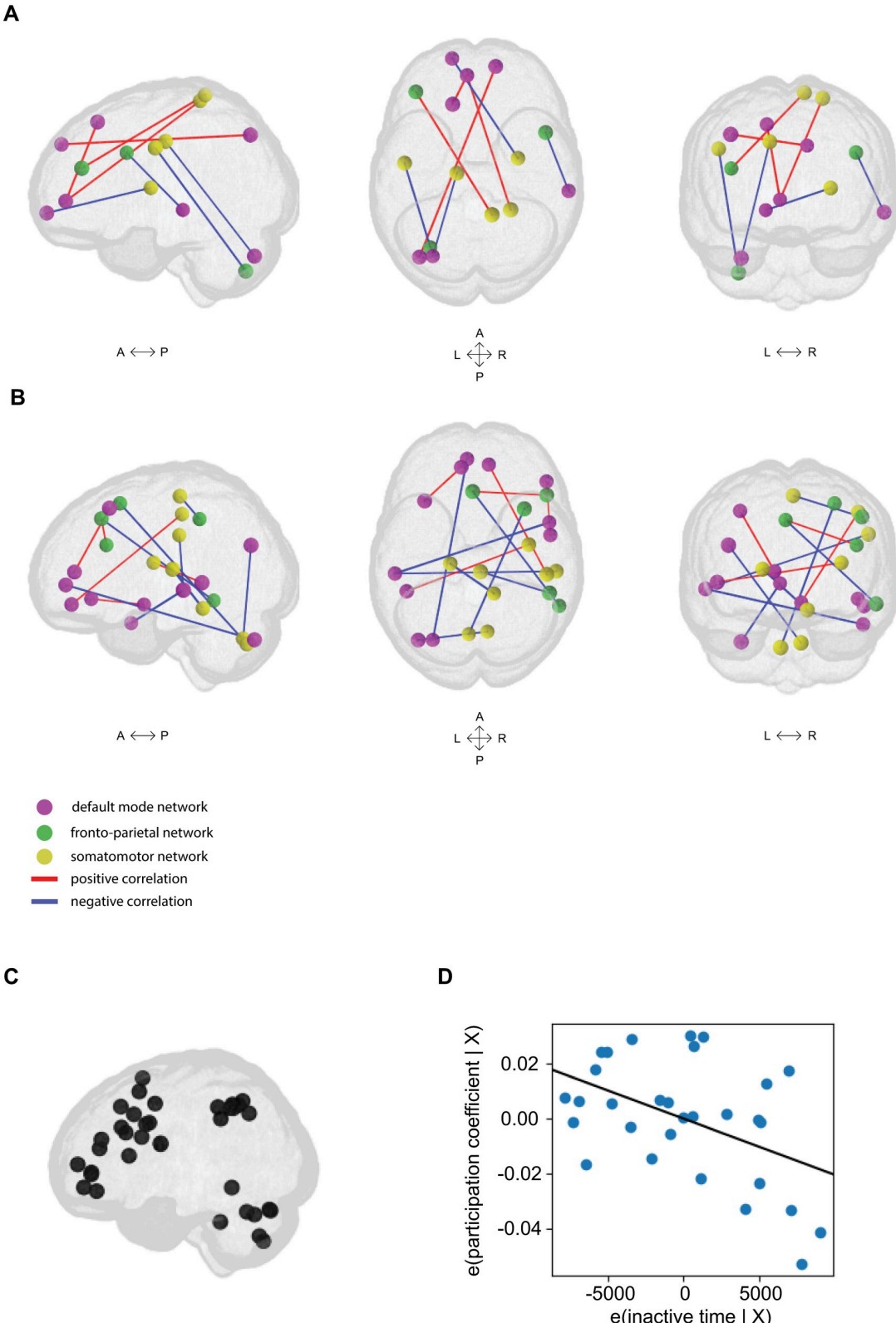

**Fig 4. Functional connectivity during working memory tasks linearly depends on both prior night's sleep quality and previous day's inactivity time.** (A) Linear regression models on individual links showed significant associations between the previous day's inactive time and links in the DMN (purple), fronto-parietal (green), and somatomotor (yellow) networks. (B) Similarly, analyses revealed significant relationships between the prior night's restless sleep and these same networks. Red colors indicate positive correlations and blue colors indicate negative correlations. (C) Nodes from the FPN employed to compute the participation coefficient. (D) Partial regression plot showing that the FPN's participation coefficient is proportionally related to the previous day's inactive time. Results are empirically thresholded via 10,000 iterations of nonparametric permutation testing and further corrected for multiple comparisons (corrected $p < 0.05$). Brain plots were generated with netplotbrain [171]. Unprocessed study data can be found in the Zenodo data set release [175]. Processed results derived from the study data are accessible in the GIT repository [176], under the results folder. DMN, default mode network; FPN, fronto-parietal network.

thalamus and the cerebellum (β = −0.88, $p < 0.01$), and the inferior parietal gyrus and the rectus gyri (β = 0.7, $p < 0.01$).

Further, regression analysis from the network-average participation coefficient identified the previous day's inactive time as the key factor influencing changes in the between-network integration of the FPN (β = −0.57, $p < 0.05$) (Fig 4C and 4D). This result suggests that less activity in the previous day is associated with less FPN connectivity with other networks. The finding was consistent upon re-analysis with an alternative parcellation (S19 Fig and S11 Table), but it did not hold when the global signal was removed. No significant results were found for global efficiency, independent of the parcellation or global signal removal.

### H3: Fluctuations in patterns of sleep and autonomic nervous system activity are correlated with functional connectivity within the fronto-parietal, default mode, and cingulo-opercular networks during resting-state fMRI

To understand the effect of the previous day's sleep, mood, and ANS activity patterns on resting-state functional connectivity, we ran a series of regression models on the unthresholded link-weights within the DMN, FPN, and CON. Results from these models revealed that the previous day's awake time in bed was negatively related to the connectivity between the left dorsolateral superior frontal gyrus and the right thalamus (β = 1, $p < 0.01$) (Fig 5A). Results also indicated that the previous day's maximum HRV was associated with the connectivity between the left insula and the left middle temporal gyrus (β = 0.95, $p < 0.01$), and the connectivity between the right posterior cingulate gyrus and the right cerebellum 8 (β = −0.73, $p < 0.01$) (Fig 5C). While all models for rs-connectivity were controlled for the percentage of microsleeps in the scanner, we also report a few links that were particularly affected by these microsleep episodes (Fig 5B and S12 Table).

Supplementary analyses, including a second parcellation and global signal regression, revealed that the prior's day maximum HRV predicted link-weights in the DMN, FPN, and CON, albeit being different from the ones yielded in the main analysis (S21 and S22 Figs and S13 and S14 Tables). Awake time in bed also emerged as a significant predictor of brain connectivity in the 3 networks when we analyzed the data with a second parcellation (S21 Fig and S13 Table), but not when the global signal was regressed. Despite the links found in each analysis being different, the emergent pattern remains: links associated with awake time have positive slopes (β ranges between 0.88 and 1.33, $p < 0.01$) and links related to HRV have mixed effects (β absolute values ranges between 0.73 and 1, $p < 0.01$). In other words, spending more time in bed without sleeping increased the connectivity in selected links. In addition, lower maximum HRV enhanced the connectivity in some links, while decreasing it on others. We also found links related to the percentage of eyes closure in the scanner for all parcellations and global signal regression analyses (β absolute values ranges between 0.68 and 1, $p < 0.01$).

Finally, regression results from the within- and between-networks estimates showed that the previous day's minimum HRV was the only significant predictor in the model, showing a

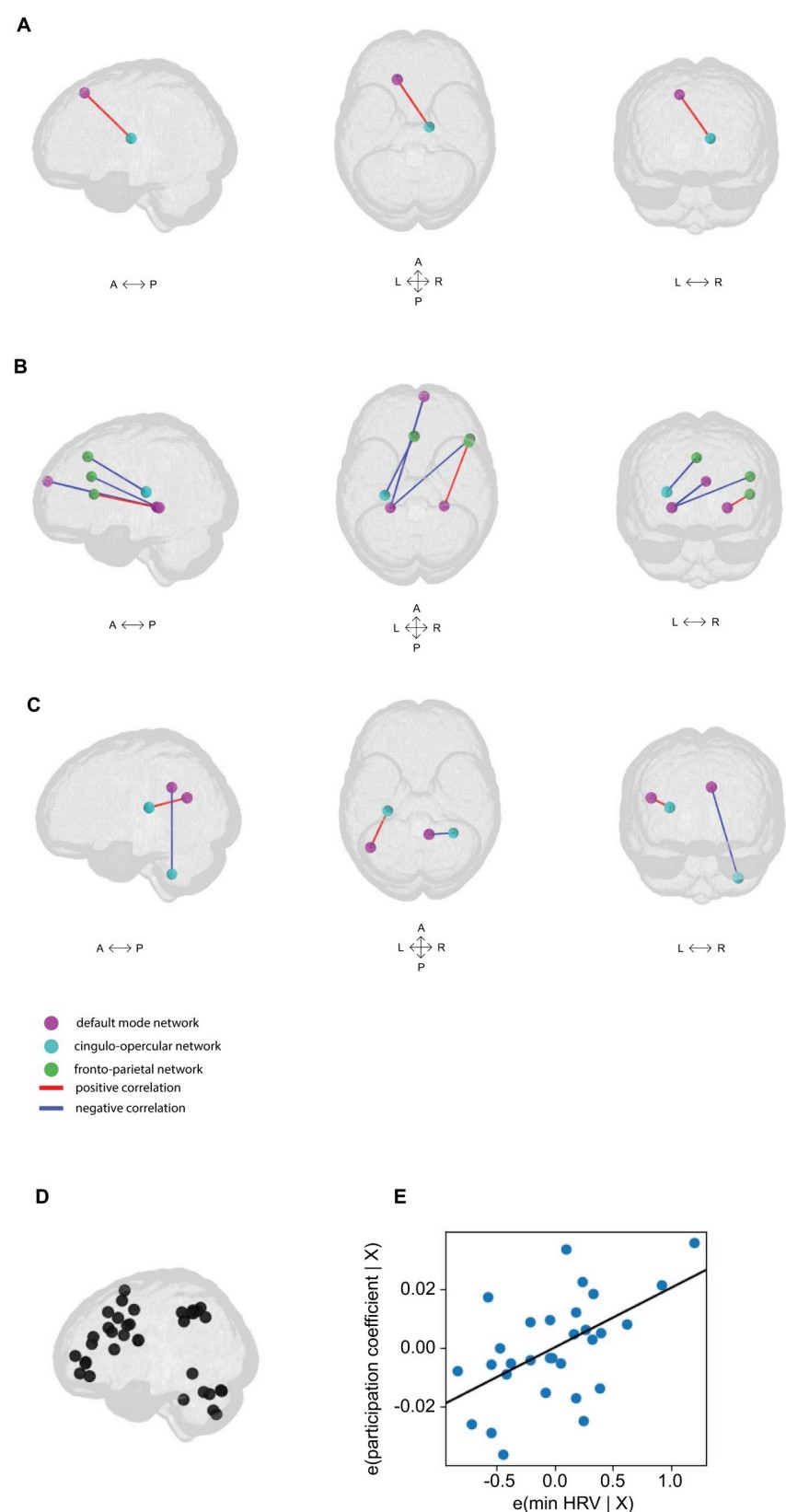

**Fig 5. Functional connectivity during resting-state is linearly dependent on prior night's sleep quality and previous day's heart rate variability.** (A) Linear regression models on individual links showed significant associations between the previous day's awake time and the connectivity between the prefrontal cortex in the DMN (purple) and a subcortical area in the CON (cyan). (B) Analyses also revealed significant relationships between the microsleep time in the scanner and the resting-state connectivity in the DMN, cingulo-opercular, and FPNs. (C) Similarly, regression analysis demonstrated a direct proportional relationship between prior night's maximum heart rate variability and the connectivity between links in the DMN and CON. Red colors indicate positive correlations and blue colors indicate negative correlations. (D) Nodes from the FPN employed to compute the network's participation coefficient. (E) Partial regression plot showing that the FPN's participation coefficient is strongly predicted by the previous day's minimum heart rate variability. Results are empirically thresholded via 10,000 iterations of nonparametric permutation testing and further corrected for multiple comparisons (corrected $p < 0.05$). Brain plots were generated with netplotbrain [171]. Unprocessed study data can be found in the Zenodo data set release [175]. Processed results derived from the study data are accessible in the GIT repository [176], under the results folder. CON, cingulo-opercular network; DMN, default mode network; FPN, fronto-parietal network.

notable impact on the FPN participation coefficient ($\beta = 0.86$, $p < 0.01$) (Fig 5D and 5E). In other words, lower minimum HRV in the previous day is associated with lower between-network integration for the FPN. This pattern persisted when the global signal was regressed from the rs-fMRI data (S22 Fig and S15 Table), but not when the data was analyzed with a second parcellation. Instead, when employing a second parcellation, we found that mean HRV was associated with CON participation coefficient ($\beta = 0.83$, $p < 0.05$). We also found that total sleep duration ($\beta = -0.62$, $p < 0.05$), mean negative affect ($\beta = -0.52$, $p < 0.05$), mean HRV ($\beta = 0.81$, $p < 0.05$), and maximum HRV ($\beta = -0.52$, $p < 0.05$) were found to drive changes in the DMN participation coefficient (S21 Fig).

### H4: Increased similarity in sleep, autonomic nervous system activity, and mood is not found to be related to increased similarity in the default mode, frontoparietal, and salience network

To assess the relationship between brain connectivity and behavior (i.e., sleep, ANS, or mood) during movie-watching, we computed Mantel tests between the pairwise ISC for each ROI and the similarity matrix for each behavioral variable. It is important to keep in mind that in order to compute the pairwise ISC for each ROI, all sessions must have comparable time series. Therefore, any volumes scrubbed from one session due to high head movement must inevitably be scrubbed from the other sessions as well. To verify the quality of the data, we computed the total number of scrubbed volumes by session, following the exclusion criteria FD < 0.2. Upon inspection, we noticed that following such rigorous scrubbing discarded 61% of the volumes for all sessions, leaving only a few untouched segments in the first third part of the film (S23 Fig).

Using this strict criterion, we found no significant relationship between behavioral factors and brain ISC for any of the parcellations. Nevertheless, when regressing the global signal, we identified a significant relationship between the brain ISC and the previous day's mean HRV ($\rho = -0.19$, $p < 0.05$) in the left medial frontal gyrus (MNI coordinates x = −5.5, y = 29.3, z = 44) using the AK model. We also observed a significant relationship between the previous day's mean negative affect ($\rho = -0.19$, $p < 0.05$) in the left middle temporal gyrus (MNI coordinates x = −53.1, y = −11.4, z = −16) using the NN model. Both results were obtained using the alternative parcellation.

In an effort to preserve the high standards of data quality while also keeping more fMRI volumes, we also adopted a percentage-based scrubbing approach for the movie watching task. This method involves discarding a particular volume from all sessions if it is flagged (FD > 0.2) in at least a percentage of sessions. For example, for a 30 session sample, a 10% threshold excludes all volumes where FD > 0.2 in at least 3 sessions. We applied this

percentage-based scrubbing with 2 values 10% and 5% and we were able to improve the percentage of discarded volumes (5% discarded and 32% discarded, respectively, for the 2 percentage-based scrubbing strategies). The number of volumes scrubbed using these percentages are reported in the Supporting information (S24 and S25 Figs).

Other external factors were found to be related to the daily-ISC when we analyzed the data with the additional scrubbing percentages and models (S16 Table). From these results, we highlight the negative relationship between the previous day's restless sleep and ISC in the left superior parietal sulcus using the NN model ($\rho = -0.26$, $p < 0.05$) for percentage-based scrubbing at 5% when the global signal is regressed (S16 Table). Likewise, the positive relationship between the mean HRV and the ISC activity in the left medial superior frontal gyrus using the AK model ($\rho = 0.19$, $p < 0.05$), which is consistent with for 2 scrubbing strategies when the data is analyzed with a second parcellation.

## Q2: Behavioral, physiological, and lifestyle factors experienced by the individual up to the preceding 15 days affect today's functional brain connectivity patterns

### H5: Sleep patterns experienced over the past 15 days are correlated with functional connectivity in the fronto-parietal, default mode, somatomotor, and cingulo-opercular networks during sustained attention tasks

We employed time-lagged cross-correlation analyses to identify the relationships between past sleep behaviors and brain network estimates (global efficiency and participation coefficient) for the DMN, FPN, CON, and somatomotor network. Null distributions were created based on 10,000 correlations between the brain estimates and surrogate data. We also corrected for multiple comparisons, assuming each tuple of variable-lag is independent, i.e., 4 networks for each variable-lag tuple.

Cross-correlation analyses revealed that the global efficiency within nodes of the DMN and FPN is correlated with the sleep habits of 2 weeks in the past. In particular, sleep duration ($\rho = 0.51$, $p < 0.05$), awake time in bed ($\rho = -0.48$, $p < 0.05$), and restless sleep ($\rho = 0.46$, $p < 0.05$) experienced on the 14th day prior correlate with the global efficiency in the DMN (Fig 6A), and the FPN (Fig 6B). In addition, the awake time spent in bed 3 days prior is associated with the FPN within-network integration estimate ($\rho = 0.54$, $p < 0.01$) (Fig 6B). For other correlation statistics, see S17 Table.

The analyses also showed that sleep patterns experienced beyond the previous day are related to how easily the DMN, FPN, CON, and somatomotor networks communicate with other networks. For example, sleep duration ($\rho = 0.4$, $p < 0.05$) experienced on the 15th day prior and restlessness ($\rho = 0.39$, $p < 0.05$) experienced on the seventh day prior are significantly correlated with the participation coefficient of the DMN (Fig 6E) and FPN (Fig 6F). Further, awake time in bed 6 days earlier is associated with the between-network connectivity for the CON ($\rho = 0.40$, $p < 0.05$) (Fig 6G) and somatomotor network ($\rho = 0.36$, $p < 0.05$) (Fig 6H). Likewise, prior sleep duration and restlessness correlate with the somatomotor network's participation coefficient with lags spanning from 6 to 14 days (for full statistics, see S18 Table).

The majority of the findings showed a positive correlation, indicating that more hours of sleep, awake time in bed, and interruptions during sleep are associated with an increase in global efficiency and participation coefficient. This pattern suggests that multiple factors involved in the sleep quality—and not only the sleep duration—are important for the efficiency of node communication within and between networks.

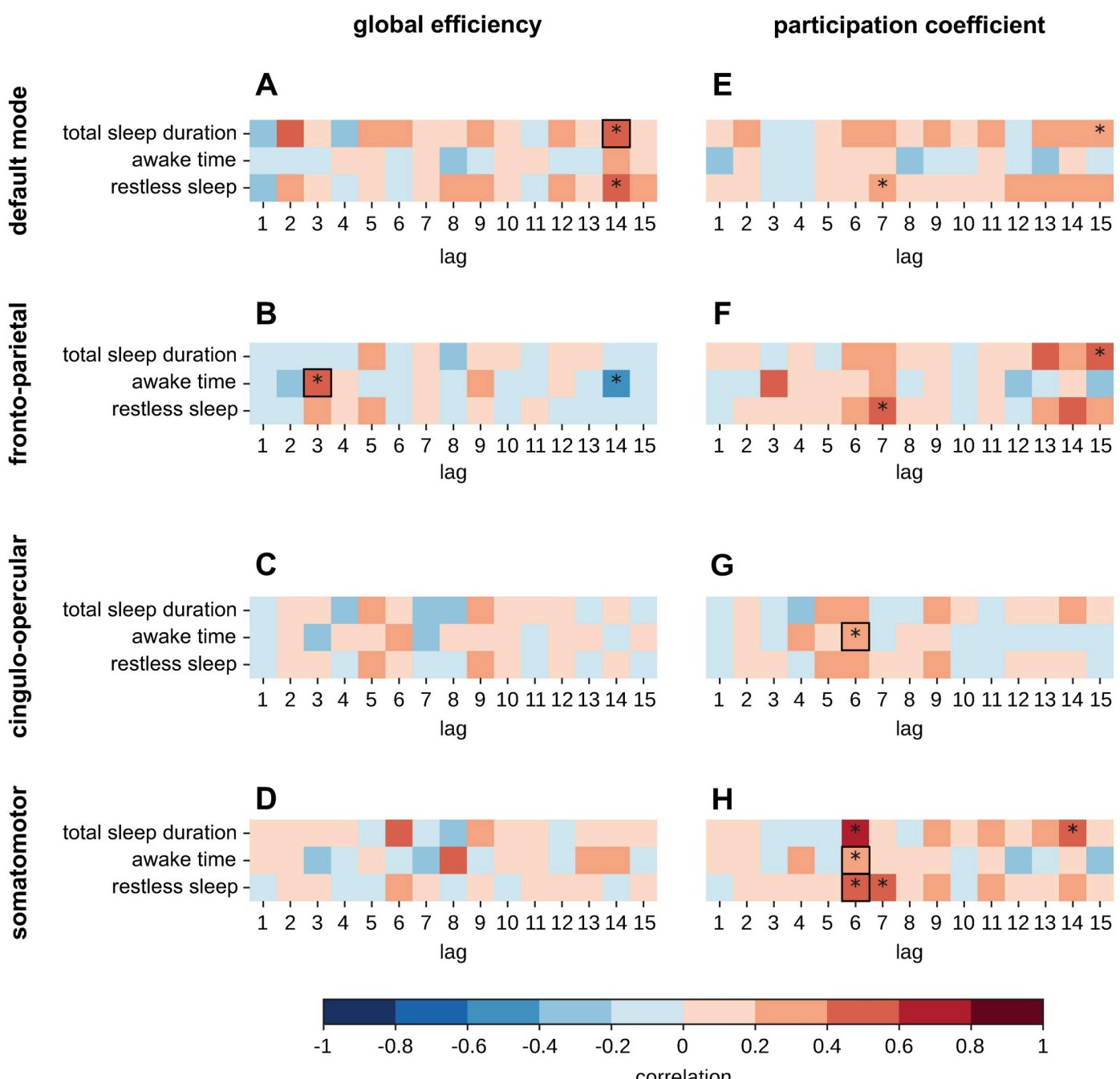

**Fig 6. Whole-brain functional connectivity during sustained attention tasks is associated with sleep patterns experienced over the past 15 days.** Sleep patterns from 3 and 14 days in the past are correlated with the global efficiency in the DMN (A), and FPNs (B). Similarly, sleep patterns from 6 to 15 days in the past are correlated with the participation coefficient in the DMN (E), fronto-parietal (F), cingulo-opercular (G), and somatomotor (H) networks. Significant correlations are shown by an asterisk (*). The data was also analyzed using different thresholds, parcellations, and regressing the global signal. Squares show correlations that we found significant in at least 2 analyses. Sleep measurements are correlated with whole-brain connectivity at different time lags, which suggests that different aspects of sleep history have specific, time-sensitive relationships with distinct brain network functionalities. Unprocessed study data can be found in the Zenodo data set release [175]. Processed results derived from the study data are accessible in the GIT repository [176], under the results folder. DMN, default mode network; FPN, fronto-parietal network.

We found different significant time-lagged cross-correlations when analyzing the data with other proportional thresholds (S26 and S27 Figs and S17 and S18 Tables), another parcellation (S28 Fig and S19 Table), and when including the global signal as a regressor (S29 Fig and S20 Table). Fig 6 shows the variable-lag tuples that were found to be significant in at least 2 separate analyses of the data. Few lags were consistent across all analyses. In particular, we observed that the DMN's and FPN's global efficiency consistently correlates with short lags of awake time and long lags of sleep duration, while their participation coefficient remained unaffected. Conversely, the CON's and somatomotor network's participation coefficient is associated with the awake time ($\rho$ = 0.4 for CON, $\rho$ = 0.36 for somatomotor, $p < 0.05$) and restlessness ($\rho$ = 0.53 for somatomotor, $p < 0.01$) 6 days prior. This result suggests that in attention tasks, the influence of awake time on both within- and between-network integration occurs within a 1-week timeframe.

### H6: Sleep and physical activity patterns experienced over the past 15 days are correlated with functional connectivity in the default mode, fronto-parietal, and somatomotor networks during working memory tasks

Using the same approach as H5, we investigated the relationship between external factors, such as past sleep and activity patterns, and brain connectivity estimates. These estimates are computed for the DMN, FPN, and somatomotor network when undergoing working memory tasks. We also corrected for multiple comparisons, assuming each tuple of variable-lag is independent, i.e., 3 networks for each variable-lag tuple.

Results from these analyses revealed correlations for short lags (i.e., less than 7 days) in awake time, number of steps, and inactive time with global efficiency in the DMN (Fig 7A) and somatomotor network (Fig 7C). For full statistics (correlation and $p$-values), see S21 Table. In particular, the inactive time ($\rho$ = 0.43, $p < 0.05$) and number of steps ($\rho$ = −0.42, $p < 0.05$) taken on the third day prior were found to be related to the DMN's global efficiency.

Similarly, we noted significant short-lag correlations between the number of steps taken 5 days earlier and the participation coefficient of the DMN ($\rho$ = −0.37, $p < 0.05$) (Fig 7D), FPN ($\rho$ = −0.48, $p < 0.01$) (Fig 7E), and somatomotor network ($\rho$ = −0.54, $p < 0.01$) (Fig 5F). These negative correlations suggest that reduced step count 5 days prior is associated with increased between-network estimate of the 3 aforementioned networks. In contrast, sleep duration 7 days prior showed a significant positive correlation with the participation coefficient in the DMN ($\rho$ = 0.4, $p < 0.05$) and somatomotor ($\rho$ = 0.4, $p < 0.05$) networks.

For time lags exceeding 7 days, our results show correlations between awake time, number of steps, and inactive time with the global efficiency in the DMN (Fig 7A) and somatomotor network (Fig 7C). For full statistics (correlation and $p$-values), see S21 Table. We also observed negative long-lag correlations between sleep patterns and the participation coefficient of the DMN (lag 15, $\rho$ = 0.43, $p < 0.05$) (Fig 7D) and somatomotor network (lag 15, $\rho$ = 0.37, $p < 0.05$) (Fig 7F). Moreover, the number of steps and inactive time experienced 11 to 12 days prior are associated with the participation coefficient in the FPN ($\rho$ = 0.49, $p < 0.01$) (Fig 7E) and somatomotor network ($\rho$ = 0.5, $p < 0.01$) (Fig 7F). These results suggest that reduced physical activity and sleep hours from more than a week ago correlates with decreased number of links between the FPN, somatomotor areas, and other networks compared to the links inside each network. Notably, many of these significant correlations occur around the eighth-day or fourteenth-day period, indicating a consistent pattern aligned with 1 and 2-week intervals.

Upon re-analyzing the data with other thresholds (S30 and S31 Figs and S21 and S22 Tables), another parcellation (S32 Fig and S23 Table), and including the global signal as a regressor (S33 Fig and S24 Table), we observed several significant time-lagged cross-

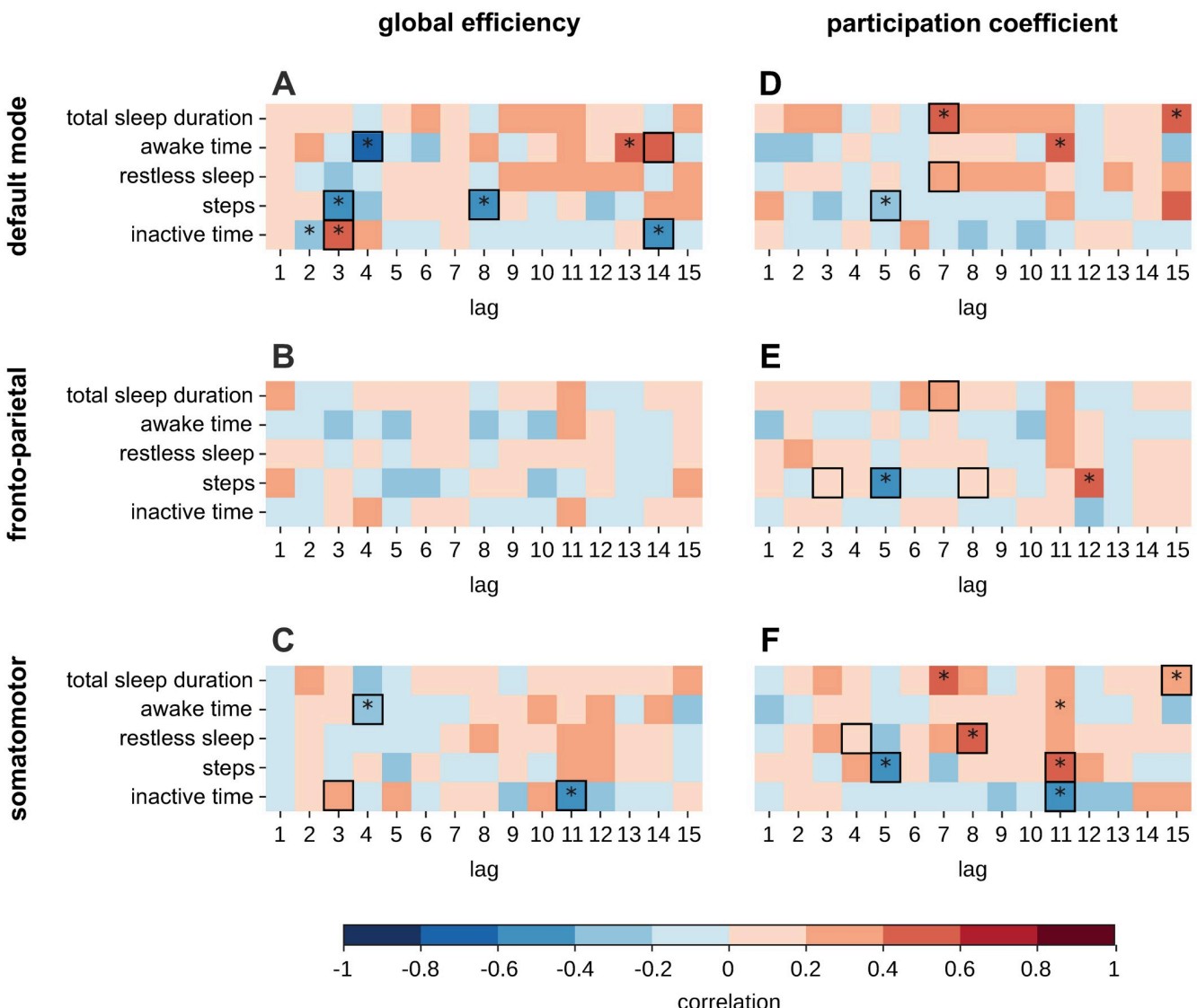

**Fig 7. Whole-brain functional connectivity during working memory tasks is associated with sleep and activity patterns experienced over the past 15 days.** Sleep and activity patterns from 2 to 14 days in the past are correlated with the global efficiency in the DMN (A), and somatomotor networks (C). Similarly, sleep and activity patterns from 5 to 15 days in the past are correlated with the participation coefficient in the DMN (D), fronto-parietal (E), and somatomotor (F) networks. Significant correlations are shown by an asterisk (*). The data was also analyzed using different thresholds, parcellations, and regressing the global signal. Squares show correlations that we found significant in at least 2 analyses. Sleep and activity measurements are correlated with whole-brain connectivity at varying time lags, which suggests that different aspects of sleep and activity history have specific, time-sensitive relationships with distinct brain network functionalities. Unprocessed study data can be found in the Zenodo data set release [175]. Processed results derived from the study data are accessible in the GIT repository [176], under the results folder. DMN, default mode network.

correlations not found in the principal analysis, especially for the FPN's participation coefficient. Fig 7 shows the stability of the correlations upon re-analyses of the data. Nevertheless, we found that several time-lagged correlations remained stable across different analyses. For example, the DMN's global efficiency showed repeated time-lag correlations with awake time and inactivity. Similarly, the network's participation coefficient consistently related to dynamic sleep and activity patterns. These results suggest that in working-memory tasks, an individual's previous sleep and, more importantly, physical activity steadily correlates both the within- and

between-network integration of the DMN and somatomotor network. The correlation appears to follow short cycles of approximately 3 days and longer cycles of about 11 days.

### H7: Sleep, autonomic nervous system activity, and mood patterns experienced over the past 15 days are correlated with functional connectivity in the default mode, fronto-parietal, and cingulo-opercular networks during resting-state fMRI

To investigate the time-lagged relationships between external factors and resting-state brain network estimates, we ran a series of cross-correlation analyses. We assessed significance by comparing our results with 10,000 correlations based on surrogate data. We also corrected for multiple comparisons, assuming each tuple of variable-lag is independent, i.e., 3 networks for each variable-lag tuple.

Our results suggest that the DMN's global efficiency predominantly correlates with ANS activity (respiratory rate and HRV) experienced in the prior days, but not with prior sleep factors (Fig 8A). All correlation and $p$-values are reported in the S25 Table. Similarly, the FPN's global efficiency is mostly associated with prior respiratory rate (lag 2, $\rho = 0.5$, $p < 0.01$ and lag 3, $\rho = -0.46$, $p < 0.05$), although sleep hours from 10 days earlier ($\rho = 0.42$, $p < 0.05$) also bear a significant correlation (Fig 8B). Instead, positive affect lags are significant factors influencing the CON's global efficiency (lag 2, $\rho = 0.4$, $p < 0.05$ and lag 8, $\rho = 0.4$, $p < 0.05$) (Fig 8C). These patterns indicate that while the DMN and FPN's global efficiency may be dynamically linked with ANS activity, mood factors may play a more substantial role in influencing the CON within-network integration.

In contrast, correlations between the DMN's participation coefficients and past ANS activity seem less dominant. Instead, factors like awake time in bed from 15 days prior ($\rho = -0.44$, $p < 0.05$) and recent mood changes, up to 2 days before (negative affect, lag 2, $\rho = -0.4$, $p < 0.05$ and positive affect, lag 3, $\rho = 0.53$, $p < 0.01$), appear more influential in the DMN (Fig 8D). Similarly, the FPN's participation coefficient shows a strong, short-lag correlation with mood effects and HRV (Fig 8E). A similar trend is seen in the CON's participation coefficient, especially with short-lag correlations. For example, sleep patterns, specifically sleep duration and restlessness from the past 3 days, suggest a correlation with the CON's between-network estimate (Fig 8F). All correlation and $p$-values are reported in the S26 Table. Therefore, we observed that the between-network integration seems more closely linked to shorter cycles of sleep, mood, and ANS activity, compared to the within-network estimate.

Notably, most of the dynamic correlations between ANS factors and the DMN's global efficiency are negative (see S26 Table). This suggests that lower respiratory rates and HRV may be linked to more efficient information transfer within the DMN (Fig 8A). Conversely, the majority of time-lagged correlations between the ANS activity and the networks' participation coefficients are positive (Fig 8D, 8E and 8F). Likewise, we noted positive short-lag correlations between mood factors and the CON's connectivity. In particular, more negative feelings (lag 5, $\rho = 0.61$, $p < 0.05$ and lag 6, $\rho = 0.48$, $p < 0.05$) and stress levels (lag 7, $\rho = 0.51$, $p < 0.05$) are linked to higher CON's within and between network integration (Fig 8C and 8F).

Similar to H5 and H6, we re-analyzed the resting-state data using other thresholds (S34 and S35 Figs and S25 and S26 Tables), parcellation (S36 Fig and S27 Table), and including the global signal as a regressor in the fMRI data preprocessing stage (S37 Fig and S28 Table). Fig 8 shows the variable-lag pairs that we found statistically significant using at least 2 different analysis approaches. Results from re-analyses indicate high stability in time-lagged correlations of ANS factors. Specifically, correlations between the DMN's global efficiency and the respiration rate 7 and 13 days prior seem consistently significant across all analysis variants. A similar pattern is observed for the FPN's participation coefficient and the previous day's HRV, showing significant correlations in every analysis. Overall, these findings suggest that for rs-

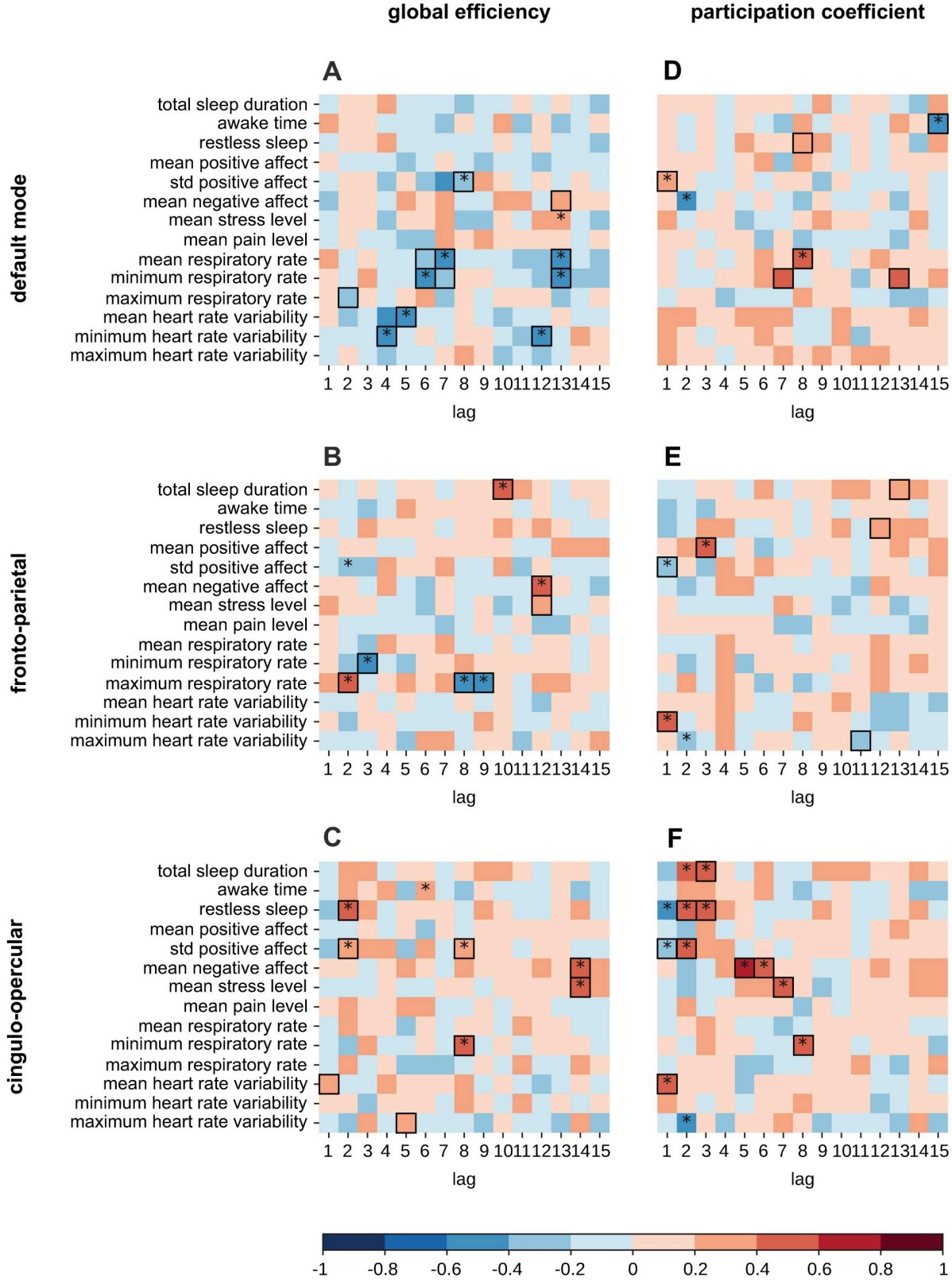

**Fig 8. Whole-brain functional connectivity during resting state is associated with sleep, mood, and autonomic nervous system activity patterns experienced over the past 15 days.** Patterns from 2 to 14 days in the past are correlated with the global efficiency in the DMN (A), fronto-parietal (B), and cingulo-opercular (C) networks. Similarly, sleep, activity and ANS activity patterns from the previous day up to 15 days in the past are correlated with the participation coefficient in the DMN (D), fronto-parietal (E), and cingulo-opercular (F) networks. Significant correlations are shown by an asterisk (*). The data was also analyzed using different thresholds, parcellations, and regressing the global signal. Squares show correlations that we found significant in at least 2 analyses. External factor measurements are correlated with whole-brain connectivity at different time lags, which suggests that different aspects of the individual's sleep, mood, and ANS history have specific, time-sensitive relationships with distinct brain networks. Unprocessed study data can be found in the Zenodo data set release [175]. Processed results derived from the study data are accessible in the GIT repository [176], under the results folder. ANS, autonomic nervous system; DMN, default mode network.

connectivity, there is an important relationship between HRV on both within- and between-network integration, occurring within a timeframe of 7 to 13 days prior.

### H8: Sleep, mood, and ANS activity experienced over the past 15 days are correlated with classification accuracy during movie-watching tasks

To understand the relationship between brain activity and past external factors such as sleep, mood, and ANS activity, during movie-watching stimuli, we run a series of regression models on classification accuracy brain maps. These maps represent the accuracy of a classifier to correctly identify specific brain patterns based on the highest similarity observed between segments of different sessions. Regression results reveal that the classification accuracy is significantly correlated with past positive affect (t = 6.55, $p < 0.05$, MNI coordinates x = 66, y = −30, z = 8), respiration rate, and sleep duration (t = 5.66, $p < 0.05$, MNI coordinates x = 24, y = −76, z = 42) (Fig 9). In particular, the maximum respiration rate experienced 5 (t = 5.47, $p < 0.05$, MNI coordinates x = −40, y = −66, z = −14), 8 (t = 6.62, $p < 0.05$, MNI coordinates x = 54, y = −12, z = 40), and 11 days (t = 5.76, $p < 0.05$, MNI coordinates x = −48, y = −74, z = 28) prior correlates with the classification accuracy in the left fusiform, the right postcentral gyrus, the right precentral gyrus, and the middle occipital gyrus.

Supplementary analyses, including longer sliding windows and global signal regression, demonstrated that the maximum respiration rate from the prior 8 and 11 days is correlated with the classification accuracy (S38–S40 Figs and for full statistics and peak MNI coordinates, see S29–S31 Tables). In fact, the maximum respiration rate experienced 8 days prior exhibited greater activation in the right postcentral gyrus (Fig 9C) across all approaches. Conversely, the maximum respiration rate experienced 11 days before showed consistent brain activation in the right precentral gyrus only when the global signal was regressed from the data (S40D Fig and S31 Table). Nevertheless, for this particular variable-lag combination, consistent activation in the right postcentral gyrus was found in longer sliding windows (S38D and S39E Figs), but not in the main analysis. Supplementary analyses also revealed a consistent association in the right superior temporal gyrus between the classification maps and the standard deviation of the positive affect experienced 2 days prior, across different sliding windows.

All the mentioned analyses employed a LOO cross-validation for the decoding maps. While each LOO analysis took up to 8 hours, a leave-20% cross-validation would require an impractical 33 days. Therefore, supplementary analyses using alternative cross-validation methods were not conducted. Nevertheless, we analyzed some centroids of ROIs and compared these findings with our main classification accuracy maps. The high correlation between these results suggest that different cross-validation methods would likely yield similar results (S41 Fig).

Together, these results suggest that when using naturalistic stimuli such as movies, brain activity is related to respiration rate patterns in specific brain regions within 8 to 11 days.

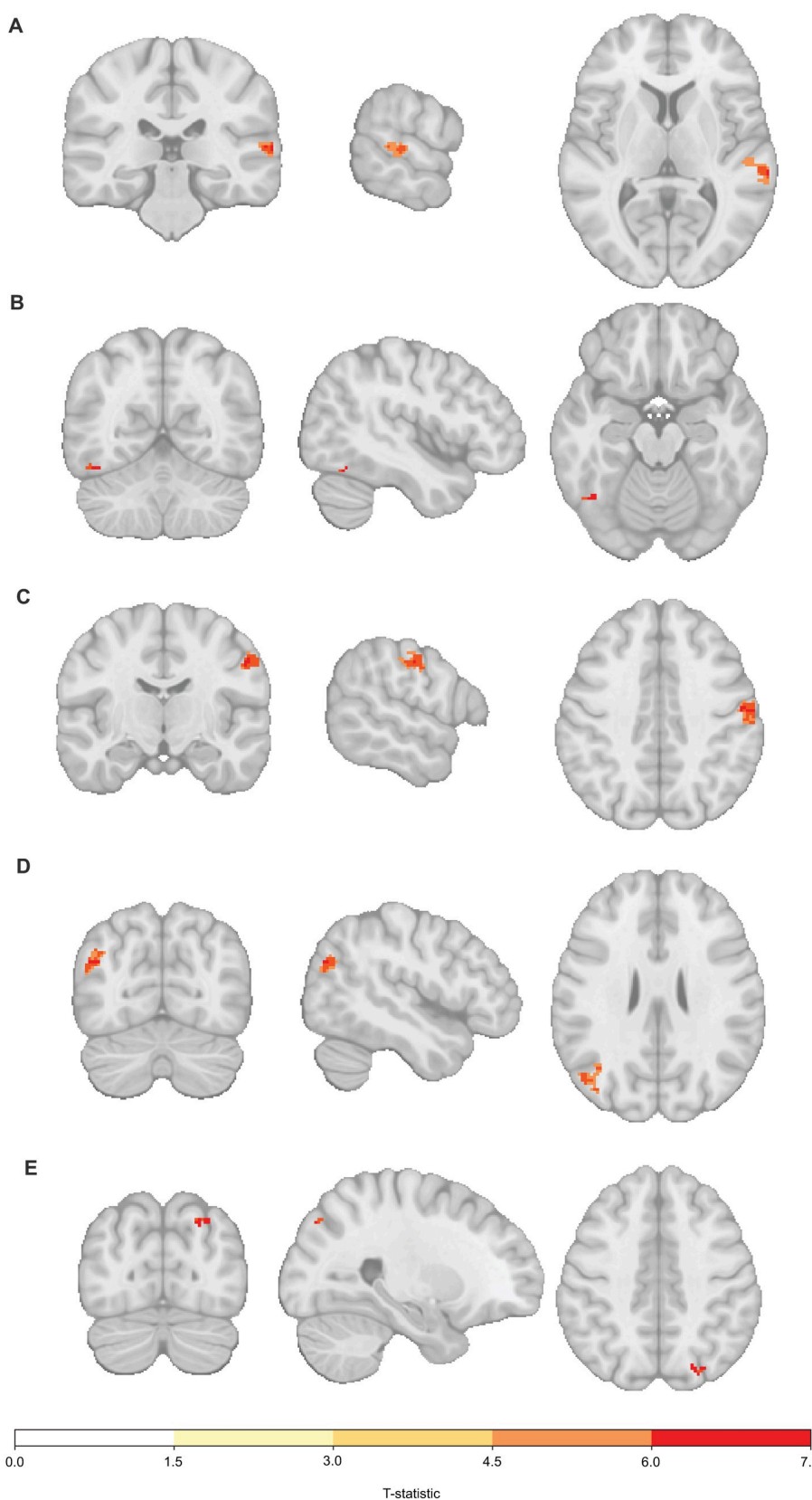

**Fig 9. Between-days movie time-segment classification accuracy is associated with sleep, mood, and respiration rate patterns experienced over the past 15 days.** The classification accuracy is related to: (A) standard deviation of the positive affect at lag 2, (B) maximum respiratory rate at lag 5, (C) maximum respiratory rate at lag 8, (D) maximum respiratory rate at lag 11, and (E) total sleep duration lag 15. Plots were generated with nilearn [170]. Unprocessed study data can be found in the Zenodo data set release [175]. Processed results derived from the study data are accessible in the GIT repository [176], under the results folder.

## Additional analysis

Following the proposed methods, we computed 3 additional analyses on the task-fMRI data. These computed mass univariate statistics for: (i) the sustained attention task; (ii) the working memory task; and (iii) the working memory task including the perceptual load. We found a significant association between the previous day's sleep duration and the BOLD signal during the PVT task (t = 7.39, $p < 0.05$, MNI coordinates x = 55.04, y = −58.54, z = 19.89) (S42 Fig). No other significant results were found.

Standardized regression analysis between the PVT's mean 1/RT and the total sleep time, sleep efficiency, and sleep latency did not yield any significant results. Similarly, we found no significant results for the regression analysis of the n-back's accuracy and the total sleep time, sleep efficiency, sleep latency, steps, and inactive time.

## Discussion

In this precision functional mapping project, we collected fMRI, physiological, and behavioral data from a single individual over a period of 133 days. Twice weekly MRI brain scans included 4 different fMRI conditions, comprising traditional cognitive tasks, resting state, and video viewing. Daily physiological and behavioral data were collected using smartphones and wearables. The combination of these 2 data sources allowed us to capture changes in brain activity and behavior that unfold over the course of 5 months in an individual. Regression models show diverse associations of external factors such as sleep, activity, mood, and physiology with brain network dynamics in the default mode, fronto-parietal, cingulo-opercular, and somatomotor networks. Time-lagged cross-correlation analyses suggest that the associations between brain connectivity and these external factors extend beyond the previous day, spanning up to 15 days in the past, possibly reflecting a more prolonged and sustained relationship between external factors and neural processes. Our results also showcase the viability of incorporating widely used wearables within precision mapping designs, extending from the prevalent physical activity and sleep sensors to the less-explored physiological measurements [177]. These details and research approaches promote integrated understanding of brain activity, adaptive physiology, and environmental factors. Such insights underpin individualized, precision medicine [178] and environmental neuroscience research [179].

In this work, we used 3 measures of functional connectivity: (i) the links, which can be defined as correlations between the averaged BOLD signal between ROIs; (ii) the global efficiency, defined as the average inverse shortest path length of the brain regions in a network; and (iii) the participation coefficient, defined as the relation between the number of links connecting a node outside its community and the total number of links for that particular node. Consequently, in this work, increased link connectivity signifies increased correlation between BOLD signals of different brain regions, increased global efficiency (or efficiency) implies more efficient signal transmission from brain regions of a subnetwork due their short pathways, and increased participation coefficient (or integration) implies that on average, the subnetwork ROIs are more functionally connected with ROIs outside of their primary subnetwork.

## Functional connectivity during sustained attention and working memory tasks is modulated by external factors experienced on the previous days

Fragmented sleep increases subjective fatigue and deteriorates inhibition, even when the total sleep duration remains unchanged [180]. Therefore, it is not surprising that supporting our hypotheses, our results show that restlessness is strongly associated with changes in brain connectivity during sustained attention tasks. Indeed, repeated disruptions in sleep are known to progressively reduce attention, with vigilant attention deficits building up across consecutive days of sleep restriction in a dose-dependent manner [181]. In line with previous cross-sectional research, our results suggest associations between past sleep duration [182,183] and restlessness [184] and brain connectivity. Also according to previous research [77,185], we observed that longer sleep hours and more interruptions relate to increased DMN global efficiency and somatomotor integration.

Sleep also plays a crucial role in efficient functioning of working memory [64,186]. We observed that longer sleep and more interruptions during sleep are associated with greater integration of the DMN and FPN. However, the effect of sleep duration on working memory does not seem to occur immediately but is observed after a 2-week delay. These results align well with our findings on sustained attention, reinforcing the idea that attention is integral to working memory processes [187]. While the long delay in the influences of sleep was unexpected, it is well known that there are notable differences in how resilient people are to sleep loss [181,188–190]. Since this is a single-subject study, it is possible that the subject is not particularly vulnerable to fluctuations in sleep duration.

Our results suggest that physical activity plays an important role in influencing changes in functional connectivity related to working memory over time. In particular, reduced physical activity on the day before relates to diminished FPN integration. This is in line with previous findings from pre-post comparisons related to a physical exercise intervention showing the influences of physical activity on working memory [66,67,191] and reduced FPN connectivity [192]. Conversely, with the present experimental design, we found associations between brain connectivity and both recent (<3 days) and earlier (>10 days) physical activity. These findings suggest that physical activity levels have both immediate and delayed effects on brain networks, which supports the idea that regular physical activity might be crucial for maintaining optimal brain network integration and communication over time.

## Prior heart rate variability was associated with resting-state functional connectivity

The neurovisceral integration model describes the physiological interplay between brain activity and heart rate, via the autonomic nervous system and feedback loops [193]. For example, resting HRV is known to correlate with cortical midline and limbic brain activity (reviewed by Matusik and colleagues [194]). As evidenced by cross-sectional group-level findings, HRV has complex associations with resting-state connectivity [83,84,195]. In particular, cognitive flexibility and related temporal adaptivity of the DMN and salience networks have been linked to increased resting HRV in healthy subjects [84,195], potentially through the inhibition of amygdala activity [85,193]. Our results suggest that recent HRV (<5 days) correlates with regional functional connectivity at rest. Specifically, FPN was predicted by the previous day's resting HRV, an association not previously reported in group-based studies. This is notable given we regressed the in-scanner heart and respiration rate from the fMRI data, underscoring the robustness of the relationship between HRV and connectivity. Although elucidation of focal brain–heart interactions continues [194], measurement of an individual's activity between the

heart and the FPN might allow researchers and clinicians to precisely quantify neurological stress responses or cognitive resource allocation.

Our results also suggest that past mood and sleep are also associated with brain connectivity at rest and during movie-watching. In line with previous dense-sampling, deep-phenotyping research [8], we observe that positive mood is correlated with network participation coefficient. Nevertheless, instead of focusing solely on same-day data, our analysis extends to lagged patterns. Unexpectedly, we found that recent past sleep is mostly related to the integration of task-control networks with other networks. Although these areas are not commonly reported in relationship with sleep, it has been suggested that sleep quality is a key factor influencing how aerobic exercise alters connectivity between the anterior insula (a CON area) and the hippocampus [196].

## Brain connectivity and external factors relate with different time delays

The human brain is in constant change, from morning to evening [197], from youth to adulthood [198]. Similarly, human behavioral patterns change at different timescales. Here, we suggest that both systems fluctuate, but not in unison. Instead, external factors such as sleep, physical activity, mood, and ANS activity seem to be dynamically related to brain connectivity with distinct time delays. This means that changes in these external factors do not immediately relate to changes in brain efficiency and integration, reflecting a more prolonged and sustained influence on neural processes.

The *delayed* relationship between behavioral changes and brain connectivity for this subject seems to manifest in 2 distinct phases: a short wave, generally lasting less than 7 days, and a long wave, typically spanning from 7 to 15 days. Our results suggest that the subject's brain connectivity relates to sleep duration in long lags and to ANS activity in short lags. Because physiological processes such a HRV exhibit more rapid fluctuations than changes in, e.g., sleep patterns, it is expected that the effects of HRV would manifest over shorter time frames rather than extended periods. This is supported by studies indicating that HRV can rapidly respond to stimuli and internal states [199,200], contrasting with sleep duration, which show slower, more gradual changes [199]. Consequently, while ANS activity might influence brain connectivity in a prompt way, the impact of other behavioral patterns is expected to unfold over longer periods of time. Other factors may also mediate these observed lags. For example, the level of estradiol rises after the seventh day of the menstrual cycle and peaks around the fourteenth day. Given that sex hormones are recognized to affect women's sleep patterns [201], HRV [202], and brain connectivity [28], it is likely that hormones are playing a mediating factor in these observed lags.

The differentiation between these observed waves is intriguing. The short wave might reflect the brain's immediate or acute reaction to external factors. For example, sustained attention can be swiftly impacted by a few nights of poor sleep [203]. However, this relationship is likely transient, allowing the brain to return to a baseline state [204]. In contrast, the observed long wave suggests a more complex, nonlinear interaction between the external factors and brain connectivity, hinting that a singular event might trigger a series of changes that unfold over time, and would not necessarily require continuous exposure to the triggering stimulus. These 2 waves seem more present in connectivity under cognitive demanding tasks such as attention and working memory in both efficiency and integration of brain networks. Conversely, for rs-fMRI, short waves seem more related to network integration, while longer waves tend to correlate with network efficiency.

## Day-to-day variability of functional connectivity

Recent studies have highlighted that functional brain networks show variability across individuals, sessions, and tasks. While some research suggests these networks remain stable over time

[205], others indicate significant variability [206,207]. In particular, Gratton and colleagues [23] demonstrated that although functional networks are fundamentally stable, they show moderate task-based changes that vary significantly among individuals, with minimal yet measurable session-based variability. This minimal variation has been also measured by other researchers using larger frequent sampling data sets [8,27,28], indicating that although connectivity remains predominantly stable, frequent sampling with sufficient power enables the detection of subtle session-to-session changes.

Although our study does not primarily focus on comparing day-to-day and task-related variations, our results offer insights into the variability associated with both daily activities and task states. We identified distinct factors influencing different tasks at various time points. With our sample size ($n$ = 30), we can detect subtle effects that might be missed in smaller data sets.

Scientists have highlighted the significance of stability in functional connectivity for biomarker discovery. Similarly, our findings suggest that while individual-based connectivity shows the most substantial variability, day-to-day variations may also play a crucial role, for example, in analyzing the progression of disorders in personalized medicine. This aspect might provide valuable insights for diagnostics and the development of targeted treatments.

## Limitations and future studies

Though the findings reported in this study are promising, several considerations should be taken into account. First, this study includes data from a single individual, which limits our ability to generalize the findings to other subjects. Due to the diversity of individual lifestyles, it is unlikely that the specific patterns of delays and variables associated with one person's functional connectivity may be exactly replicated in another [208]. Therefore, to enrich our understanding of the dynamic relationship between environmental, physiological, and lifestyle factors with functional connectivity, this precision mapping study should be extended to a representative sample. This would allow us to examine the consistency of the dynamic connectivity patterns between subjects, regardless of their lifestyle. We expect that patterns related to physiological processes (e.g., respiration rate) to be more consistent between individuals than those associated with behavioral factors (e.g., mood), which are heavily influenced by a range of external disturbances [209]. In addition, examining day-to-day variation in network organization during controlled, induced changes, similarly to Newbold and colleagues [210], would provide more information about how the brain reorganizes to adapt, withstand, and recover from external perturbations.

Second, we used 2 models to understand the dynamic relationship between external factors and brain connectivity. The first model took the previous day's factors into account, while the second model included longer lags. However, the second model analyzed the variables and lags independently, so no interplay between the variables was assessed. Future studies that extend the lag ranges in the GLM, or apply partial directed coherence, could allow us to comprehend the temporal dynamics and potential delayed effects of external factors in brain connectivity. Similar to previous research [28], employing more complex methods such as vector autoregression (VAR) in future studies could also improve our understanding of the brain dynamics, as brain connectivity would be modeled not just using past external factors, but also using its previous states.

Third, while our analysis considered nonparametric measures and linear relationships, we did not analyze the potential nonlinear relationships between external factors and brain connectivity. However, many human processes can be modeled as nonlinear, e.g., HRV during exercise [211], functional connectivity between brain regions [212], and brain connectivity

during sleep [213]. Therefore, incorporating analysis of nonlinear dynamics in future research may deepen our understanding of the complex interplay between external factors and brain connectivity.

Fourth, our experimental design allowed investigating how external factors influence functional connectivity up to 15 days in the past. However, it is unknown if the influence of these factors extend beyond this 2-week time range. Moreover, while we sampled all factors daily, not all factors seem to relate to connectivity on the same timescale. This suggests that the ideal sampling frequency may vary depending on the specific factor being studied. Future work is needed to understand these temporal dynamics and to determine the most appropriate sampling rate for each factor.

Fifth, in this work, we focused on associating current brain connectivity with past environmental, physiological, and lifestyle factors. However, we did not explore the possibility that present connectivity may be predictive of future factors, i.e., consider the idea that not only our brain is influenced by past experiences, but that our connectivity shapes our responses to future events. This is particularly relevant given the established bidirectional relationships in time observed in other areas, such as the interplay between sleep and the maintenance of cortical thickness [13]. Therefore, future research should explore this bidirectionality to gain a more comprehensive understanding of these brain dynamics.

Sixth, our primary focus was on examining the direct dynamic impact of external variables on brain connectivity. However, it is important to acknowledge several unexplored factors that may contribute to the observed results. For example, we need to consider whether there is autocorrelation in the sleep, mood, and activity data, and how it might influence the models, particularly the lagged-cross correlations. Moreover, what might initially appear as an effect of past sleep on current functional connectivity could potentially stem from factors like gradually accumulating sleep debt over time. Similarly, some effects could be influenced by the differences of the week versus weekend behavioral patterns. Therefore, while our primary aim has been to establish and demonstrate the existence and measurability of these relationships, it is important to recognize that many other questions remain open.

Finally, we employed group-based atlases [158] to improve the generalizability of our findings because the atlases provide brain regions demonstrated to be reliable across hundreds of subjects. Additionally, group-based parcellations facilitate comparisons with existing literature by providing a common framework. However, recent work has shown that group-based parcellations obscure individual differences [26] and individual parcellations also change with cognitive states [214,215]. While reanalysis with individual parcellations is beyond the scope of this current study, we might speculate that such an approach could not only corroborate the relationships we have observed between external factors and brain connectivity, but might also refine our insights into the spatial dynamics of the brain. Therefore, future analyses using individual parcellations per task are needed to understand whether external factors drive reconfigurations in the parcels. In addition, re-analyses of the data with individualized parcellations could provide further insights into the stability and reproducibility of the results.

## Implications and broader impact

Our study signals complex dynamic interactions between various behavioral factors and brain connectivity. Precision mapping designs emerge as a promising framework for investigating these mechanisms. Due to its exploratory nature, our research suggests several external factors that are related to daily functional connectivity and might be worth exploring in other individuals. Such factors could serve as a foundation for identifying which aspects should be further investigated and for studying the mechanisms driving the observed relationships. This may be

particularly relevant as factors such as sleep are known to impact cortical structure [13], synaptic strength [216], and cerebrospinal fluid flow rhythms [217], similar to other biological factors (e.g., hormones [28], synaptic proteins [218], and glial processes [219]) that may affect connectivity changes.

Different precision mapping studies have established that individual functional networks are largely stable [23], with subtle variations over time [8,28,210]. Our findings confirm this variability and extend previous research by demonstrating the feasibility of relating the variability with changes in several behavioral measurements in ecologically valid contexts. This insight is crucial, considering that mental health disorders show increased variability in brain function and patterns measured by digital phenotyping over the span of weeks [27]. Our findings highlight the potential of combining these models to assess whether alterations in individual functional connectivity correspond to personalized models of behavior during distinct disorder states [220]. Furthermore, the use of wearables and smartphones suggests a cost-effective alternative to MRI for monitoring patient progress and correlating it with brain connectivity. As an example, researchers could monitor the behavioral impacts of connectivity-based transcranial magnetic stimulation interventions. Scientists might also obtain higher adherence rates in subjects as wearables could potentially inform about changes in brain connectivity with fewer MRI sessions required.

This experiment was designed to assess how behavioral, physiological, and lifestyle factors relate to the variability of brain connectivity, challenging the assumption that just a few trials are sufficient for correctly sampling an individual's brain activity and behavior. While we captured daily variability in brain and behavior, the results shown here raise critical questions: How do these dynamic factors contribute to the heterogeneity observed in functional connectivity, and how can researchers account for them, especially if they fall outside the study's focus? For example, our results suggest that there is a time-lagged correlation in HRV and respiration rate with current brain connectivity. These results persist even when controlling for heart and respiration rates during scanning, suggesting that past physiological states may continue to influence the BOLD signal and consequently, our indirect measures of neural activity. Such influence may hinder the interpretation of fMRI studies, as suggested by Duyn and colleagues [221].

While we explored the impact of specific lags, it may be impractical to control for each of these in an fMRI study, especially as the number of subjects increases. An alternative would be to investigate a cumulative effect of past days, e.g., the previous week. However, whether this would yield similar results to the individual lags is still an open question. If true, using a cumulative approach might be a more practical alternative for controlling external factors deemed as confounds. If managing these lags turns out challenging, we recommend that fMRI researchers at least consider recent behavioral and physiological patterns of their subjects as potential confounds in their analyses. For example, researchers should consider stratifying subjects according to their measured sleep patterns using actigraphs and including these subgroups into the analysis as possible confounding factors. This inclusion, even at a group level, might significantly enhance the accuracy of fMRI findings and provide more information about heterogeneous results.

Finally, network science has become a valuable framework for understanding complex interactions across biological, psychological, environmental, and neuronal systems. While using the multilayer framework to integrate brain imaging data with behavioral data [222] has been suggested, actual experiments employing these approaches remain limited to analysis of momentaneous brain data variation [223]. We suggest that precision mapping studies that also collect dense-sampling behavioral data provide a sufficient amount of high-resolution data to apply multilayer approaches in novel ways. For example, different data sources can be modeled

as layers, different days as nodes, and correlations between daily time series as links. By doing so, we could integrate different behavioral and biological aspects as layers within a single network, addressing the interdependence of biological, psychological, and environmental factors. Moreover, these models could allow predicting how changes in one layer (the brain) might affect another (the behavior), acknowledging their interdependent nature and moving away from reductionist views where the brain is analyzed in isolation.

## Conclusions

In this precision mapping study, we collected task, resting-state, and naturalistic fMRI data, conducting one of the first studies to use these stimuli in a deep sampling design. More specifically, we also extended the methods for collecting behavioral data in precision functional mapping studies by incorporating smartphones and wearables, ensuring more reliable data. Our findings suggest that behavioral, physiological, and lifestyle factors correlate with brain connectivity across different timescales, in both the short (<7 days) and longer term (<2 weeks). These results are useful for the generation of new hypotheses and can serve as a basis for further investigations of intraindividual variability. Likewise, the publicly available data offer a unique opportunity to develop new methods for analyzing multivariate brain and sensor data. They also provide a basis for power analyses for subsequent longitudinal studies, especially when combining fMRI, smartphones, and wearables over days, weeks, and months. The integration of brain activity, physiology data, and environmental cues will support precision healthcare [178] and future environmental neuroscience research [178,179].

## Supporting information

**S1 Text. Additional information about the pilots and additional analysis.** This includes description of the pilot studies and information about additional analysis and quality controls for the full dataset.
(DOCX)

**S1 Fig. Cognitive tasks schematics.** (A) PVT task [42]. The participant must observe a red screen and press a button as soon as a yellow counter appears on it. The counter shows the milliseconds elapsed between the start of the counter and the button press, i.e., reaction time (RT). Lapses are counted if the button is pressed when no stimulus is shown, or if the RT is longer than 500 ms. The period between the last response and the new stimulus varies between 2 and 10 s. (B) N-back task [43]. Auditory and visual stimuli are presented simultaneously. Only 1 stimulus (either visual or auditory) changes per trial. Auditory stimuli and visual stimuli are sinewave gratings with occasional auditory or visual distractors. The participant should press the buttons up/down when the pitch has increased/decreased. The participant should also press the buttons left/right when the grading has changed counterclockwise/clockwise.
(TIF)

**S2 Fig. PVT scores vs. time for Pilot study I.** The participant tested the PVT task for 15 consecutive days ($n = 15$) and her performance was monitored. The PVT scores were computed according to Basner and Dinges [42]. The figure shows the scores for (A) mean 1/RT, (B) slowest 1/RT, (C) median RT, (D) fastest 10% RT, (E) number of lapses, (F) lapse probability, and (G) performance. Similar to previous reports [42], we observed no apparent learning effects in any of the scores for the PVT task. Pilot I data can be found in the Zenodo release [174].
(TIF)

**S3 Fig. Quality assessment for data collected with smartphones and wearables during pilot study I ($n$ = 15).** We plot 1 feature from each sensor and highlight in red the days where data are missing. We estimate the missing data ratio as the proportion of missing data points over the total number of data points each sensor should have gathered. We show the data quality for (A) sleep, (B) activity, (C) Experience Sampling Method (ESM), (D) battery, (E) light, (F) WiFi, and (G) GPS location data. Overall, the data quality is good, with most sensors losing less than 10% of the data. Only the light sensor lost approximately 20% of the data and there seems to be a pattern of lost data every second day that cannot be associated with other sensors. Pilot I data can be found in the Zenodo release [174].
(TIF)

**S4 Fig. N-back learning effects.** The participant tested the n-back task for 14 consecutive days ($n$ = 14) and her performance was monitored. Three n-back scores were computed: the number of correct answers, the number of wrong answers, and the number of missing answers (i.e., the subject failed to press a button). The figure shows the scores for (A) 1-back, and (B) 2-back. For both tasks, there is high variability in the number of correct and missing answers at the beginning of the pilot study. As time goes by, the variability decreases and the number of correct and missing answers slightly oscillates over a stable count. It takes approximately 7 days for the 1-back task, and 10 days for the 2-back task to reach this stability. In addition, the task difficulty d' was assessed for (C) pitch changes (octaves), and (D) visual angle (degrees). Pilot II data can be found in the Zenodo release [174].
(TIF)

**S5 Fig. Preprocessing strategy analysis.** We inspected how our preprocessing choices affected the BOLD signal by comparing the images before and after denoising. Our aim is to remove the chosen confounds, without losing the signal, so a cleaning effect needs to be evident, but moderate. We see that: (A) Most of the affected voxels are in the cortex, where the signal of interest lies. Moreover, the correlation coefficient shows that the impact is moderate. (B) Similarly, the distribution of the correlation coefficient of voxels shows a low proportion of the voxels being hardly affected (i.e., harsh denoising). In this case, higher correlations imply a stronger effect of the denoising strategy. (C) The denoising strategy impacts the global signal (GS). Drift effects are apparent before regressing the confounds and filtering. (D) The denoising effect is also visible in individual voxel signals. We chose a voxel from the cortex at random and plotted its signal before and after the confound regression and filtering. Both signals looked similar, but some peaks have been attenuated after denoising (voxel after). (E) Correlation between the global signal and the brain voxels. Most of the global signal was highly correlated with voxels in the cortex and areas of interest. (F) Correlation of the global signal and a voxel from the cortex chosen at random. (G) Carpet plots [224] of the signal before and after denoising. The carpet plot after denoising (bottom) is less noisy and more greatly resembles white noise than the carpet plot before denoising (top), a feature that is desirable. Brain plots were generated with nilearn [170]. Pilot III data can be found in the Zenodo release [174].
(TIF)

**S6 Fig. Temporal signal-to-noise ratio (tSNR) across tasks and sessions.** For each task and session, a tSNR map was generated by computing the ratio between mean and standard deviation for each voxel across time. This plot shows the tSNR values within a brainmask. Across tasks and sessions, the mean tSNR was 120.11. As expected, we observed signal dropout in areas close to air-tissue borders. Brain plots were generated with nilearn [170]. Pilot III data can be found in the Zenodo release [174].
(TIF)

**S7 Fig. Framewise displacement (FD) across tasks and sessions.** For each task and session, we plotted the FD as computed by fmriprep. In general, the signal quality was good; 99.2% of the time peaks were below 0.2 and no peaks were over 0.5. Pilot III data can be found in the Zenodo release [174].
(TIF)

**S8 Fig. Quality assessment for data collected with smartphones and wearables during pilot study III (*n* = 37 days).** We plotted 1 feature from each sensor and highlighted in red the days where data were missing. We estimate the missing data ratio as the proportion of missing data points over the total number of data points each sensor should have gathered. We show the data quality for (A) sleep, (B) activity, (C) ESM, (D) battery, (E) light, (F) WiFi, and (G) GPS location data. Overall, the data quality was good, with most sensors losing less than 10% of the data. Light and location sensors had more than 15% missing data points. While the light sensor had a pattern, the location sensor did not. Given the battery sensor had less than 2% missing data points, the light and location sensor missingness could be associated with technical problems of their own, rather than other factors such as the subject shutting down the phone. Pilot III data can be found in the Zenodo release [174].
(TIF)

**S9 Fig. Correlations between the smartring features.** (A) Full correlation matrix. (B) Correlation between a subset of sleep features. (C) Correlation between a subset of activity features. Features in B and C were selected based on 2 reasons: they are not scores computed automatically by the manufacturer and they are not related to sleep staging. For all matrices, we thresholded the upper triangle to show correlation coefficients whose absolute value is greater than 0.5. Pilot III data can be found in the Zenodo release [174].
(TIF)

**S10 Fig. Comparison of performance between different imputation methods for missing data.** We deleted 20% of the (A) sleep and (B) ESM collected data at random. Then, we employed each imputer to generate values on the artificial missing data. Finally, we computed the mean squared error (MSE) between the true value and the value yielded by the imputer and repeated the procedure 10,000 times. The zero, mean, and common imputers replaced missing data points with zeros, the mean, and the most frequent value of the data set, respectively. The linear imputer calculates a linear model based on the existent data and uses it to predict the missing data value. The knn imputer completes missing values based on the k-nearest neighbors method. The iterative imputer estimates new values based on other features from the same dataset. Finally, mice imputes incomplete multivariate data by chained equations [225]. Pilot III data can be found in the Zenodo release [174].
(TIF)

**S11 Fig. ISC comparison between pilot study data and data from part 2 of "The Grand Budapest Hotel" [44].** (A) ISC from the pilot study data. (B) ISC from the Budapest data set. We observe that similar areas in the brain are activated across sessions (pilot study) and subjects (Budapest). Nevertheless, the correlation values are stronger in the pilot study data. Brain plots were generated with nilearn [170]. Pilot III data can be found in the Zenodo release [174].
(TIF)

**S12 Fig. ISC comparison between the pilot study sessions.** (A) Session 1 vs. session 2. (B) Session 1 vs. session 3, and (C) session 2 vs. session 3. We observed high values for the ISC between the 3 sessions. We also observed similar activation clusters, with similar intensity.

Brain plots were generated with nilearn [170]. Pilot III data can be found in the Zenodo release [174].
(TIF)

**S13 Fig. Data quality for the main data set.** For each day (x-axis), we plotted the percentage of available data for each data source (y-axis). The percentage of available data represents the number of data points that were successfully collected over the total number of data points that were planned to be collected per data source in a day. Some sources provided data only a few days per week, such as the weekly questionnaires that were collected each Sunday (indicated by the red dotted lines), and the MRI sessions, scheduled on Mondays and Fridays. The data set shows minimal missingness, with most of the daily data reaching at least 60% completion, except for pulse rate variability and breathing rate, which have a lower collection rate 20% and 40%. Unprocessed study data can be found in the Zenodo data set release [175].
(TIF)

**S14 Fig. MRI and eye-tracker data quality.** (A) The subject head movement was minimal across most of the sessions and tasks, with mean average framewise displacement (FD) values between 0.07 and 0.17. (B) The subject also remained alert for most of the sessions, only experiencing microsleeps in 18 out of 120 acquisitions. Data from 8 acquisitions was corrupted and could not be used. Unprocessed study data can be found in the Zenodo dataset release [175].
(TIF)

**S15 Fig. Framewise displacement for each session.** The framewise displacement distributions are shown as violin plots for all sessions during (A) the PVT task, (B) resting state, (C) movie-watching, and (D) the n-back task. The average FD remained under 0.2 for all sessions during all tasks, albeit some high movement peaks. Higher peaks tended to happen during the n-back task. FD data as preprocessed by fmriprep [134,135] are accessible in the GIT repository [176], under the results folder.
(TIF)

**S16 Fig. Covariance matrix of the external factors measured with smartphone and wearables.** High correlations ($\rho > 0.7$) are marked with stars. For sleep measurements, high correlations between sleep efficiency and sleep latency are notorious. For mood measurements, high correlations between different statistics on pain levels, negative affects, and stress levels are relevant. In addition, the minimum negative affect and minimum stress levels did not exhibit variations (white stripes). For physiological measurements, we see high correlations between mean respiratory rate and median respiratory rate, maximum respiratory rate and the standard deviation of respiratory rate, mean HRV and median HRV, and maximum HRV and the standard deviation of HRV. Unprocessed study data can be found in the Zenodo dataset release [175].
(TIF)

**S17 Fig. Functional connectivity during sustained attention tasks is linearly dependent on the quality of sleep from the previous day.** (A) These results are derived using a second parcellation (set2 from Seitzman and colleagues [158]). In this case, restless sleep is associated with connectivity among nodes within the DMN (purple), cingulo-opercular (cyan), and somatomotor (yellow) networks. Red colors indicate positive correlations and blue colors indicate negative correlations. Results are empirically thresholded via 10,000 iterations of nonparametric permutation testing and further corrected for multiple comparisons (corrected $p < 0.05$). All links are listed in the S5 Table. Brain plots were generated with netplotbrain

[171]. Unprocessed study data can be found in the Zenodo dataset release [175]. Processed results derived from the study data are accessible in the GIT repository [176], under the results folder.
(TIF)

**S18 Fig. Functional connectivity during sustained attention tasks is linearly dependent on the quality of sleep from the previous day.** These results are derived by including the global signal as a regressor and using the set1 from Seitzman and colleagues [158]. (A) For this case, restless sleep is associated with connectivity among nodes within the DMN (purple), cingulo-opercular (cyan), fronto-parietal (green), and somatomotor (yellow) networks. Red colors indicate positive correlations and blue colors indicate negative correlations. (B) Nodes from the somatomotor network employed to compute the participation coefficient. (C) Partial regression plot showing that the somatomotor network's participation coefficient is proportionally related to the previous night's restless sleep. Results are empirically thresholded via 10,000 iterations of nonparametric permutation testing and further corrected for multiple comparisons (corrected $p < 0.05$). All links are listed in the S6 Table. Brain plots were generated with netplotbrain [171]. Unprocessed study data can be found in the Zenodo data set release [175]. Processed results derived from the study data are accessible in the GIT repository [176], under the results folder.
(TIF)

**S19 Fig. Functional connectivity during working memory tasks linearly depends on both prior night's sleep quality and previous day's inactivity time.** These results are derived using a second parcellation (set2 from Seitzman and colleagues [158]). (A) Linear regression models on individual links showed significant associations between the previous day's inactive time and links in the DMN (purple), fronto-parietal (green), and somatomotor (yellow) networks. (B) Similarly, analyses revealed significant relationships between the prior night's restless sleep and these same networks. Red colors indicate positive correlations and blue colors indicate negative correlations. (C) Nodes from the fronto-parietal network employed to compute the participation coefficient. (D) Partial regression plot showing that the fronto-parietal network's participation coefficient is proportionally related to the previous day's inactive time. Results are empirically thresholded via 10,000 iterations of non-parametric permutation testing and further corrected for multiple comparisons (corrected $p < 0.05$). All links are listed in the S9 Table. Brain plots were generated with netplotbrain [171]. Unprocessed study data can be found in the Zenodo data set release [175]. Processed results derived from the study data are accessible in the GIT repository [176], under the results folder.
(TIF)

**S20 Fig. Functional connectivity during working memory tasks linearly depends on both prior night's sleep quality and previous day's inactivity time.** These results are derived by including the global signal as a regressor and using the set1 from Seitzman and colleagues [158]. (A) Linear regression models on individual links showed significant associations between the previous day's inactive time and links in the DMN (purple), fronto-parietal (green), and somatomotor (yellow) networks. (B) Similarly, analyses revealed significant relationships between the prior night's restless sleep and these same networks. Red colors indicate positive correlations and blue colors indicate negative correlations. Results are empirically thresholded via 10,000 iterations of nonparametric permutation testing and further corrected for multiple comparisons (corrected $p < 0.05$). All links are listed in the S10 Table. Brain plots were generated with netplotbrain [171]. Unprocessed study data can be found in the Zenodo data set release [175]. Processed results derived from the study data are accessible in the GIT

repository [176], under the results folder.
(TIF)

**S21 Fig. Functional connectivity during resting-state is linearly dependent on prior night's sleep quality and previous day's heart rate variability.** These results are derived using a second parcellation (set2 from Seitzman and colleagues [158]). (A) Linear regression models on individual links showed significant associations between the previous day's awake time in bed and the connectivity between the cingulo-opercular network (cyan), FPN(green), and the DMN (purple). (B) Analyses also revealed significant relationships between the microsleep time in the scanner and the resting-state connectivity in the DMN, cingulo-opercular, and fronto-parietal networks. (C) Similarly, regression analysis demonstrated a direct proportional relationship between prior night's maximum heart rate variability and the connectivity between links in the DMN, FPN, and cingulo-opercular network. Red colors indicate positive correlations and blue colors indicate negative correlations. (D) Nodes from the DMN employed to compute the network's participation coefficient. Partial regression plots showing that the DMN participation coefficient is strongly predicted by the previous day's (E) total sleep duration, (F) mean negative affect, (G) mean heart rate variability, and (H) maximum heart rate variability. (I) Nodes from the cingulo-opercular network employed to compute the network's participation coefficient. (J) Partial regression plot showing that the CON participation coefficient is strongly predicted by the previous day's mean heart rate variability. Results are empirically thresholded via 10,000 iterations of nonparametric permutation testing and further corrected for multiple comparisons (corrected $p < 0.05$). All links are listed in the S13 Table. Brain plots were generated with netplotbrain [171]. Unprocessed study data can be found in the Zenodo data set release [175]. Processed results derived from the study data are accessible in the GIT repository [176], under the results folder.
(TIF)

**S22 Fig. Functional connectivity during resting-state is linearly dependent on the previous day's heart rate variability.** These results are derived by including the global signal as a regressor and using the set1 from Seitzman and colleagues [158]. (A) Linear regression models on individual links showed significant associations between the prior night's maximum heart rate variability and the connectivity between the cingulo-opercular network (cyan), FPN(green), and the DMN (purple). (B) Analyses also revealed significant relationships between the microsleep time in the scanner and the resting-state connectivity in the DMN, cingulo-opercular, and fronto-parietal networks. (C) Nodes from the FPN employed to compute the network's participation coefficient. Partial regression plots showing that the FPN participation coefficient is strongly predicted by the previous day's (D) minimum heart rate variability, (F) mean heart rate variability. Results are empirically thresholded via 10,000 iterations of nonparametric permutation testing and further corrected for multiple comparisons (corrected $p < 0.05$). All links are listed in the S14 Table. Results for the participation coefficient are listed in the S15 Table. Brain plots were generated with netplotbrain [171]. Unprocessed study data can be found in the Zenodo data set release [175]. Processed results derived from the study data are accessible in the GIT repository [176], under the results folder.
(TIF)

**S23 Fig. Movie data quality.** Because the ISC computations require fMRI data with equal number of volumes, ordinary scrubbing requires detecting all volumes with high movement (FD > 0.2) in each session and censoring them across all sessions. Applying this scrubbing technique censors 60.11% of the volumes for all sessions. We plotted the (A) volumes to be censored using ordinary scrubbing for the 30 sessions. Each TR volume is presented in the x-

axis, while sessions are shown in the y-axis. Gray colors show the FD and censored volumes are in red. (B) We also plot the number of sessions that exceeded the FD threshold. Unprocessed study data can be found in the Zenodo data set release [175]. Processed results derived from the study data are accessible in the GIT repository [176], under the results folder.
(TIF)

**S24 Fig. Movie data quality.** For each volume, percentage scrubbing involves detecting the total number of sessions ($n_s$) with high movement (FD > 0.2) and censoring the volume across all sessions only if $n_s$ is above a threshold. This threshold is usually related to a percentage of the sessions. Applying this scrubbing technique censors 31.47% of the volumes for all sessions. We plotted the (A) volumes to be censored using percentage scrubbing at 5% (i.e., minimum 2 sessions with high movement). Each TR volume is presented in the x-axis, while sessions are shown in the y-axis. Gray colors show the FD and censored volumes are in red. (B) We also plot the number of sessions that exceeded the FD threshold. Unprocessed study data can be found in the Zenodo data set release [175]. Processed results derived from the study data are accessible in the GIT repository [176], under the results folder.
(TIF)

**S25 Fig. Movie data quality.** For each volume, percentage scrubbing involves detecting the total number of sessions ($n_s$) with high movement (FD > 0.2) and censoring the volume across all sessions only if $n_s$ is above a threshold. This threshold is usually related to a percentage of the sessions. Applying this scrubbing technique censors 4.63% of the volumes for all sessions. We plotted the (A) volumes to be censored using percentage scrubbing at 10% (i.e., minimum 3 sessions with high movement). Each TR volume is presented in the x-axis, while sessions are shown in the y-axis. Gray colors show the FD and censored volumes are in red. (B) We also plot the number of sessions that exceeded the FD threshold. Unprocessed study data can be found in the Zenodo data set release [175]. Processed results derived from the study data are accessible in the GIT repository [176], under the results folder.
(TIF)

**S26 Fig. Whole-brain functional connectivity during sustained attention tasks is associated with sleep patterns experienced over the past 15 days.** These results are derived by thresholding the network at 20% proportional threshold. Sleep patterns from 3 and 14 days in the past are correlated with the global efficiency in the (A) DMN, and (B) FPN. No significant correlations are found for sleep patterns and (C) CON, and the (D) somatomotor network. Similarly, no correlation patterns are found for the participation coefficient of the (E) DMN or (F) FPN. Nevertheless, sleep patterns from 6 to 9 days in the past are correlated with the participation coefficient in the (G) CON, and (H) somatomotor network. Significant correlations are shown by an asterisk (*). Unprocessed study data can be found in the Zenodo data set release [175]. Processed results derived from the study data are accessible in the GIT repository [176], under the results folder.
(TIF)

**S27 Fig. Whole-brain functional connectivity during sustained attention tasks is associated with sleep patterns experienced over the past 15 days.** These results are derived by thresholding the network at 30% proportional threshold. Sleep patterns from 3 days in the past are correlated with the global efficiency in the (B) FPN. No significant correlations are found for sleep patterns and the (A) DMN, (C) CON, and the (D) somatomotor network. Similarly, no correlation patterns are found for the participation coefficient of the (E) DMN or (F) FPN. Nevertheless, sleep patterns from 4 to 6 days in the past are correlated with the participation coefficient in the (G) CON, and (H) somatomotor network. Significant correlations are shown

by an asterisk (*). Unprocessed study data can be found in the Zenodo data set release [175]. Processed results derived from the study data are accessible in the GIT repository [176], under the results folder.
(TIF)

**S28 Fig. Whole-brain functional connectivity during sustained attention tasks is associated with sleep patterns experienced over the past 15 days.** These results are derived by using the set2 from Seitzman and colleagues [158] with 10% proportional threshold. Sleep patterns from 3 to 12 days in the past are correlated with the global efficiency in the (A) DMN and the (D) somatomotor network. No significant correlations are found for sleep patterns and the (B) FPN, and (C) CON. Similarly, sleep patterns from 3 to 14 days in the past are correlated with the participation coefficient in the (E) DMN and (F) FPN. No correlation patterns are found for the participation coefficient of the (G) CON or (H) somatomotor network. Significant correlations are shown by an asterisk (*). Unprocessed study data can be found in the Zenodo data set release [175]. Processed results derived from the study data are accessible in the GIT repository [176], under the results folder.
(TIF)

**S29 Fig. Whole-brain functional connectivity during sustained attention tasks is associated with sleep patterns experienced over the past 15 days.** These results are derived by including the global signal as a regressor and using the set1 from Seitzman and colleagues [158] with 10% proportional threshold. Sleep patterns from 9 to 12 days in the past are correlated with the global efficiency in the (A) DMN and the (D) somatomotor network. No significant correlations are found for sleep patterns and the (B) FPN, and (C) CON. Similarly, no correlation patterns are found for the participation coefficient of the (E) DMN, (F) FPN, or (G) CON. Nevertheless, sleep patterns from 6 days in the past are correlated with the participation coefficient in the (H) somatomotor network. Significant correlations are shown by an asterisk (*). Unprocessed study data can be found in the Zenodo data set release [175]. Processed results derived from the study data are accessible in the GIT repository [176], under the results folder.
(TIF)

**S30 Fig. Whole-brain functional connectivity during working memory tasks is associated with sleep and activity patterns experienced over the past 15 days.** These results are derived by thresholding the network at 20% proportional threshold. Sleep and activity patterns from 3 to 14 days in the past are correlated with the global efficiency in the (A) DMN, (B) FPN, and (C) somatomotor networks. Similarly, sleep and activity patterns from 3 to 8 days in the past are correlated with the participation coefficient in the (D) DMN, (E) FPN, and (F) somatomotor networks. Significant correlations are shown by an asterisk (*). Unprocessed study data can be found in the Zenodo data set release [175]. Processed results derived from the study data are accessible in the GIT repository [176], under the results folder.
(TIF)

**S31 Fig. Whole-brain functional connectivity during working memory tasks is associated with sleep and activity patterns experienced over the past 15 days.** These results are derived by thresholding the network at 30% proportional threshold. Sleep and activity patterns from 3 to 14 days in the past are correlated with the global efficiency in the (A) DMN, (B) FPN, and (C) somatomotor networks. Similarly, sleep and activity patterns from 3 to 13 days in the past are correlated with the participation coefficient in the (D) DMN, (E) FPN, and (F) somatomotor networks. Significant correlations are shown by an asterisk (*). Unprocessed study data can be found in the Zenodo data set release [175]. Processed results derived from the study

data are accessible in the GIT repository [176], under the results folder.
(TIF)

**S32 Fig. Whole-brain functional connectivity during working memory tasks is associated with sleep and activity patterns experienced over the past 15 days.** These results are derived by using the set2 from Seitzman and colleagues [158] with 10% proportional threshold. Sleep and activity patterns from 3 to 15 days in the past are correlated with the global efficiency in the (A) DMN, and the (B) FPN. No results are found for the (C) somatomotor network. Similarly, sleep and activity patterns from 1 to 15 days in the past are correlated with the participation coefficient in the (D) DMN, (E) FPN, and (F) somatomotor networks. Significant correlations are shown by an asterisk (*). Unprocessed study data can be found in the Zenodo data set release [175]. Processed results derived from the study data are accessible in the GIT repository [176], under the results folder.
(TIF)

**S33 Fig. Whole-brain functional connectivity during working memory tasks is associated with sleep and activity patterns experienced over the past 15 days.** These results are derived by including the global signal as a regressor and using the set1 from Seitzman and colleagues [158] with 10% proportional threshold. Sleep and activity patterns from 2 to 14 days in the past are correlated with the global efficiency in the (A) DMN, (B) FPN, and (C) somatomotor networks. Similarly, sleep and activity patterns from 4 to 12 days in the past are correlated with the participation coefficient in the (E) FPN, and (F) somatomotor networks. No significant results are found for the (D) DMN. Significant correlations are shown by an asterisk (*). Unprocessed study data can be found in the Zenodo data set release [175]. Processed results derived from the study data are accessible in the GIT repository [176], under the results folder.
(TIF)

**S34 Fig. Whole-brain functional connectivity during resting state is associated with sleep, mood, and autonomic nervous system activity patterns experienced over the past 15 days.** These results are derived by thresholding the network at 20% proportional threshold. Patterns from the previous day to 14 days in the past are correlated with the global efficiency in the (A) DMN, (B) FPN, and (C) CON. Similarly, sleep, activity and ANS activity patterns from the previous day up to 13 days in the past are correlated with the participation coefficient in the (D) DMN, (E)FPN, and (F) CON. Significant correlations are shown by an asterisk (*). Unprocessed study data can be found in the Zenodo data set release [175]. Processed results derived from the study data are accessible in the GIT repository [176], under the results folder.
(TIF)

**S35 Fig. Whole-brain functional connectivity during resting state is associated with sleep, mood, and autonomic nervous system activity patterns experienced over the past 15 days.** These results are derived by thresholding the network at 30% proportional threshold. Patterns from the 2 to 14 days in the past are correlated with the global efficiency in the (A) DMN, (B) FPN, and (C) CON. Similarly, sleep, activity and ANS activity patterns from the previous day up to 15 days in the past are correlated with the participation coefficient in the (D) DMN, (E) FPN, and (F) CON. Significant correlations are shown by an asterisk (*). Unprocessed study data can be found in the Zenodo data set release [175]. Processed results derived from the study data are accessible in the GIT repository [176], under the results folder.
(TIF)

**S36 Fig. Whole-brain functional connectivity during resting state is associated with sleep, mood, and autonomic nervous system activity patterns experienced over the past 15 days.**

These results are derived by using the set2 from Seitzman and colleagues [158] with 10% proportional threshold. Patterns from the previous day to 15 days in the past are correlated with the global efficiency in the (A) DMN, (B) FPN, and (C) CON. Similarly, sleep, activity and ANS activity patterns from the previous day up to 15 days in the past are correlated with the participation coefficient in the (D) DMN, (E)FPN, and (F) CON. Significant correlations are shown by an asterisk (*). Unprocessed study data can be found in the Zenodo data set release [175]. Processed results derived from the study data are accessible in the GIT repository [176], under the results folder.
(TIF)

**S37 Fig. Whole-brain functional connectivity during resting state is associated with sleep, mood, and autonomic nervous system activity patterns experienced over the past 15 days.** These results are derived by including the global signal as a regressor and using the set1 from Seitzman and colleagues [158] with 10% proportional threshold. Patterns from the 2 to 15 days in the past are correlated with the global efficiency in the (A) DMN, (B) FPN, and (C) CON. Similarly, sleep, activity and ANS activity patterns from the previous day up to 15 days in the past are correlated with the participation coefficient in the (D) DMN, (E)FPN, and (F) CON. Significant correlations are shown by an asterisk (*). Unprocessed study data can be found in the Zenodo data set release [175]. Processed results derived from the study data are accessible in the GIT repository [176], under the results folder.
(TIF)

**S38 Fig. Between-days movie time-segment classification accuracy is associated with mood and respiration rate patterns experienced over the past 15 days.** These results are derived by shifting the sliding window with 2TRs. The classification accuracy is related to: (A) standard deviation of positive affect at lag 2, (B) mean positive affect at lag 4, (C) maximum respiration rate at lag 8, (D) maximum respiratory rate at lag 11, and (E) maximum respiratory rate at lag 12. Brain plots were generated with nilearn [170]. Unprocessed study data can be found in the Zenodo data set release [175]. Processed results derived from the study data are accessible in the GIT repository [176], under the results folder.
(TIF)

**S39 Fig. Between-days movie time-segment classification accuracy is associated with mood and respiration rate patterns experienced over the past 15 days.** These results are derived by shifting the sliding window with 4TRs. The classification accuracy is related to: (A) standard deviation of positive affect at lag 2. (B) Total sleep duration at lag 2. (C) Mean positive affect at lag 4. (D) Maximum respiration rate at lag 8. (E) Maximum respiratory rate at lag 11. (E) Maximum respiratory rate at lag 12. Brain plots were generated with nilearn [170]. Unprocessed study data can be found in the Zenodo data set release [175]. Processed results derived from the study data are accessible in the GIT repository [176], under the results folder.
(TIF)

**S40 Fig. Between-days movie time-segment classification accuracy is associated with mood and respiration rate patterns experienced over the past 15 days.** These results are derived by including the global signal as a regressor. The classification accuracy is related to: (A) mean pain lag 3. (B) Mean positive affect lag 4. (C) Maximum respiration rate lag 8. (D) Maximum respiratory rate lag 11. Brain plots were generated with nilearn [170]. Unprocessed study data can be found in the Zenodo data set release [175]. Processed results derived from the study data are accessible in the GIT repository [176], under the results folder.
(TIF)

**S41 Fig. Comparisons between the between-days movie time-segment classification accuracy with LOO (black) and 20% leave-out methods (median over 100 iterations in red).** For 3 different regions of interest the correlations between the LOO and the median of the 20%-out are reported. Processed results derived from the study data are accessible in the GIT repository [176], under the results folder.
(TIF)

**S42 Fig. Brain region showing a significant linear association between sleep duration and activations in the PVT task-fMRI.** The cluster is located at MNI coordinates x = 55.04, y = −-58.54, z = 19.89. The global signal was included as a regressor in the fMRI data preprocessing. Brain plots were generated with nilearn [170]. Unprocessed study data can be found in the Zenodo data set release [175]. Processed results derived from the study data are accessible in the GIT repository [176], under the results folder.
(TIF)

**S1 Table. Description of physiological, behavioral, and lifestyle data collected in the study.** Each device provides streams of signals that we use to understand the subject's fluctuations in functional brain connectivity, physiology, behavior, mental states, and cognition. We monitor the subject under laboratory conditions (MRI scanner) and in her own environment.
(XLSX)

**S2 Table. Daily questionnaire.** The subject is encouraged to answer these questions every day. There are questions asked exclusively in the morning and evening, whereas other questions are asked at random times during the day and along with the morning and evening questions.
(XLSX)

**S3 Table. Selected behavioral, physiological and lifestyle variables for hypothesis testing.**
(XLSX)

**S4 Table. Significant links associated with sleep external factors during sustained attention tasks.** For each link, we list only the external factor which yielded significance, along with its standardized β value. We also list the linked nodes in MNI coordinates and their networks according to the set1 from Seitzman and colleagues [158]. Stable links across different analyses are shown by an asterisk (*). Unprocessed study data can be found in the Zenodo data set release [175]. Processed results derived from the study data are accessible in the GIT repository [176], under the results folder.
(XLSX)

**S5 Table. Significant links associated with sleep external factors during sustained attention tasks.** These results were derived using a second parcellation (set2 from Seitzman and colleagues [158]). For each link, we list only the external factor which yielded significance, along with its standardized β value. We also list the linked nodes in MNI coordinates and their networks. All *p*-values have been corrected for multiple comparisons. Unprocessed study data can be found in the Zenodo data set release [175]. Processed results derived from the study data are accessible in the GIT repository [176], under the results folder.
(XLSX)

**S6 Table. Significant links associated with sleep external factors during sustained attention tasks.** These results are derived by including the global signal as a regressor and using the set1 from Seitzman and colleagues [158]. For each link, we list only the external factor which yielded significance, along with its standardized β value. We also list the linked nodes in MNI coordinates and their networks. All *p*-values have been corrected for multiple comparisons.

Unprocessed study data can be found in the Zenodo data set release [175]. Processed results derived from the study data are accessible in the GIT repository [176], under the results folder.
(XLSX)

**S7 Table. Significant global efficiency associated with sleep external factors during sustained attention tasks.** Each network was thresholded at different proportional values. Then, we computed the global efficiency and ran the regression model described in Table 1, H1. Here, we list only the significant factor which yielded significance, along with its standardized β value. These results are derived by including the global signal as a regressor and using the set1 from Seitzman and colleagues [158]. All *p*-values have been corrected for multiple comparisons. Unprocessed study data can be found in the Zenodo data set release [175]. Processed results derived from the study data are accessible in the GIT repository [176], under the results folder.
(XLSX)

**S8 Table. Significant links associated with external factors during working memory tasks.** For each link, we list only the external factor which yielded significance, along with its standardized β value. We also list the linked nodes in MNI coordinates and their networks according to the set1 from Seitzman and colleagues [158]. Stable links across different analyses are shown by an asterisk (*). All *p*-values have been corrected for multiple comparisons. Unprocessed study data can be found in the Zenodo data set release [175]. Processed results derived from the study data are accessible in the GIT repository [176], under the results folder.
(XLSX)

**S9 Table. Significant links associated with external factors during working memory tasks.** For each link, we list only the external factor which yielded significance, along with its standardized β value. We also list the linked nodes in MNI coordinates and their networks according to the set2 from Seitzman and colleagues [158]. All *p*-values have been corrected for multiple comparisons. Unprocessed study data can be found in the Zenodo data set release [175]. Processed results derived from the study data are accessible in the GIT repository [176], under the results folder.
(XLSX)

**S10 Table. Significant links associated with external factors during working memory tasks.** These results are derived by including the global signal as a regressor and using the set1 from Seitzman and colleagues [158]. For each link, we list only the external factor which yielded significance, along with its standardized β value. We also list the linked nodes in MNI coordinates and their networks. All *p*-values have been corrected for multiple comparisons. Unprocessed study data can be found in the Zenodo data set release [175]. Processed results derived from the study data are accessible in the GIT repository [176], under the results folder.
(XLSX)

**S11 Table. Significant participation coefficient associated with sleep and activity external factors during working memory tasks.** Here, we list only the significant factor which yielded significance, along with its standardized β value. All *p*-values have been corrected for multiple comparisons. Unprocessed study data can be found in the Zenodo data set release [175]. Processed results derived from the study data are accessible in the GIT repository [176], under the results folder.
(XLSX)

**S12 Table. Significant links associated with external factors during resting state.** For each link, we list only the external factor which yielded significance, along with its standardized β

value. We also list the linked nodes in MNI coordinates and their networks according to the set1 from Seitzman and colleagues [158]. All *p*-values have been corrected for multiple comparisons. Unprocessed study data can be found in the Zenodo data set release [175]. Processed results derived from the study data are accessible in the GIT repository [176], under the results folder.
(XLSX)

**S13 Table. Significant links associated with external factors during resting state.** For each link, we list only the external factor which yielded significance, along with its standardized β value. We also list the linked nodes in MNI coordinates and their networks according to the set2 from Seitzman and colleagues [158]. All *p*-values have been corrected for multiple comparisons. Unprocessed study data can be found in the Zenodo data set release [175]. Processed results derived from the study data are accessible in the GIT repository [176], under the results folder.
(XLSX)

**S14 Table. Significant links associated with external factors during resting state.** These results are derived by including the global signal as a regressor and using the set1 from Seitzman and colleagues [158]. For each link, we list only the external factor which yielded significance, along with its standardized β value. We also list the linked nodes in MNI coordinates and their networks according to the set2 from Seitzman and colleagues [158]. All *p*-values have been corrected for multiple comparisons. Unprocessed study data can be found in the Zenodo data set release [175]. Processed results derived from the study data are accessible in the GIT repository [176], under the results folder.
(XLSX)

**S15 Table. Significant participation coefficient associated with external factors during resting state.** Each network was thresholded at different proportional values. Then, we computed the global efficiency and ran the regression model described in Table 1, H3. Here, we list only the significant factor which yielded significance, along with its standardized β value. All *p*-values have been corrected for multiple comparisons. Unprocessed study data can be found in the Zenodo data set release [175]. Processed results derived from the study data are accessible in the GIT repository [176], under the results folder.
(XLSX)

**S16 Table. Significant nodes associated with external factors during the movie-watching condition.** Here, we list only the significant factor which yielded significance, along with its correlation value. To assess how the *p*-value of each significant node varies according to the analyses approaches, we also listed their *p*-value. All *p*-values have been corrected for multiple comparisons. Unprocessed study data can be found in the Zenodo data set release [175]. Processed results derived from the study data are accessible in the GIT repository [176], under the results folder.
(XLSX)

**S17 Table. Significant relationships between lagged external factors and the global efficiency during attention tasks.** For each network, we list only the eternal factor at the lag that yielded significance, along with its correlation value (ρ). All *p*-values have been corrected for multiple comparisons. Unprocessed study data can be found in the Zenodo data set release [175]. Processed results derived from the study data are accessible in the GIT repository [176], under the results folder.
(XLSX)

**S18 Table. Significant relationships between lagged external factors and the participation coefficient during attention tasks.** For each network, we list only the eternal factor at the lag that yielded significance, along with its correlation value (ρ). All *p*-values have been corrected for multiple comparisons. Unprocessed study data can be found in the Zenodo data set release [175]. Processed results derived from the study data are accessible in the GIT repository [176], under the results folder.
(XLSX)

**S19 Table. Significant relationships between lagged external factors and network estimates during attention tasks.** These results are derived by using the set2 from Seitzman and colleagues [158] with 10% proportional threshold. For each network, we list only the eternal factor at the lag that yielded significance, along with its correlation value (ρ). All *p*-values have been corrected for multiple comparisons. Unprocessed study data can be found in the Zenodo dataset release [175]. Processed results derived from the study data are accessible in the GIT repository [176], under the results folder.
(XLSX)

**S20 Table. Significant relationships between lagged external factors and network estimates during attention tasks.** These results are derived by including the global signal as a regressor and using the set1 from Seitzman and colleagues [158] with 10% proportional threshold. For each network, we list only the eternal factor at the lag that yielded significance, along with its correlation value (ρ). All *p*-values have been corrected for multiple comparisons. Unprocessed study data can be found in the Zenodo data set release [175]. Processed results derived from the study data are accessible in the GIT repository [176], under the results folder.
(XLSX)

**S21 Table. Significant relationships between lagged external factors and the global efficiency during working memory tasks.** For each network, we list only the eternal factor at the lag that yielded significance, along with its correlation value (ρ). All *p*-values have been corrected for multiple comparisons. Unprocessed study data can be found in the Zenodo data set release [175]. Processed results derived from the study data are accessible in the GIT repository [176], under the results folder.
(XLSX)

**S22 Table. Significant relationships between lagged external factors and the global efficiency during working memory tasks.** For each network, we list only the eternal factor at the lag that yielded significance, along with its correlation value (ρ). All *p*-values have been corrected for multiple comparisons. Unprocessed study data can be found in the Zenodo dataset release [175]. Processed results derived from the study data are accessible in the GIT repository [176], under the results folder.
(XLSX)

**S23 Table. Significant relationships between lagged external factors and network estimates during working memory tasks.** These results are derived by using the set2 from Seitzman and colleagues [158] with 10% proportional threshold. For each network, we list only the eternal factor at the lag that yielded significance, along with its correlation value (ρ). All *p*-values have been corrected for multiple comparisons. Unprocessed study data can be found in the Zenodo data set release [175]. Processed results derived from the study data are accessible in the GIT repository [176], under the results folder.
(XLSX)

**S24 Table. Significant relationships between lagged external factors and network estimates during working memory tasks.** These results are derived by including the global signal as a regressor and using the set1 from Seitzman and colleagues [158] with 10% proportional threshold. For each network, we list only the eternal factor at the lag that yielded significance, along with its correlation value ($\rho$). All *p*-values have been corrected for multiple comparisons. Unprocessed study data can be found in the Zenodo data set release [175]. Processed results derived from the study data are accessible in the GIT repository [176], under the results folder. (XLSX)

**S25 Table. Significant relationships between lagged external factors and the global efficiency during resting state.** For each network, we list only the eternal factor at the lag that yielded significance, along with its correlation value ($\rho$). All *p*-values have been corrected for multiple comparisons. Unprocessed study data can be found in the Zenodo data set release [175]. Processed results derived from the study data are accessible in the GIT repository [176], under the results folder. (XLSX)

**S26 Table. Significant relationships between lagged external factors and the participation coefficient during resting state.** For each network, we list only the eternal factor at the lag that yielded significance, along with its correlation value ($\rho$). All *p*-values have been corrected for multiple comparisons. Unprocessed study data can be found in the Zenodo data set release [175]. Processed results derived from the study data are accessible in the GIT repository [176], under the results folder. (XLSX)

**S27 Table. Significant relationships between lagged external factors and network estimates during resting state.** These results are derived by using the set2 from Seitzman and colleagues [158] with 10% proportional threshold. For each network, we list only the eternal factor at the lag that yielded significance, along with its correlation value ($\rho$). All *p*-values have been corrected for multiple comparisons. Unprocessed study data can be found in the Zenodo data set release [175]. Processed results derived from the study data are accessible in the GIT repository [176], under the results folder. (XLSX)

**S28 Table. Significant relationships between lagged external factors and network estimates during resting state.** These results are derived by including the global signal as a regressor and using the set1 from Seitzman and colleagues [158] with 10% proportional threshold. For each network, we list only the eternal factor at the lag that yielded significance, along with its correlation value ($\rho$). All *p*-values have been corrected for multiple comparisons. Unprocessed study data can be found in the Zenodo data set release [175]. Processed results derived from the study data are accessible in the GIT repository [176], under the results folder. (XLSX)

**S29 Table. Between-days movie time-segment classification accuracy is associated with mood and respiration rate patterns experienced over the past 15 days.** These results are derived by shifting the sliding window with 2TRs. We list the t-value and MNI coordinates of the lowest *p*-value. All clusters reported are *p* < 0.05 and have been corrected for multiple comparisons. Unprocessed study data can be found in the Zenodo data set release [175]. Processed results derived from the study data are accessible in the GIT repository [176], under the results folder. (XLSX)

**S30 Table. Between-days movie time-segment classification accuracy is associated with mood and respiration rate patterns experienced over the past 15 days.** These results are derived by shifting the sliding window with 4TRs. We list the t-value and MNI coordinates of the lowest *p*-value. All clusters reported are $p < 0.05$ and have been corrected for multiple comparisons. Unprocessed study data can be found in the Zenodo data set release [175]. Processed results derived from the study data are accessible in the GIT repository [176], under the results folder.
(XLSX)

**S31 Table. Between-days movie time-segment classification accuracy is associated with mood and respiration rate patterns experienced over the past 15 days.** These results are derived by including the global signal as a regressor. We list the t-value and MNI coordinates of the lowest *p*-value. All clusters reported are $p < 0.05$ and have been corrected for multiple comparisons. Unprocessed study data can be found in the Zenodo data set release [175]. Processed results derived from the study data are accessible in the GIT repository [176], under the results folder.
(XLSX)

## Acknowledgments

We acknowledge the computational resources provided by the Aalto Science-IT project.

## Author Contributions

**Conceptualization:** Ana María Triana, Juha Salmi, Jari Saramäki, Enrico Glerean.

**Data curation:** Ana María Triana, Enrico Glerean.

**Formal analysis:** Ana María Triana.

**Funding acquisition:** Ana María Triana, Enrico Glerean.

**Investigation:** Ana María Triana, Nicholas Mark Edward Alexander Hayward, Enrico Glerean.

**Methodology:** Ana María Triana, Juha Salmi, Jari Saramäki, Enrico Glerean.

**Project administration:** Ana María Triana.

**Resources:** Jari Saramäki.

**Software:** Ana María Triana.

**Supervision:** Juha Salmi, Nicholas Mark Edward Alexander Hayward, Jari Saramäki, Enrico Glerean.

**Visualization:** Ana María Triana.

**Writing – original draft:** Ana María Triana.

**Writing – review & editing:** Juha Salmi, Nicholas Mark Edward Alexander Hayward, Jari Saramäki, Enrico Glerean.

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
