## [Editor Report · Decision Letter 0]

25 Apr 2022

Dear Dr Triana, 

Thank you for submitting your manuscript entitled "Effects of daily environmental, physiological, and lifestyle factors in functional brain connectivity" for consideration as a Preregistered Research Article by PLOS Biology.

Your manuscript has now been evaluated by the PLOS Biology editorial staff, as well as by an academic editor with relevant expertise and an academic editor focused on specifically evaluating Preregistered Research Articles. I am writing to let you know that we would like to send your submission out for external peer review.

Prior to full submission, we ask that you address some of the conceptual framing concerns that have been raised by the Academic Editors. Specifically, both felt that the current proposal has the potential to provide useful information about daily behavioral factors that covary with individual brain variability. However, they expressed concern that the specific hypotheses laid out do not currently take full advantage of the individualized project design, tending to overlap with ideas already in the literature based on group studies, and that the broad nature of the hypotheses makes it difficult to understand how specific outcomes will help to confirm/refute these ideas. As it stands, there was concern expressed that any activity within these large-scale networks could be viewed a supporting your hypotheses. We therefore ask that you spend time developing a more precise breakdown of predicted activity under specific conditions and outcomes and how this will confirm or disconfirm each hypothesis. 

Please also move the power analysis into the main text, expanding upon this analysis while also making it clear that this is an N=1.

Finally, please note the open access data availability requirements at PLOS Biology and ensure that your submission adheres to these guidelines:

https://plos-marketing.s3.amazonaws.com/Marketing/Biology+Preregistered+Articles+Guidelines+for+Authors.pdf

https://journals.plos.org/plosbiology/s/data-availability

Before we can send your manuscript to reviewers, we will also need you to complete your submission by providing the metadata that is required for full assessment. To this end, please login to Editorial Manager where you will find the paper in the 'Submissions Needing Revisions' folder on your homepage. Please click 'Revise Submission' from the Action Links and complete all additional questions in the submission questionnaire.

Once your full submission is complete, your paper will undergo a series of checks in preparation for peer review. Once your manuscript has passed the checks it will be sent out for review. To provide the metadata for your submission, please Login to Editorial Manager (https://www.editorialmanager.com/pbiology) within one week.

If your manuscript has been previously reviewed at another journal, PLOS Biology is willing to work with those reviews in order to avoid re-starting the process. Submission of the previous reviews is entirely optional and our ability to use them effectively will depend on the willingness of the previous journal to confirm the content of the reports and share the reviewer identities. Please note that we reserve the right to invite additional reviewers if we consider that additional/independent reviewers are needed, although we aim to avoid this as far as possible. In our experience, working with previous reviews does save time. 

If you would like to send previous reviewer reports to us, please email me at kdickson@plos.org to let me know, including the name of the previous journal and the manuscript ID the study was given, as well as attaching a point-by-point response to reviewers that details how you have or plan to address the reviewers' concerns. 

Kind regards,

Kris

Kris Dickson

Neurosciences Senior Editor/Section Manager

PLOS Biology

kdickson@plos.org

---

## [Decision Letter · Decision Letter 1]

14 Jul 2022

Dear Dr Triana,

Thank you for your patience while your manuscript "Effects of daily environmental, physiological, and lifestyle factors in functional brain connectivity" was peer-reviewed at PLOS Biology. It has now been evaluated by the PLOS Biology editors, an Academic Editor with relevant expertise, and by three independent reviewers. 

In light of the reviews, which you will find at the end of this email, we would like to invite you to revise the work to thoroughly address the reviewers' reports. As you will see below, the reviewers are broadly positive about your proposed study, they each have a number of requests and suggestions that will need addressing before further consideration.

Given the extent of revision needed, we cannot make a decision about publication until we have seen the revised manuscript and your response to the reviewers' comments. Your revised manuscript is likely to be sent for further evaluation by all or a subset of the reviewers.

**IMPORTANT - SUBMITTING YOUR REVISION**

*Re-submission Checklist*

*Published Peer Review*

*PLOS Data Policy*

*Blot and Gel Data Policy*

Sincerely,

Roli Roberts

Roland G Roberts PhD

Senior Editor

PLOS Biology

on behalf of

Kris Dickson, Ph.D. (she/her)

Neurosciences Senior Editor/Section Manager

PLOS Biology

kdickson@plos.org

REVIEWERS' COMMENTS:

Reviewer #1:

This manuscript presents a pre-registered report of a study with deep phenotyping, monitoring, and brain imaging in a single individual. This approach is an important direction for future research, and the dataset generated from this study is exciting. The collection of daily behavioral data via wearables and smartphone questionnaires along with frequent neuroimaging is novel and may provide a more reliable approach to measuring brain-behavior associations. I commend the authors for taking on this project. 

I have some comments and questions for the authors to address: 

1. It is not clear why all aspects of hypotheses 3 and 4 are specific to resting state fMRI and movie watching, respectively. For example, why is mood part of H4, but not H3? Would you predict that mood would be associated with resting state vs. movie watching differently?

2. Given how well this subject holds still in the scanner, it is not clear why more conservative methods for accounting for motion aren't planned. The proponents of strict motion methods in the field are cited in this paper (e.g., Power et al. 2014), yet the recommended criteria are not planned. For example, motion censoring is often recommended for volumes exceeding FD=0.2mm and global signal regression is touted as best accounting for motion (Power et al. 2014, Ciric et al. 2017, 2018, Satterthwaite et al. 2019). The authors state that they will not censor volumes from the movie watching scans because the proposed analyses require a consistent number of volumes for each day. A straightforward approach would be to censor volumes at a threshold of FD < 0.2mm, identify the scan day with the least number of volumes remaining, and then randomly censor additional volumes on the other days in order to match the numbers across days. Given how well this subject holds still, it is unlikely that this will yield too much data loss. But, it will ensure that the few high motion points do not contaminate the data. In any case, the authors should propose to test more strict processing methods in addition to the currently planned methods. 

Regarding global signal regression, I understand this is a contentious topic in the field. Therefore, I recommend that the authors propose to analyze the data both with and without global signal regression. The carpet plots in the supplement actually give some support for using global signal regression. They clearly show the beneficial effects of denoising, but there is still some banding where motion affected the signal across the whole brain. This is why the author of the paper cited describing the carpet plots (Power 2017) recommends global signal regression. 

3. The authors state that the adjacency matrices will be "subdivided into smaller network-specific matrices according to Power and colleagues138." There is a lot that goes into this analysis, so more detail is needed. For example, do the authors plan to use the Infomap algorithm? If so, what thresholds will you use? How do you propose to create consensus communities? There are more updated and improved methodological details for these approaches as well, such as that reported in Gordon et al. 2017, which is probably more appropriate for the proposed study. I recommend that the authors describe the details of these planned analyses. 

Minor:

1. I recommend using the term "Precision functional mapping (PFM)" instead of FPM to be consistent with the literature (e.g., Gordon et al. 2017). 

2. Pg. 3, line 109: what is meant by "exacerbates the performance in attentional task"?

3. There is likely a typo on pg. 3, line 117 "sustained ascending arousal from the thalamus." 

4. What is the reason for using two versions of the PVT (10 min in the scanner and 5 min out of the scanner)? 

5. "MRI will be performed in a fixed schedule, Mondays and Fridays at 1400 hours, subject to the scanner availability." How likely are the experimenters to follow this schedule given the typical availability of the scanner? 

6. What is the reason for having a fixed order of scans rather than counterbalancing the order of the different scans?

7. If possible, collecting more data is always better, especially for resting state fMRI and movie watching. How long are the proposed movie scans? Can another resting state fMRI scan be added to each session? How long is the scan run for the N-back task?

8. For the task fMRI data, do the authors plan to analyze the data in the more standard event-related way as well? You may have more power to detect effects with such an approach than looking at functional connectivity during tasks. 

9. Pg. 17, where does the 256 x 256 number come from? Power et al. 2011 was 264 regions, and the authors propose 246 regions prior to this. 

10. Be consistent with the tense used in the text. 

Reviewer #2:

This Initial Research Submission describes a planned dense phenotyping study in which a single individual will participate in 30 MRI scan sessions across 15 weeks as well as behavioral data collection sessions for 19 weeks. Rest, movie, and attention and working memory task data will be collected during fMRI. Smartphones and wearable devices will also record data about factors such as physical activity and sleep. Overall this is an exciting data collection effort and a rigorous preregistration plan. Although the hypotheses themselves are somewhat diffuse and varied, I am enthusiastic about the project and its potential to inform understanding of within-subject relationships between brain activity and functional connectivity, attention and working memory, and sleep and lifestyle factors. I have some questions and comments below that, if addressed, may strengthen the plan.

Major comments

1. Hypotheses 1 and 2 refer to brain activity. How will brain activity be operationalized here? The methods provide detail about the planned functional connectivity and intersubject correlation analyses and would benefit from additional detail about planned activation analyses. 

2. The manuscript includes a well-thought-out scan plan and detailed description of the tasks. Although I understand the justification for including a working memory task, what is the benefit of using an adaptive design? Stimulus timing parameters that differ across days due to staircasing could introduce confounds in data analyses. In addition, recent work has found that adaptive tasks, such as the stop-signal task (SST) and monetary incentive delay (MID) task, have poor reliability within-subjects (e.g., Rapuano et al., 2022, NeuroImage, An open-access accelerated adult equivalent of the ABCD Study neuroimaging dataset (a-ABCD); Kennedy et al., 2021, Reliability and stability challenges in ABCD Task fMRI data). It would be helpful to include additional rationale for using an adaptive design or consider a non-adaptive version. Related to point (1), I also suggest considering block-contrasts for measuring session-specific brain activity (rather than event-related regressors); Rapuano et al. and Kennedy et al. both show very poor within-subject reliability of trial-evoked activation patterns in the SST and MID. 

3. How will the analyses using PVT and n-back task data account for day-to-day differences in task performance? Will brain activity/connectivity changes be related to changing performance for the PVT and staircasing for the n-back task? Will performance be used as a confound regressor? It is not clear whether hypotheses 1 and 2 also include predictions about how task performance will relate to fluctuations in sleep and physical activity, and how this will be tested or taken into account in fMRI analyses.

4. The inclusion of naturalistic data is a strength of this study. Previous work, however, suggests that participants' engagement with naturalistic stimuli such as movies modulates ISC (e.g., Son et al., 2021, PNAS). Given that the participant will watch the same movie 30 times, they will probably be less and less engaged by the story over time, likely affecting ISC. Does this change the projected number of sessions needed to see stable ISC ("For naturalistic stimuli previous work(111) has established that 30 scans are within the optimal number of samples for the ISC statistics to converge."), since this previous work was based on participants' first viewing of a naturalistic stimulus, and here repeated viewing may significantly impact the individual's cognitive state dynamics during the movie? Is it feasible to measure the participant's engagement with the movie (e.g., with eye tracking physiological measures, and/or subjective report)? 

5. Did the Aalto University Research Ethics Committee approve the participant's participation without compensation? Compensation is often required by review boards.

6. The data availability statement states that, "The data use agreement will restrict potential reuses of the data by the receiving party, as the only purpose for receiving the data is peer-review of this registered report." Will DUAs also restrict data re-use for groups that request these data when the final paper is published? The dataset will be a valuable contribution of this study, and clarity about when and how it will be available to the scientific community will be useful. 

Minor comments

1. It could increase readability to avoid non-standard acronyms such as "FPM".

2. In the section, "Inter-daily Representational Similarity Analysis," the authors write that the: "behavior similarity matrix will be computed using the Anna-Karenina structure". Is there a reason to expect the AnnaK model to best represent cross-day similarity here? It would be useful to compare the AnnaK model to another model such as nearest neighbors to empirically test this hypothesis.

3. Previous work has censored time-points in fMRI data collected during naturalistic tasks (e.g., Finn et al., 2018, Nat Commun), so this could be an option to reduce the influence of high- motion frames on movie-watching data analyses. 

Reviewer #3:

This is a pre-registered study that plans to collect and analyze richly sampled neuroimaging, physiological, and behavioral measures from a single individual over 133 days. As studies of this nature are rare, this work and the accompanying dataset are likely to generate valuable information about the links between brain connectivity measures and individual variability (though generalizability to other subjects will need further investigation). To my understanding, several of the current hypotheses are based on prior cross-sectional studies, but have not been shown to directly extend to day-to-day variations in an individual subject, which this work would address. Hypotheses 4 and 5 are more specific to the proposed design. Specific comments and questions are below:

- With regard to one of the editors' previous comments, it seems that still more may be done in terms of formulating hypotheses that leverage the unique longitudinal design (for instance, looking at effects over different time-scales).

- The MRI tasks will be carried out in the same order for each scan. I wondered why the decision was not made to counterbalance MRI tasks, in light of potential systematic effects over the course of the scan (such as drowsiness)?

- The motivation for selecting participation coefficient and global efficiency as the two graph metrics of interest wasn't clear to me, and it would help to provide the rationale. 

- Hypothesis 2 relates to changes in several brain networks during a working memory task. The working memory task proposed here is an adaptive n-back task, aimed at keeping overall performance constant. However, I would assume that since behavior will vary across days, so will the adaptive difficulty levels, which may be important to account for in the analyses. I didn't find this mentioned. 

- In the section on quality control, it would be helpful to provide information about quality assurance of variables acquired in the scanner, such as the respiratory, pulse, and eye-tracking measures, which can be susceptible to artifacts.

- The subject will be instructed to keep eyes open during the tasks, but eye-closures might sill occur. Since eye-tracking data will be collected, including a regressor to model the eye opening/closing may benefit the connectivity analysis in this study.

- In lines 596-598, it is mentioned that spatial smoothing will be applied only to the movie-watching task. Why apply this step only to the movie scans?

- For the exploratory Hypothesis 5 (time-segment classification), I believe the results will be averaged across temporal segments to create one map per scan? If so, it would seem there is also an opportunity to examine variability in the processing of different scenes in the movie; the authors could consider expanding the analysis. 

- Supplementary Fig 5(panel e) indicate a drop-off of correlation with global signal in anterior brain regions. A lower SNR in prefrontal regions is also mentioned by the authors, and while it is good that the tSNR values are comparable to some other studies, this may still be an overall limitation of the data acquisition. 

Minor comments:

- In the Experimental overview, it is noted that the subject will undergo MRI sessions.. and "simultaneously", smartphones and wearables will be used. This could be misinterpreted by some readers, suggesting that the latter will be collected during the MRI scans themselves, so rephrasing may be helpful.

- line 624, "Shortly," (?) another word may be more suitable

- line 519, "menstrual cycle data will be binarized.." - in what way would this variable be binarized?

- line 312, auditory and visual gratings -> auditory tones and visual gratings ?

- line 313, trial -> trials

- line 487, "..volumes.. with high movement will be discarded from the time series because the adjacency matrix does only depend on one run" - was not clear what was meant here.

---

## [Decision Letter · Decision Letter 2]

30 Sep 2022

Dear Dr Triana,

Thank you for your patience while we considered your revised manuscript "Effects of daily environmental, physiological, and lifestyle factors in functional brain connectivity" for publication as a Preregistered Research Article at PLOS Biology. This revised version of your manuscript has been evaluated by the PLOS Biology editors, our two Academic Editors and the original reviewers. Based on the reviews, we are likely to accept this manuscript for publication, provided you satisfactorily address the remaining points raised by the reviewers. 

We expect to receive your revised manuscript within two weeks. 

*Published Peer Review History*

*Press*

Please do not hesitate to contact me should you have any questions, or if you feel you will need additional time to address the remaining concerns.

Sincerely,

Kris

Kris Dickson, Ph.D. (she/her)

Neurosciences Senior Editor/Section Manager,

kdickson@plos.org,

PLOS Biology

Reviewer remarks:

Reviewer #1: The authors addressed all my comments. I only have 2 minor points remaining:

Regarding point #2 about censoring at FD>0.2mm, the authors propose to use a threshold of 0.2mm (great!), but if more than 20% of the data exceeds that threshold, they will raise it to 0.5mm. I don't think this 20% criteria is necessary. Increasing the FD to 0.5mm will only lead to including submillimeter movements that cause artifact. We are in agreement that this is unlikely to occur anyway, but you should still have plenty of data even if there are some sessions with more than 20% data loss.

Regarding point #3, using the Power ROIs seems fair. Of course, note that the ROIs are not parcels, but spheres. Might I suggest proposing to also run analyses using the Gordon 333 parcels (Gordon et al., 2016 Cerebral Cortex) plus additional subcortical & cerebellar ROIs, which can be defined anatomically (e.g., Freesurfer) or functionally (as in Seitzman et al. 2020, A set of functionally-defined brain regions with improved representations of the subcortex and cerebellum. NeuroImage). This would provide an updated sampling of functional areas across the whole brain. 

Reviewer #2: I thanks the authors for addressing my comments and am interested in seeing the outcome of this work. As a minor note, I suggest specifying "functional connectivity" throughout (rather than "brain connectivity", which could be interpreted as structural connectivity). I also find the wording "Fluctuations up to fifteen days..." somewhat confusing and suggest clarifying the wording. Does this mean the fluctuations are measured in a time series with 15 observations, or that the period of the fluctuations is 15 days?

Reviewer #3: I believe the authors' revisions have helped to improve the clarity and quality of the manuscript. I have only a few additional minor suggestions on the revised material:

- As suggested, the authors have now included a potential analysis to examine timescales of relationships between behavioral and neuroimaging data. The modified Q1 states "how is functional brain connectivity influenced by previous day behavioral….?" However, since (to my understanding) a range of days will be queried in the cross-correlation analysis, this wording could be clarified to convey that the analysis considers past history longer than one day.

- I thank the authors for clarifying the rationale for not counterbalancing the scans. While the description in the reply to reviewers is more extensive, the explanation added to the manuscript was very brief. Expanding a bit on the rationale may be helpful for readers who may have the same question. 

- H6-H7: "leveraged on" -> "leveraged"

- Line 578: typo, "different hour or day of the day" 

- Line 818: "Despite we intend to" -> "Despite our intention to"

---

## [Editor Report · Decision Letter 3]

28 Oct 2022

Dear Dr Triana,

Thank you for your patience while your revised Pre-Registered Research Article "Effects of daily environmental, physiological, and lifestyle factors on functional brain connectivity" was re-reviewed for PLOS Biology. Your Stage 1 manuscript has been evaluated by the PLOS Biology editors and by the two Academic Editors. This latest version was evaluated in-house and was not sent back out to the reviewers.

As both of our Academic Editors are satisfied with these additional revisions, we are happy to issue a Stage 1 'in-principle acceptance' decision, with a commitment to publish the final Stage 2 Preregistered Research Article (after revision, if needed), pending successful completion of the study. Please carefully read all the following information.

The study should now be completed according to the Stage 1 approved methods and analytic procedures, and the final manuscript should include an evidence-based interpretation of the results. Please see the review criteria for Stage 2 manuscripts here:

https://journals.plos.org/plosbiology/s/reviewer-guidelines#loc-reviewing-preregistered-research-articles

Subsequent editorial decisions for this study will not be based on the perceived importance or novelty of the results obtained during the data gathering and analysis phase of the work. It is critical however that you adhere to the approved Stage 1 study design when performing the study. Any deviation from these experimental procedures would need to be justified and approved by the editors (and potentially the reviewers), as otherwise it could lead to rejection of the manuscript at Stage 2. Please consult the editors immediately for advice if you need to alter this approved study plan.

**IMPORTANT**: Please follow the link below for important information regarding the Stage 2 manuscript template and review criteria. Please carefully read the guidelines on Stage 2 data collection BEFORE performing your study and completing your Stage 2 manuscript. 

AUTHOR GUIDELINES: https://genweb.plos.org/Marketing/Biology%20Preregistered%20Articles%20Guidelines%20for%20Authors.pdf

*Depositing this Stage 1 Protocol*

PLOS Biology does not publish Stage 1 Protocols immediately following an in-principle acceptance. Instead they are held and integrated into a single, completed 'Preregistered Research Article' following review and acceptance of the final Stage 2 manuscript. You are however required to register this approved Stage 1 Protocol with the Center for Open Science (https://cos.io/prereg/) or another recognised repository. This may be done publicly or under private embargo until submission of the Stage 2 manuscript. Stage 1 Protocols can be quickly and easily registered using a tailored mechanism for Registered Reports (https://osf.io/rr/). Please do this now. You will need to include the URL to this deposited protocol in your Stage 2 manuscript.

*Timeline*

We understand that carrying out the study will require a significant length of time and are willing to allow you 6 months to perform the study and analyze the resulting data. Please email us at plosbiology@plos.org to discuss this if you have any questions or concerns, or to discuss an alternate timeline.

At this stage, your manuscript remains formally under active consideration at our journal. Please notify us by email if you do not wish to submit a Stage 2 manuscript or wish to pursue publication elsewhere, so that we may withdraw your manuscript. 

*Resubmission Checklist*

Before submitting the Stage 2 manuscript, please review the following resubmission checklist: https://plos.io/Biology_Checklist

Please note that for PRA stage 2, the response to reviewers file does not follow the standard format, but should rather be a document for the reviewers detailing the changes made to the manuscript since the stage 1 accept.

*Published Peer Review*

*PLOS Data Policy*

Please note that as a condition of publication, PLOS' data policy (http://journals.plos.org/plosbiology/s/data-availability) requires that you make available all data used to draw the conclusions arrived at in your manuscript. Please note that for this article type, the raw data itself should be archived and made freely available in a public repository rather than submitted as supplementary material. Please make sure to read the Stage 2 submission guidelines online regarding how this data should be annotated and appropriately time stamped to show that data was collected after this Stage 1 in-principle acceptance and not before.

*Blot and Gel Data Policy*

To enhance the reproducibility of your results, we recommend that, if applicable, you deposit your laboratory protocols in protocols.io, where a protocol can be assigned its own identifier (DOI) such that it can be cited independently in the future. For instructions see: https://journals.plos.org/plosbiology/s/submission-guidelines#loc-materials-and-methods

Thank you again for your submission to PLOS Biology. We hope that our editorial process has been constructive thus far, and we welcome your feedback at any time. Please don't hesitate to contact us if you have any questions or comments.

Sincerely,

Kris Dickson, Ph.D., (she/her)

Neurosciences Senior Editor/Section Manager

PLOS Biology

kdickson@plos.org

---

## [Decision Letter · Decision Letter 4]

16 Apr 2024

Dear Ana Triana,

Thank you for your patience while we considered your revised manuscript "Effects of daily environmental, physiological, and lifestyle factors on functional brain connectivity" for consideration as a Preregistered Research Article at PLOS Biology. Your revised study has now been evaluated by the PLOS Biology editors, the Academic Editors, and the original reviewers. 

In light of the reviews, which you will find at the end of this email, we are pleased to offer you the opportunity to address the comments from the reviewers in a revision that we anticipate should not take you very long. We will then assess your revised manuscript and your response to the reviewers' comments with our Academic Editor aiming to avoid further rounds of peer-review, although might need to consult with the reviewers, depending on the nature of the revisions.

In particular, we would like to flag the departure from the Stage 1 manuscript and encourage you to clarify why so much was lost at the stringent cut-off, which seems a bit surprising. Please also address the other concerns mentioned by the reviewers.

In addition, we would like you to address the following editorial comments:

* We would like to suggest a different title to improve readability/accuracy: "Longitudinal single-subject neuroimaging study reveals effects of environmental, physiological and lifestyle factors on functional brain connectivity"

* Please include information about the form of consent (written/oral) given for research involving human participants and if the study has been conducted according to the principles expressed in the Declaration of Helsinki.

DATA POLICY:

Regardless of the method selected, please ensure that you provide the individual numerical values that underlie the summary data displayed in the following figure panels as they are essential for readers to assess your analysis and to reproduce it: SI Figures 6E, 10, 15 and 41. 

CODE POLICY

Per journal policy, if you have generated any custom code during the curse of this investigation, please make it available without restrictions upon publication. Please ensure that the code is sufficiently well documented and reusable, and that your Data Statement in the Editorial Manager submission system accurately describes where your code can be found. [IF APPLICABLE: As the code that you have generated to XXX is important to support the conclusions of your manuscript, its deposition is required for acceptance.]

**IMPORTANT - SUBMITTING YOUR REVISION**

*Resubmission Checklist*

*Published Peer Review*

*PLOS Data Policy*

*Blot and Gel Data Policy*

Sincerely,

Christian

Christian Schnell, PhD

Senior Editor

PLOS Biology

cschnell@plos.org

REVIEWS:

Reviewer #1: It was a pleasure to see this previously pre-registered report carried out and to read the results in this manuscript. I commend the authors on an ambitious study that has great potential to move the field forward. I was also heartened that the authors took my previous comments into consideration when running their study. Overall, the manuscript is well written and thorough. I have several comments/questions for the authors to address. 

1. Perhaps I am confused about the motion scrubbing threshold, so some clarification is warranted. If I am understanding correctly, an FD threshold of 0.5mm was chosen because a lower threshold (0.2mm) removed too much data. However, in the pilot study an FD threshold of 0.2mm removed very little data. Why would the percentage of data removed at this threshold change so significantly from the pilot study to this study on the same participant? Did something change? When examining the carpet plots as in Power et al., is there no observable motion artifact at timepoints during which there is a spike in FD above 0.2 but below 0.5? 

2. Given previous precision functional mapping papers showing individual variability in functional connectivity and network assignments in particular brain regions, do the authors think that their results are affected by the use of canonical, group-defined ROI's? Would you expect differences in the results if you used individually defined ROI's for your participant? Ideally, one would use individually-defined ROI's. However, identification of these ROI's is not trivial, and so it is understandable if running such analyses is outside the scope of this first study with this dataset. Still, it would be interesting to hear the authors thoughts on how their results may be affected by group-defined vs. individually-defined ROI's. 

3. The authors should consistently report the precise statistics from their analyses throughout the results (i.e., statistic and p-value, not just that the result was significant or not). There are very few instances of actual statistics reported. 

4. I suggest the authors discuss their results in the context of the findings from Gratton et al. (2018), which demonstrated that individual variability in functional connectivity was predominantly driven by individual-specific features that were largely stable over time. While contributions of day-to-day (session) and task state (e.g., rest vs. task) variation were measurable, the effects were much smaller than those of stable individual features. Thus, it is important for the authors discuss their findings of within-individual day-to-day variability in this context. 

5. In the Discussion, when using words like "efficiency" and "integration", please operationalize what these terms mean, i.e., do they simply mean increased functional connectivity? 

6. The Ethics Information section still has "will" statements from the pre-registered report. 

Reviewer #2: I applaud the authors for an impressive data collection and analysis effort. The paper is dense with many analyses, but this is justified given the preregistration. Many open questions remain about the specificity of findings to the task types and networks tested here, but I do not recommend adding additional analyses to test them. 

My evaluation of this stage 2 submission is below.

1) Whether the introduction, rationale and stated hypotheses are the same as the approved Stage 1 Protocol submission.

Yes

2) Whether the authors adhered precisely to the approved Stage 1 experimental procedures.

Yes. The minor changes to the frame-censoring approach and sleep variables used are sufficiently justified. 

3) Whether the data are able to test the authors' proposed hypotheses by satisfying the approved outcome-neutral conditions (such as quality checks, positive controls).

Yes

4) Whether the authors' conclusions are justified given the data.

Partially. The tests of question 1 are generally convincing in showing that yesterday's experiences affect today's functional connectivity patterns. I was unsure exactly how permutation testing and multiple comparisons correction were performed and suggest adding this to the methods if appropriate at this stage of the submission process.

The tests of question 2 left me less convinced of robust relationships between experiences during specific days in the past and today's functional connectivity patterns. The effects appeared less replicable across brain atlas and preprocessing pipelines and the huge number of multiple comparisons raises concerns about false positive results. It would be useful to specify exactly how many tests are being performed.

There are also a number of unexplored explanations for the results. For example, is there autocorrelation in the sleep, mood, and activity data? Could what appears to be an effect of past sleep on current FC be the result of, for example, slowly accruing sleep debt over time? Could what looks like an effect of last Monday's mood on this Monday's FC arise because mood tends to be consistent on Mondays? My concrete suggestion is to tone down the interpretation of these effects given their instability and the range of possible explanations (other than the one the authors focus on, which is a direct effect of variable x on day y on day z's FC).

5) Whether any post-hoc analyses added by the authors are justified, methodologically sound, and informative.

N/A 

6) Whether, in accordance with the PLOS' data availability policy, data has been made freely available in a public repository. Data files should be appropriately time stamped to show that data was collected after Stage 1 Protocol approval and not before.

Yes

Other comments:

There are a few places in the text where the future tense of the stage 1 submission was not changed to past tense (e.g., "Wearables and smartphone data were collected two weeks before the MRI sessions [began] and two weeks after the MRI sessions end[ed].")

Consider replacing the phrase "sanity check" with something such as "data check".

I believe that "COP" and "CON" are used at different times to refer to the same functional network. 

In H4, please unpack the direction of the observed Anna Karenina effect. Do sessions with higher or lower PAF scores show higher/lower ISC?

In the Q2 results section, the text slips into describing functional connectivity as "communication" or "information transfer" between nodes. This is not justified as functional connectivity does not necessarily reflect communication or shared information between regions; it could reflect shared noise for example.

---

## [Decision Letter · Decision Letter 5]

15 Jul 2024

Dear Dr Triana,

Thank you for your patience while we considered your revised manuscript "Longitudinal single-subject neuroimaging study reveals effects of daily environmental, physiological, and lifestyle factors on functional brain connectivity" for publication as a Preregistered Research Article at PLOS Biology. This revised version of your manuscript has been evaluated by the PLOS Biology editors, the Academic Editor and the original reviewers.

Based on the reviews and on our Academic Editors' assessment of your revision, we are likely to accept this manuscript for publication, provided you satisfactorily address the remaining points raised by the reviewers. Please also make sure to address the following data and other policy-related requests:

* I could not find a reference/link to the deposited Stage 1 manuscript anywhere in the paper. Can you please add this reference?

* I realized there is one ',' too much in the title. Can you please delete the last ',' after 'physiological'?

* DATA POLICY:

Regardless of the method selected, please ensure that you provide the individual numerical values that underlie the summary data displayed in the following figure panels as they are essential for readers to assess your analysis and to reproduce it: S6E, S10, S15 and S41.

CODE POLICY

We expect to receive your revised manuscript within two weeks. 

*Published Peer Review History*

*Press*

Sincerely,

Christian

Christian Schnell, PhD

Senior Editor

cschnell@plos.org

PLOS Biology

Reviewer remarks:

Reviewer #1: The authors have addressed all comments.

Reviewer #2: The authors have address my comments. I leave the decision to the authors and editor, but personally prefer removing the word "study" from the updated title.

I have two minor suggestions to the text.

1) "To understand the effect of the previous day's sleep, mood, and ANS activity patterns on resting-state functional connectivity, we run a series of regression models on the unthresholded link-weights within the DMN, FPN, and CON." 

I believe "run" should be changed to "ran".

2) "The majority of the findings showed a positive correlation, indicating that more hours of sleep, awake time in bed, and interruptions during sleep are associated with an increase in brain networks estimates." Please specify which brain network estimates. Does this refer to estimates of global efficiency?

---

## [Editor Report · Decision Letter 6]

8 Aug 2024

Dear Dr Triana,

Thank you for the submission of your revised Preregistered Research Article "Longitudinal single-subject neuroimaging study reveals effects of daily environmental, physiological and lifestyle factors on functional brain connectivity" for publication in PLOS Biology. On behalf of my colleagues and the academic editors, Laura Lewis and Chris Chambers, I am pleased to say that we can in principle accept your manuscript for publication, provided you address any remaining formatting and reporting issues. These will be detailed in an email you should receive within 2-3 business days from our colleagues in the journal operations team; no action is required from you until then. Please note that we will not be able to formally accept your manuscript and schedule it for publication until you have completed any requested changes.

PRESS

Sincerely, 

Christian

Christian Schnell, PhD

Senior Editor

PLOS Biology

cschnell@plos.org